# DGD$^2$: A Linearly Convergent Distributed Algorithm For High-dimensional Statistical Recovery

**Marie Maros**
School of Industrial Engineering
Purdue University
mmaros@purdue.edu

**Gesualdo Scutari**
School of Industrial Engineering
Purdue University
gscutari@purdue.edu

## Abstract

We study linear regression from data distributed over a network of agents (with no master node) under high-dimensional scaling, which allows the ambient dimension to grow faster than the sample size. We propose a novel decentralization of the projected gradient algorithm whereby agents iteratively update their local estimates by a "double-mixing" mechanism, which suitably combines averages of iterates and gradients of neighbouring nodes. Under standard assumptions on the statistical model and network connectivity, the proposed method enjoys global *linear* convergence up to the *statistical precision* of the model. This improves on guarantees of (plain) DGD algorithms, whose iteration complexity grows undesirably with the ambient dimension. Our technical contribution is a novel convergence analysis that resembles (albeit different) algorithmic stability arguments extended to high-dimensions and distributed setting, which is of independent interest.

## 1 Introduction

Consider M-estimation over a network of $m$ agents, modeled as an undirected graph with no server (termed *mesh* network). Each agent $i$ owns a sample of $n$ i.i.d observations, drawn from an unknown, common distribution on $\mathcal{Z} \subseteq \mathbb{R}^p$, and collected in the set $S_i$. Agents aim at minimizing the total empirical loss over $N = m \cdot n$ samples, resulting in the following empirical risk minimization (ERM):

$$\hat{\theta} \in \operatorname*{argmin}_{\theta \in \Omega : \mathcal{R}(\theta) \leq R} \mathcal{L}(\theta) \triangleq \frac{1}{m} \sum_{i=1}^{m} \mathcal{L}_i(\theta), \quad \text{with} \quad \mathcal{L}_i(\theta) \triangleq \frac{1}{2n} \sum_{j \in S_i} (x_j^\top \theta - y_j)^2, \tag{1}$$

where $\mathcal{L}_i(\theta)$ is the empirical loss of agent $i$, with $x_j \in \mathbb{R}^d$ being the vector of predictors and $y_j \in \mathbb{R}$ the associated response; $\mathcal{R}$ is a (convex) regularizer (with $R > 0$) controlling the complexity of the solution; and $\Omega$ is some (convex) subset of $\mathbb{R}^d$.

The ERM (1) is a surrogate of the minimization of the (strongly convex) population loss

$$\theta^\star = \operatorname*{argmin}_{\theta \in \mathbb{R}^d} \bar{\mathcal{L}}(\theta) \triangleq \frac{1}{2} \mathbb{E}_{(x,y) \sim \mathbb{P}}[(x^\top \theta - y)^2]. \tag{2}$$

We are interested in estimation problems that are underdetermined, meaning that the ambient dimension $d$ exceeds (and grows faster than) the total sample size $N$. Therefore, we assume that $\theta^\star$ lies in a smaller subset of $\Omega$ or is well approximated by a member of it–the regularizer $\mathcal{R}$ in (1) enforces this constraint. Instances of such M-estimation problems include $\ell_1$-constrained sparse linear models, low-rank matrix recovery via nuclear norm, and matrix regression with soft-rank constraints [2].

Our goal is to study statistical *and* computational guarantees of first-order distributed algorithms for (1). The benchmark is the centralized Projected Gradient Algorithm (PGA) [2]: under suitably

restricted notions of strong convexity and smoothness of $\mathcal{L}$ (see Sec. 2)–which hold with high probability for a variety of statistical models–as well as conditions for statistical consistency–e.g., $s \log d/N = o(1)$ for $s$-sparsity–the iterates $\{\theta^t\}$ generated by the PGA (starting from $\theta^0$) satisfy:

$$\|\theta^t - \hat{\theta}\|^2 \leq \lambda^t \|\theta^0 - \hat{\theta}\|^2 + o(\|\hat{\theta} - \theta^\star\|^2), \tag{3}$$

with $\lambda \in (0, 1)$. In words: the optimization error $\|\theta^t - \hat{\theta}\|^2$ decays *linearly* with rate $\lambda$, up to a tolerance of a smaller order than the statistical error of the model, $\|\hat{\theta} - \theta^\star\|^2$. Therefore every limit point of $\{\theta^t\}$ is within the statistical error from $\theta^\star$. This is the the best one can hope for statistically (ignoring lower order terms) and computationally (within first-order, non accelerated methods).

The PGA is not implementable on mesh networks: agents cannot compute locally the full gradient $\nabla \mathcal{L}$. This calls for the design of distributed algorithms. The following questions arise naturally:

**(Q1)** Can (3) be achieved by distributed algorithms? Of primary interest is the regime wherein the local sample size $n$ is not sufficient for statistical consistency while the total one $N$ is so.

**(Q2)** How do sample rate and convergence rate scale with the model parameters ($d$, $n$, $N$) and network parameters (connectivity, size $m$, and topology)?

As documented next, current literature does not provide a satisfactory answer to the above questions for the majority of distributed algorithms. In a nutshell, most of existing studies are of pure optimization type, they lack statistical analysis, and break down under high-dimensional scaling $d/N \to \infty$.

## 1.1 Related works

The ERM (1) is an instance of a (non strongly) convex optimization problem–several distributed algorithms are applicable. Early decentralizations of the PGA have been proposed in [18, 19]; when applied to (1), they reduce to the following distributed gradient descent (DGD):

$$\theta_i^{t+1} = \prod_{\Omega \cap \{\mathcal{R}(\theta) \leq R\}} \left( \sum_{j \in \mathcal{N}_i} \tilde{w}_{ij} \, \theta_j^t - \alpha \nabla \mathcal{L}_i(\theta_i^t) \right), \quad i = 1, \ldots, m, \tag{4}$$

where $\theta_i^t$ is the estimate from agent $i$ of the common variable $\theta$ at iteration $t$; $\alpha \in (0, 1]$ is the stepsize; $\mathcal{N}_i$ is the set of neighbors of agent $i$; $\tilde{w}_{ij}$'s are suitably chosen nonnegative weights; and $\prod_{\Omega \cap \mathcal{R}(\theta) \leq R}(\bullet)$ is the Euclidean projection of its argument onto the convex set $\{\theta \in \Omega \, : \, \mathcal{R}(\theta) \leq R\}$. Subsequent works [33, 34, 7] extended convergence analyses of (4) to other classes of loss functions, and [5, 6, 24] proposed alternative forms of mixing of local and neighboring information.

When applied to the minimization of an average loss $f(\theta) = 1/m \sum_{i=1}^m f_i(\theta)$, convergence guarantees of the above DGD-like algorithms, including (4), can be roughly summarized as follows: (i) when $f_i$ are smooth, strongly convex, DGD schemes using constant stepsize converge at linear rate, but only to a neighborhood of the minimizer of the average-loss $f$ [33, 34]. Convergence to the exact minimizer is achieved employing diminishing stepsize rules, at the price of slower sublinear rate [34, 12]. These results are unsatisfactory when applied to the ERM (1). First, for fix $d$ and $N$, they would predict sublinear convergence rate, as the loss $\mathcal{L}$ is convex but not strongly convex (recall $d > N$); this would suggest a negative answer to **(Q1)**–a conclusion that seems to be confuted by recent experiments [28, Fig. 1], showing instead linear convergence of DGD (4) up to a tolerance. Second, when $d$ grows faster than $N$–the typical situation in high-dimension–these optimization studies break down. In fact, they all require *global smoothness* of the loss functions $\mathcal{L}_i$'s and $\mathcal{L}$, a property that no longer holds under the high-dimensional scaling $d/N \to \infty$: for commonly used designs of predictors $x_i$'s, the Lipschitz constant of $\nabla \mathcal{L}$ grows indefinitely with $d/N$ [30].

A statistical study of a DGD-like algorithm solving the LASSO problem (in the Lagrangian form) over mesh networks was recently proposed in [13]: For standard statistical models of predictors, the iterates generated by the algorithm enter an $\varepsilon$-neighborhood of a statistically optimal estimate after

$$\mathcal{O}\left( \frac{\kappa}{1 - \rho} \cdot d \cdot m^2 \log m \cdot \log \frac{1}{\varepsilon} \right) \quad \text{iterations,} \tag{5}$$

where $\rho \in [0, 1)$ is a measure of the connectivity of the network and $\kappa$ is the condition number of the data covariance matrix (population loss). This improves on classical optimization analyses, showing

that this instance of DGD achieves centralized statistical accuracy at a linear rate. However, this rate scales as $\mathcal{O}(d)$. In this paper, we prove that the DGD scheme (4) applied to (1) inherits the same undersirable scaling of the rate–see Appendix H. This would provide a negative answer to **(Q1)**.

A natural question is whether other distributed algorithms offer a more favorable rate scaling, in particular those employing some form of correction of the local gradient direction in the agents' updates. Examples include: primal-dual schemes [27, 11, 25, 26, 14] and gradient tracking methods [21, 17, 32, 8, 29]. However convergence of these algorithms in high-dimension has not been investigated. For fixed $d$ and $N$, with $d > N$, existing analyses predict a pessimistic *sublinear* convergence rate of those methods applied to (1). Furthermore, with the exception of [28], they lack of any statistical analysis and, as those for DGD methods, they break down under high-dimension scaling, $d/N$ growing. The very recent work [28] studied the LASSO problem over networks and proposed a decentralization of PGA based on gradient tracking, termed NetLASSO. Under suitable assumptions, the scheme is proved to reach a neighborhood of a statistically optimal estimate of an $s$-sparse parameter at linear rate. The analysis of NetLASSO is ad-hoc, as it builds on the specific dynamics of the algorithm–hence questions **(Q1)** and **(Q2)** remain open for other distributed algorithms. Furthermore, [28] seems to suggest that gradient-tracking is needed to achieve dimension independent (linear) convergence rates over mesh networks. In this paper we prove that gradient tracking is actually not necessary.

## 1.2 Major contributions

**(i) Algorithm design:** To cope with the 'speed-accuracy dilemma' of DGD in high-dimensions, we propose a new distributed algorithm whereby agents update their local estimate and optimization directions via a 'double-mixing' mechanism, aiming at enforcing consensus on *both* iterates and local gradients. We termed the algorithm $\text{DGD}^2$. At the high-level, our approach brings a new perspective to the design of distributed algorithms in high-dimension: the ultimate goal is not solving the ERM (1) *exactly* but reaching instead an estimate of $\theta^\star$ *within the statistical error* . This can significantly relax the algorithmic design. For instance, contrary to what one might infer from [28], gradient-tracking *is not needed* to efficiently achieve statistically optimal solutions, as our design and analysis prove.

**(ii) Statistical guarantees:** We identify deterministic conditions under which the iterates generated by $\text{DGD}^2$ converge at *linear* rate to a limit point that is within a fixed tolerance from the unknown parameter $\theta^\star$. When customized to statistical models–including sparse linear regression with $\ell_1$ regularization, low-rank matrix recovery with nuclear norm, and matrix completion–our conditions hold with high-probability, and the tolerance becomes of the order of the statistical error $\|\hat{\theta} - \theta^\star\|$. For such models, $\text{DGD}^2$ enters an $\varepsilon$-neighborhood of a *statistically optimal* estimate of $\theta^\star$ after

$$\mathcal{O}\left(\kappa \cdot \log \frac{1}{\varepsilon}\right) \text{ iterations (gradient calls)} \quad \text{and} \quad \mathcal{O}\left(\frac{\kappa}{1 - \rho} \cdot \log(m \cdot \kappa) \cdot \log \frac{1}{\varepsilon}\right) \text{ communications,}$$
(6)

for *any* $\rho \in [0, 1)$. The former matches the rate of PGA; the latter compares favorably with the complexity of DGD in (5), which instead degrades with $d$. Finally, (6) is showed to be invariant under high-dimensional scaling $d/N \to \infty$, as long as $\|\hat{\theta} - \theta^\star\| = \mathcal{O}(1)$. All this addresses **(Q1)** and **(Q2)**. Quite interestingly, it also shows that gradient-tracking as in [28] *is not needed in high-dimensions*.

**(iii) New convergence analysis:** We put forth a new analysis, inspired by the idea of algorithmic stability for stochastic optimization [4, 9]–note that that theory is *not* applicable in high-dimensions and distributed setting. Leveraging the more favorable landscape of the population risk, our approach contrasts the trajectory of $\text{DGD}^2$ to that of a "virtual" $\text{DGD}^2$ instance applied to a population (penalized) variant of the ERM, and establishes conditions for the two trajectories to stay within the desired tolerance while letting $\text{DGD}^2$ inherit fast linear converge of its population counterpart. This contrasts with existing works of distributed algorithms in high-dimension [13, 28], whose convergence is established on the empirical landscape, splitting thus the estimation error $\|\theta_i^t - \theta^\star\|$ between optimization $\|\theta_i^t - \hat{\theta}\|$ and statistical $\|\hat{\theta} - \theta^\star\|$ errors. This makes the analysis very algorithmic-dependent and hard to extend to other solution methods (see remark in Sec. 4.2 for technical details). On the contrary, our approach leads to simpler proofs, potentially applicable to other distributed algorithms than $\text{DGD}^2$ in high-dimensions (e.g., primal-dual). An example is given in Appendix H, where convergence of a variant of DGD in [13] applied to the constrained LASSO problem is provided. Our approach will be applied to other distributed algorithms in future works.

## 2   Setup and Background

We develop our analyses under the following assumptions.

**Assumption 1.** *The population risk $\bar{\mathcal{L}}$ is $\mu$-strongly convex and $L$-smooth on $\mathbb{R}^d$, with $\mu, L \in (0, \infty)$. The condition number is $\kappa \triangleq L/\mu$.*

We consider regularizers $\mathcal{R}$ in ERM (1) that are *decomposable* [2]. Let $\mathcal{M} \subseteq \Omega$ be the so-called *parameter* subspace, capturing constraints from the model (e.g., vectors with a particular support or a subspace of low-rank matrices); and let $\bar{\mathcal{M}}^{\perp}$ be the *perturbation* subspace, representing the deviations from the model subspace $\mathcal{M}$. Decomposability is defined as follows.

**Assumption 2** ([30]). *(i) $\mathcal{R} : \Omega \to \mathbb{R}$ is a norm, with $\Omega \subseteq \mathbb{R}^d$ convex. Furthermore, given a subspace pair $(\mathcal{M}, \bar{\mathcal{M}}^{\perp})$, such that $\mathcal{M} \subseteq \bar{\mathcal{M}}$: (ii) $\mathcal{R}$ is $(\mathcal{M}, \bar{\mathcal{M}}^{\perp})$-decomposable, that is, $\mathcal{R}(x + y) = \mathcal{R}(x) + \mathcal{R}(y)$, for all $x \in \mathcal{M}$ and $y \in \bar{\mathcal{M}}^{\perp}$; and (iii) When $\bar{\mathcal{M}} \neq \{0\}$, $\mathcal{R}$ is $\Psi(\bar{\mathcal{M}})$-Lipschitz over $\bar{\mathcal{M}}$ with respect to some norm $\| \bullet \|$: $\Psi(\bar{\mathcal{M}}) \triangleq \sup_{\theta \in \bar{\mathcal{M}} \setminus \{0\}} \mathcal{R}(\theta)/\|\theta\|$. If $\bar{\mathcal{M}} = \{0\}$, we set $\Psi(\{0\}) = 0$.*

Fast convergence of the PGA relies on the empirical risk $\mathcal{L}$ satisfying suitably restricted notions of strong convexity and smoothness [2]. Under Assumption 1, this can be enforced by controlling the deviation of the Hessian $\nabla^2 \mathcal{L}$ of $\mathcal{L}$ from the Hessian $\nabla^2 \bar{\mathcal{L}}$ of the population loss $\bar{\mathcal{L}}$, along certain directions, as stated below. Note that $\nabla^2 \bar{\mathcal{L}} \succ 0$ is the covariance matrix of the predictors $x_i$'s while $\nabla^2 \mathcal{L} = (1/N) X^T X$, with $X = [x_1, x_2, \ldots, x_N]^{\top}$.

**Assumption 3.** *There exist $\gamma_{g,1}, \gamma_{g,2}, \tau_g > 0$ and a matrix $\Sigma' = \Sigma'^{\top} \succ 0$ such that*

$$\Delta^{\top}\big(\nabla^2 \bar{\mathcal{L}} - \nabla^2 \mathcal{L}\big)\Delta \leq \gamma_{g,1}\|\Delta\|_{\Sigma'}^2 + \tau_g \mathcal{R}^2(\Delta), \ \forall \Delta \in \Omega';$$

$$\Delta^{\top}\big(\nabla^2 \bar{\mathcal{L}} - \nabla^2 \mathcal{L}\big)\Delta \geq -\gamma_{g,2}\|\Delta\|_{\Sigma'}^2 - \tau_g \mathcal{R}^2(\Delta), \ \forall \Delta \in \Omega'.$$

Assumption 3 enforces a curvature property on $\nabla^2 \mathcal{L}$ along the directions $\Delta \in \Omega' \subseteq \mathbb{R}^d$ where $\mathcal{R}^2(\Delta)$ is sufficiently small. In the distributed setting, we need also a local counterpart of Assumption 3, associated with the agents' Hessian matrices $\nabla^2 \mathcal{L}_i = (1/n) X_i^{\top} X_i$, with $X_i = (x_j)_{j \in S_i}$.

**Assumption 4.** *There exist $\gamma_{\ell,1}, \gamma_{\ell,2} \, \tau_{\ell} > 0$ and a matrix $\Sigma' = \Sigma'^{\top} \succ 0$ such that, for all $i$,*

$$\Delta^{\top}\big(\nabla^2 \bar{\mathcal{L}} - \nabla^2 \mathcal{L}_i\big)\Delta \leq \gamma_{\ell,1}\|\Delta\|_{\Sigma'}^2 + \tau_{\ell} \mathcal{R}^2(\Delta), \ \forall \Delta \in \Omega';$$

$$\Delta^{\top}\big(\nabla^2 \bar{\mathcal{L}} - \nabla^2 \mathcal{L}_i\big)\Delta \geq -\gamma_{\ell,2}\|\Delta\|_{\Sigma'}^2 - \tau_{\ell} \mathcal{R}^2(\Delta) \ \forall \Delta \in \Omega'.$$

The practical utility of Assumptions 3 and 4 is that they can be certified with high probability by a variety of random data generations. Here, we consider the following widely used statistical models, which cover a variety of estimation tasks, such as sparse linear models with $\ell_1$ regularization, low-rank matrix recovery with nuclear norm, and matrix regression with soft-rank constraints [15, 2, 30].

**Assumption 5.** *The random predictors $x_i \in \mathbb{R}^d$ are i.i.d. and fulfill one of the following conditions:*

*(i) $x_i \sim \mathcal{N}(0, \Sigma)$ and are i.i.d., for some $\Sigma \succ 0$;*

*(ii) $x_i$ are $(\tau^2, \Sigma)$−sub-Gaussian and i.i.d., with $\Sigma \succ 0$;*

*(iii) $x_i = e_j$ where $j \sim uniform[1, d]$ and i.i.d.*

Under Assumption 5 the curvature conditions in Assumptions 3 and 4 hold with high probability. The following lemma makes this formal under Assumption 5(i), and is a minor modification of [2, Prop. 1, supplementary]. Similar results under Assumptions 5(ii)-(iii) can be found in Appendix F.

**Lemma 1.** *Under Assumption 5(i), there exist universal constants $c_0, c_1, c_2 > 0$ such that, with probability at least $1 - c_0 \exp(-c_1 N + \log m)$ and $N \geq 10$, Assumptions 3 hold with parameters*

$$\Sigma' = \nabla^2 \bar{\mathcal{L}}, \ \gamma_{g,1} = 1/2, \ \gamma_{g,2} = 1, \ \tau_g = c_2 (\mathbb{E}[\mathcal{R}^*(x_i)])^2/N;$$

*and so does Assumption 4 with parameters*

$$\Sigma' = \nabla^2 \bar{\mathcal{L}}, \ \gamma_{\ell,1} = 1, \ \gamma_{\ell,2} = 4m, \ \tau_{\ell} = c_2 (\mathbb{E}[\mathcal{R}^*(x_i)])^2/n,$$

*where $\mathcal{R}^*$ is the dual norm of $\mathcal{R}$. When $\mathcal{R}(\cdot) = \| \cdot \|_1$, $(\mathbb{E}[\mathcal{R}^*(x_i)])^2 \leq 9(\max_j (\Sigma)_{jj}) \log(d)$.*

**Network setup:** Agents are embedded in a communication network, modelled as an undirected graph $\mathcal{G} = \{\mathcal{V}, \mathcal{E}\}$, where the vertices $\mathcal{V} = [m] \triangleq \{1, \ldots, m\}$ correspond to the agents and $\mathcal{E}$ is the set of edges of the graph; $(i,j) \in \mathcal{E}$ if and only if there is a communication link between agents $i$ and $j$. The set of neighbors of each agent $i$ is denoted by $\mathcal{N}_i \triangleq \{j \in \mathcal{V} : (i,j) \in \mathcal{E}\} \cup \{i\}$. We focus on distributed algorithms using gossip weight matrices in their communication steps, as below.

**Assumption 6.** $W = [w_{ij}]_{ij=1}^m$ *satisfies:* **(i)** $w_{ij} > 0$, *if* $(i,j) \in \mathcal{E}$; *otherwise* $w_{ij} = 0$; *furthermore,* $w_{ii} > 0$, *for all* $i \in [m]$; **(ii)** $W = W^\top$ *and* $W1 = 1$ *(stochastic); and* **(iii)** *there holds* $\rho \triangleq \|W - J\|_2 < 1$, *where* $J \triangleq 11^\top / m$.

Assumption 6 is standard in the literature of distributed algorithms and is satisfied by several weight matrices; see, e.g., [16]. Note that $\rho < 1$ holds true by construction for connected graphs. Roughly speaking, $\rho$ measures how fast the network mixes information; the smaller $\rho$, the faster the mixing.

# 3 Algorithm Design: DGD$^2$

We begin providing some rationale on the new design. Our goal is to cope with the speed-accuracy dilemma of DGD (4)–see (5)–while retaining linear convergence to statistically optimal solutions.

It is convenient to rewrite (4) in matrix/vector form. Define the "augmented" quantities:

$$\boldsymbol{\theta}^t \triangleq [(\theta_1^t)^\top, \ldots, (\theta_m^t)^\top]^\top, \quad \boldsymbol{\mathcal{L}}(\boldsymbol{\theta}) \triangleq \frac{1}{m} \sum_{i=1}^m \mathcal{L}_i(\theta_i) \quad \text{and} \quad \tilde{\mathbf{W}} \triangleq \tilde{W} \otimes I_d, \tag{7}$$

with $\otimes$ denoting the Kronecker product. Then, DGD (4) can be rewritten as

$$\boldsymbol{\theta}^{t+1} = \prod_{\Omega \cap \mathcal{R}(\theta) \leq R} \left( \tilde{\mathbf{W}} \boldsymbol{\theta}^t - \alpha \, m \, \nabla \boldsymbol{\mathcal{L}}(\boldsymbol{\theta}^t) \right), \tag{8}$$

where the projector operator is now intended acting block-wise on each $d$-dimensional block. It is well-known that (8) can be interpreted as the iterate dynamics of the projected gradient algorithm (with stepsize $m \alpha$) applied to the following optimization problem [33]: using $\boldsymbol{\theta} \triangleq [(\theta_1)^\top, \ldots, (\theta_m)^\top]^\top$,

$$\min_{\theta_i \in \Omega \, : \, \mathcal{R}(\theta_i) \leq R, \, \forall i \in [m]} \boldsymbol{\mathcal{L}}(\boldsymbol{\theta}) + \frac{1}{2 \, m \, \alpha} \|\boldsymbol{\theta}\|_{I - \tilde{\mathbf{W}}}^2, \tag{9}$$

The quadratic penalty in the objective aims at enforcing consensus among agents' variables $\theta_i$'s, as $\alpha \downarrow 0$. Our analysis (see Appendix H) shows that exact consensus is not needed; in fact, $\alpha = \mathcal{O}(d^{-1})$ is enough to drive the estimation error $(1/m)\|\boldsymbol{\theta}^t - 1 \otimes \theta^*\|^2$ within an $\epsilon$- neighborhood of a *statistically optimal* solution of (2) in a number of communications steps as in (5). Numerical results confirm that $\alpha = \mathcal{O}(d^{-1})$ is necessary for statistical consistency.

This undesirable scaling of the rate (5) with $d$ is a consequence of the restrictive condition $\alpha = \mathcal{O}(d^{-1})$. At a high level our goal is then clear: we need to ease such a constraint. Our idea is to share the consensus-achieving burden in (9) between $\alpha$ (the penalty term) and the first term $\boldsymbol{\mathcal{L}}(\boldsymbol{\theta})$. Notice that at consensus, $\boldsymbol{\theta} = \mathbf{J}\boldsymbol{\theta}$, and thus $\boldsymbol{\mathcal{L}}(\boldsymbol{\theta}) = \boldsymbol{\mathcal{L}}(\mathbf{J}\boldsymbol{\theta})$, where $\mathbf{J} \triangleq J \otimes I_d$. However, $\boldsymbol{\mathcal{L}}(\mathbf{J}\boldsymbol{\theta})$ (and its gradient) is not computable distributively, as agents cannot perform full averages in a single communication step, as subsumed by the operation $\mathbf{J}\boldsymbol{\theta}$. We propose then to approximate $\boldsymbol{\mathcal{L}}(\mathbf{J}\boldsymbol{\theta})$ by $\boldsymbol{\mathcal{L}}(\mathbf{W}\boldsymbol{\theta})$, where $\mathbf{W} \triangleq W \otimes I_d$, for some gossip matrix $W$ (Assumption 6). The gradient of $\boldsymbol{\mathcal{L}}(\mathbf{W}\boldsymbol{\theta})$ is now additively separable across the agents and thus locally computable. We thus replace (9) with

$$\min_{\theta_i \in \Omega \, : \, \mathcal{R}(\theta_i) \leq R, \, \forall i \in [m]} \boldsymbol{\mathcal{L}}(\mathbf{W}\boldsymbol{\theta}) + \frac{1}{2 \, m \, \alpha} \|\boldsymbol{\theta}\|_{I - \tilde{\mathbf{W}}}^2. \tag{10}$$

Our new algorithm is obtained as projected gradient method applied to (10) (with stepsize $m\alpha$):

$$\boldsymbol{\theta}^{t+1} = \prod_{\Omega \cap \mathcal{R}(\theta) \leq R} \left( \tilde{\mathbf{W}} \boldsymbol{\theta}^t - \alpha \, m \, \mathbf{W} \nabla \boldsymbol{\mathcal{L}}(\mathbf{W}\boldsymbol{\theta}^t) \right).$$

This algorithm is fully distributed. However, for arbitrary $\mathbf{W}$ and $\tilde{\mathbf{W}}$, it requires three communication exchanges per iteration. This number can be reduced to two by choosing $\tilde{\mathbf{W}} = \mathbf{W}^2$, yielding

$$\boldsymbol{\theta}^{t+1} = \prod_{\Omega \cap \mathcal{R}(\theta) \leq R} \left( \mathbf{W} \left( \mathbf{W}\boldsymbol{\theta}^t - \alpha \, m \, \nabla \boldsymbol{\mathcal{L}}(\mathbf{W}\boldsymbol{\theta}^t) \right) \right). \tag{11}$$

The agent-level implementation is given in Algorithm 1. Two rounds of mixing (communications) are performed by each agent; hence, we term the algorithm DGD$^2$.

**Algorithm 1 : DGD$^2$**

**Data:** Any feasible $\theta_i^0$, for all $i = 1, \ldots, m$, $\alpha \in (0, 1]$;
**Iterate** $t = 1, 2, \ldots$
**[S.1]:** Each agent $i$ exchanges $\theta_i^t$ with $\mathcal{N}_i$ and updates

$$s_i^{t+1/2} = \sum_{j \in \mathcal{N}_i} w_{ij} \theta_j^t - \alpha \nabla \mathcal{L}_i \Big( \sum_{j \in \mathcal{N}_i} w_{ij} \theta_j^t \Big);$$

**[S.2]:** Each agent $i$ exchanges $s_i^{t+1/2}$ with $\mathcal{N}_i$ and updates:

$$\theta_i^{t+1} = \prod_{\Omega \, \cap \, \mathcal{R}(\theta) \leq R} \left( \sum_{j \in \mathcal{N}_i} w_{ij} s_j^{t+1/2} \right).$$

## 4 Statistical-Computational Guarantees

This section studies convergence of DGD$^2$. Our first result (Theorem 1) establishes conditions under which the estimation error $(1/m) \sum_{i=1}^m \|\theta_i^t - \theta^\star\|_2^2$ shrinks at *linear* rate up to some tolerance $\Delta^2$. In a number of subsequent corollaries, we show that our conditions hold with high probability for the statistical models in Assumption 5, and the tolerance $\Delta^2$ is within minimax statistical rates.

We begin introducing some notation. The tolerance $\Delta^2$ is composed of two terms, $\Delta_{\text{stat}}^2$ and $\Delta_{\text{net}}^2$:

$$\Delta_{\text{stat}}^2 \triangleq \underbrace{\frac{\Psi^2(\bar{\mathcal{M}})}{\mu^2} \mathcal{R}^*(\nabla \mathcal{L}(\theta^\star))^2}_{\text{Stat. error}} + \underbrace{\frac{\mathcal{R}(\Pi_{\mathcal{M}^\perp}(\theta^\star))}{\mu} \mathcal{R}^*(\nabla \mathcal{L}(\theta^\star))}_{\text{Misspecification error}}$$

$$\Delta_{\text{net}}^2 \triangleq \frac{\Psi^2(\bar{\mathcal{M}})}{\mu^2} m^3 \rho^2 \max_{j \in [m]} (\mathcal{R}^*(\nabla \mathcal{L}_j(\theta^\star)))^2 + \frac{\mathcal{R}(\Pi_{\mathcal{M}^\perp}(\theta^\star))}{\mu} m^{3/2} \rho \max_{j \in [m]} \mathcal{R}^*(\nabla \mathcal{L}_j(\theta^\star)),$$

where $\Pi_{\mathcal{M}^\perp}$ (resp. $\Pi_{\bar{\mathcal{M}}}$) denotes the Euclidean projection onto the orthogonal complement of the subspace $\mathcal{M}$ (resp. onto $\bar{\mathcal{M}}$). Notice that $\Delta_{\text{stat}}^2$ matches the tolerance achievable by the PGA in the centralized setting [2] on star networks. The second term in $\Delta_{\text{stat}}^2$ is zero if $\theta^\star \in \mathcal{M}$ (as in sparse vector recovery). On the other hand, $\Delta_{\text{net}}^2$ is a network dependent error, and is the price for decentralization. Observe that the smaller $\rho$, the smaller $\Delta_{\text{net}}^2$. The overall tolerance is defined as

$$\Delta^2 \triangleq \left(1 + \frac{\zeta(\bar{\mathcal{M}}, W)}{2L}\right) (\Delta_{\text{stat}}^2 + \Delta_{\text{net}}^2) + \frac{\mathcal{R}(\Pi_{\mathcal{M}^\perp}(\theta^\star))^2}{\mu \Psi^2(\bar{\mathcal{M}})} \zeta(\bar{\mathcal{M}}, W) \left(1 + \frac{\zeta(\bar{\mathcal{M}}, W)}{2L}\right).$$

where

$$\zeta(\bar{\mathcal{M}}, W) \triangleq \Psi^2(\bar{\mathcal{M}}) \cdot \tau_g + \Psi^2(\bar{\mathcal{M}}) \cdot \tau_\ell \cdot \rho \cdot m^{1.5}. \tag{12}$$

Notice that $\Delta^2$ depends also on the parameters $\tau_g$ and $\tau_\ell$ (see Assumptions 3 and 4) via $\zeta(\bar{\mathcal{M}}, W)$. The first term in (12) accounts for the lack of strong convexity and smoothness in a global sense of the empirical loss–the same dependence is observed in the centralized setting [2]–while the second term, depending on $\rho$, is a consequence of the distributed nature of the estimation.

We are now equipped to state the main convergence result.

**Theorem 1.** *Consider the ERM problem* (1) *and associated population minimization* (2), *under Assumption 1, and regularity conditions in Assumptions 3 and 4 such that:* $R = \mathcal{R}(\theta^\star)$,

$$\gamma_{g,1} \Sigma' + \frac{\mu}{2} I_d \preceq \nabla^2 \bar{\mathcal{L}} \quad and \quad \Psi^2(\bar{\mathcal{M}}) \cdot \tau_g \leq C_0, \tag{13}$$

*for some constant* $C_0 > 0$. *Let* $\{(\theta_i^t)_{i=1}^m\}$ *be the sequence generated by DGD$^2$, with stepsize* $\alpha = (2L)^{-1}$ *and gossip matrix $W$ satisfying Assumption 6 and such that*

$$\rho \leq \text{poly} \left(\kappa m \max(\gamma_{\ell_1}, \gamma_{\ell_2}) \frac{\tau_\ell}{\tau_g}\right)^{-1}. \tag{14}$$

*Then, the estimation error $r^t \triangleq (1/m) \sum_{i=1}^{m} \|\theta_i^t - \theta^\star\|_2^2$ satisfies: for some constant $C_1 > 0$,*

$$r^t \leq \left( \frac{1 - (8\kappa)^{-1} + C_1 \cdot \left(\zeta(\bar{\mathcal{M}}, W)\right)/L}{1 - C_1 \cdot \left(\zeta(\bar{\mathcal{M}}, W)\right)/L} \right)^t r^0 + \mathcal{O}\left(\Delta^2\right). \tag{15}$$

The expression of the constants, a more general statement (Theorem 2), and its proof can be found in Appendix D. A sketch of the proof highlighting our new analysis is presented in Sec.4.2.

*On the rate and tolerance:* Theorem 1 guarantees linear convergence of the estimation error up to a tolerance $\mathcal{O}\left(\Delta^2\right)$. The contraction factor depends on the condition number $\kappa$ of the population loss as well as the network connectivity $\rho$ and the tolerance parameters $\tau_g$ and $\tau_\ell$ via $\zeta(\bar{\mathcal{M}}, W)$–the latter being a consequence of the lack of strong convexity and smoothness of the empirical loss globally. Notice that, for sufficiently small $\zeta(\bar{\mathcal{M}}, W)$, a condition that holds for increasing sample size $N$ and decreasing $\rho$, this rate is of the order of $1 - (8\kappa)^{-1}$, which matches that of the PGA in the centralized setting. Referring to $\Delta^2$, it will be shown in the examples to follow (see Sect. 4.1) that, for a variety of statistical models $\Delta^2$ is of the order of the *centralized* statistical error, $\Delta^2 = O(\|\hat{\theta} - \theta^\star\|^2)$.

*On the conditions (13) and (14):* Conditions in (13) are the same as those required in the centralized setting [2]; they are satisfied by a variety of statistical models (as those considered in Sec. 4.1); in particular, $\Psi^2(\bar{\mathcal{M}}) \cdot \tau_g \leq C_0$ calls for a minimum number of samples $N$ which, for the models in Sec. 4.1, is near (minimax) optimal. Notice that no extra conditions are imposed on the local samples size $n$, which can be as small as one. On the other hand, (14) is a consequence of the decentralization of the M-estimation; it calls for a sufficiently connected network–the larger $m$ or the more ill-conditioned the population loss (larger $\kappa$), the smaller $\rho$. When the graph $\mathcal{G}$ is not part of the design and $W$ is given, (14) can be enforced by employing multiple rounds of communications per gradient evaluations. For instance, given $W$ satisfying Assumption 6, one can build a new matrix, $\bar{W} = W^K$, whose associated $\bar{\rho} = \|\bar{W} - J\| = \rho^K$ satisfies (14); it is sufficient to choose

$$K = \left\lceil (1 - \rho)^{-1} \log \left( \frac{108 \sigma_1(\Sigma') \gamma_\ell m^{1.5}}{\mu} \right) \right\rceil.$$

Algorithm 1 is then implemented using the weights $\bar{W}$, which results in $K$ rounds of communications per iteration $t$, each one using the weights $W$. An $\varepsilon$-estimate of $\theta^\star$, with $\varepsilon \geq \mathcal{O}\left(\Delta^2\right)$, is then achived in $\mathcal{O}\left(\kappa \log 1/\varepsilon\right)$ and $\widetilde{\mathcal{O}}\left(\kappa(1 - \rho)^{-1} \log 1/\varepsilon\right)$ numbers of iterations (gradient evaluations) and communications, respectively, where $\widetilde{\mathcal{O}}$ hides log-factors *independent* on the dimension $d$. If instead $\bar{W}$ is chosen according to Chebyshev polynomials [31], the dependence on $\rho$ in the communication complexity can be improved to $(1 - \rho)^{-1/2}$.

## 4.1   Statistical guarantees

We now develop some consequences of Theorem 1 for specific statistical models. Specifically, Corollary 1 below considers (hard and weak-)sparse linear regression with $\ell_1$ regularization while Corollary 2 deals with a matrix completion problem. In Appendix C, we study sub-Gaussian linear regression (Corollary 3) and matrix regression with nuclear norm constraint (Corollary 4).

*1) Sparse vector regression:* We assume that $\theta^\star$ is sparse. To capture hard- and weak-sparsity we let $\|\theta^\star\|_q \leq R_q$, with $q \in [0, 1]$. Note that $q = 0$ corresponds to vectors with support on a set of cardinality at most $R_0$ (hard-sparsity); the case $q \in (0, 1]$ models approximate sparsity. We consider the following linear generation model over the network: $y_j = x_j^\top \theta^\star + w_j$, $j \in S_i$ and $i \in [m]$, where $w_j \sim \mathcal{N}(0, \sigma^2)$ is the measure noise and $x_j$ satisfies Assumption 5(i). For convenience, we define $\eta_\Sigma \triangleq \max_i([\Sigma]_{ii})$. With this setup we have the following consequence of Theorem 1.

**Corollary 1.** *Consider the linear regression problem above via ERM (1), with $\mathcal{R}(\cdot) = \|\cdot\|_1$, $R = \|\theta^\star\|_1$, and $\Omega \equiv \mathbb{R}^d$. Suppose that (13) holds, which becomes: $\Psi^2(\bar{\mathcal{M}}) \cdot \tau_g = C_1 R_q \eta_\Sigma (\log d/N)^{1-q/2} \leq C_0$, for some constants $C_0, C_1 > 0$, and given $q \in [0, 1]$. Let $\{(\theta_i^t)_{i=1}^m\}$ be the sequence generated by DGD$^2$ under the conditions of Theorem 1, where (14) becomes $\rho \leq C_2(\kappa m^{5/2})^{-1}$, for some constant $C_1 > 0$. Then, with probability at least $1 - c_0 \exp\left(-c_1 N + \log(m)\right)$, for some $c_0, c_1 > 0$ $C_3, C_4 > 0$, the following holds:*

$$r_t^2 \leq \lambda^t r_0^2 + \mathcal{O}\left(\Delta^2\right) + o(\Delta^2),$$

*where*

$$\lambda = \frac{1 - (8\,\kappa)^{-1} + C_3 \cdot (\Psi^2(\bar{\mathcal{M}}) \cdot \tau_g / L)}{1 - C_3 \cdot (\Psi^2(\bar{\mathcal{M}}) \cdot \tau_g / L)} \ \textit{and}$$

$$\Delta^2 = C_4 R_q \left( \frac{\log d}{N} \right)^{1 - \frac{q}{2}} \left( \frac{\eta_\Sigma \sigma^2}{\mu^2} + \frac{\eta_\Sigma^{\frac{1}{2}} \sigma}{\mu} (1 - \delta(q)) \right),$$

*and $\delta(q) = 1$ when $q = 0$, and $0$ otherwise.*

Note that $\Delta_{\text{stat}}^2$ above is (near) minimax optimal for either cases of $q = 0$ and $q \in (0, 1]$ [22]. Furthermore, the rate $\lambda$ is *invariant* under high-dimensional scaling $d/N \to \infty$ and fixed $m$, as long as $\Delta_{\text{stat}}^2 = \mathcal{O}(1)$, a condition necessary for statistical consistency. This matches guarantees of the PGA in the centralized setting [2]. Also, **(i)** it improves on communication complexity (5) of DGD (4) (applicable here when $q = 0$), which scales undesirably with $d$ [13]; and **(ii)** compares favorably with [28], exhibiting same convergence rate order but sightly more favorable scaling of $\rho$ with $m$ ($m^{2.5}$ vs. $m^8$). This shows that gradient-tracking is not needed to match centralized statistical errors.

*2) Matrix Completion:* In this model, the observation $y_j$ is a noisy version of a randomly selected entry $[\Theta^\star]_{a(j),b(j)}$ of the unknown matrix $\Theta^\star \in \mathbb{R}^{p \times p}$. We assume that $\|\Theta^\star\|_q \leq R_q$, with $q \in [0, 1]$, where $\|\Theta^\star\|_q \triangleq \sum_{i=1}^{p} |\sigma_i(\Theta^\star)|^q$, and $\sigma_i(\Theta^\star)$ denotes the $i$-th singular value of $\Theta^\star$. Note that $q = 0$ corresponds to matrices $\Theta^\star$ with rank at most $R_q$ while $q \in (0, 1]$ promotes matrices of all ranks but with relatively fast rate of decay of the singular values. Denoting by $X_j$ the $p \times p$ matrix with $[X_j]_{a(j),b(j)} = 1$ and all the other entries zero, we can write: $y_j = \langle X_j, \Theta^\star \rangle + (\sigma/d) w_j$, $j \in S_i$ and $i \in [m]$, where $\langle \cdot, \cdot \rangle$ is the trace inner product; $w_i$ are noise samples, assumed to be sub-exponential with parameter 1, i.i.d., and independent from $X_i$; $x_i \triangleq \text{vec}(X_i)$ satisfies Assumption 5(iii); $\sigma > 0$ is given, and $d = p^2$. This M-estimation can be written in the ERM form (1), with $\theta = \text{vec}(\Theta) \in \mathbb{R}^d$, with $\Theta \in \mathbb{R}^{p \times p}$ being the unknown matrix, and $x_i = \text{vec}(X_i)$. We choose the set $\Omega = \{\Theta \in \mathbb{R}^{p \times p} : \|\Theta\|_\infty \leq \omega/p\}$ of candidate matrices with bounded element-wise $\ell_\infty$ norm, with $\omega \geq 1$. This eliminates matrices that concentrate too much their mass in a single position. With this setup, we can specialize Theorem 1 as follows.

**Corollary 2.** *Consider the ERM (1) solving the matrix completion problem above, with $\mathcal{R}(\Theta) = \|\Theta\|_1$ and $R = \|\Theta^\star\|_1$. Suppose that (13) holds, which becomes: $\left( p \log p/N \right)^{1 - cq/2} \leq C_0$, for some constant $C_0 > 0$ and $q \in (0, 1]$. Let $\{(\theta_i^t)_{i=1}^m\}$ be the sequence generated by DGD$^2$ under the conditions of Theorem 1, where $\alpha$ reduces to $\alpha = 1$ and (14) becomes $\rho \leq C_1 m^{-5}$, for some constant $C_1 > 0$. Then, with probability at least $1 - c_0(m + 1) \exp(-p \log(p))$, for some $c_0 > 0$, $C_2 > 0$ there holds:*

$$r_t^2 \leq (1 - 1/8)^t r_0^2 + \mathcal{O}\left( \Delta^2 \right), \quad \textit{with} \quad \Delta^2 = C_2 R_q \left( p \log p/N \right)^{1 - \frac{q}{2}} \left( \omega^{2-q} + \sigma^2 \right)$$

Although quantitative aspects of the rates are different, Corollary 2 is analogous to Corollary 1: linear convergence is guaranteed up to a tolerance of the order of (near optimal) statistical minimax rates for the matrix completion problem under the soft-rank model [23]. The more demanding scaling of $\rho$ with $m^{-5.5}$ (vs. $m^{-2.5}$) is due to the slower rate the empirical loss concentrates around the population for such statistical models.

## 4.2 Sketch of proof of Theorem 1

We highlight here the key novelty of our convergence analysis–see Appendix D for the full proof.

At the high level, our proof studies the dynamics of the estimation error $r^t = (1/m) \sum_{i=1}^{m} \|\theta_i^t - \theta^\star\|_2^2$ using as potential function the following *population* instance of (10), which inherits the curvature properties of $\bar{\mathcal{L}}$:

$$\bar{\mathcal{L}}^\alpha(\boldsymbol{\theta}) \triangleq \bar{\mathcal{L}}(\mathbf{W}\boldsymbol{\theta}) + \frac{1}{2\,m\,\alpha} \|\boldsymbol{\theta}\|_{I-\mathbf{W}}^2, \quad \text{with} \quad \bar{\mathcal{L}}(\boldsymbol{\theta}) \triangleq \frac{1}{m} \sum_{i=1}^{m} \bar{\mathcal{L}}(\theta_i). \tag{16}$$

Leveraging *global* strong-convexity and smoothness of $\bar{\mathcal{L}}$, we establish the following decay for $r^{t+1}$ (see Lemma 2 in Appendix E):

$$r^{t+1} \leq \lambda\, r^t + 2\alpha \langle \nabla \mathcal{L}(\mathbf{W}\boldsymbol{\theta}^t) - \nabla \bar{\mathcal{L}}(\mathbf{W}\boldsymbol{\theta}^t), \mathbf{W}(\boldsymbol{\theta}^{t+1} - \boldsymbol{\theta}^\star) \rangle. \tag{17}$$

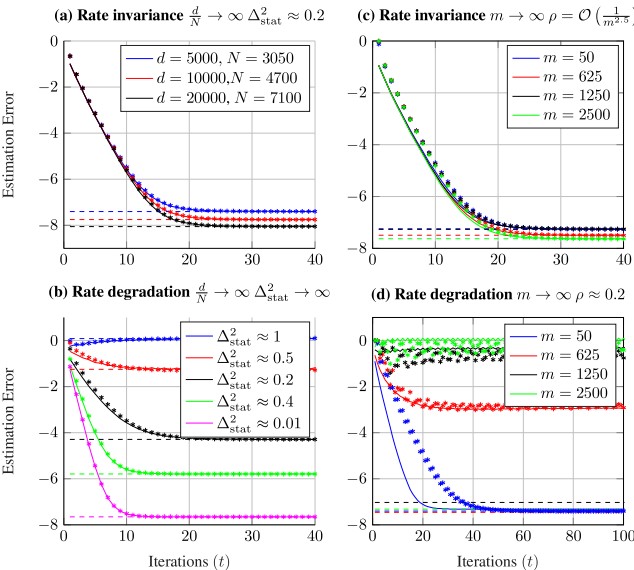

Figure 1: Estimation errors vs. iterations, for DGD$^2$ (solid lines) and NetLASSO (dotted lines); centralized statistical accuracy (horizontal dashed lines). **(a)** Rate invariance under $d/N \uparrow$, $m = 5$, and $\Delta_{\text{stat}}^2 \approx 0.2$; **(b)** Growing $\Delta_{\text{stat}}^2$, with $m = 5$ and $W$ fixed. **(c)** Fixed $d$, and $m \uparrow$, with $\rho \downarrow 0$. **(d)** Same as **(c)** with $\rho$ constant.

for some $\lambda \in (0, 1)$. The first term on the RHS is contractive, due to the strong convexity of $\bar{\mathcal{L}}^\alpha$, while the second term captures the discrepancy between the gradient of the empirical loss and that of the proxy population loss along the direction $\mathbf{W}(\boldsymbol{\theta}^{t+1} - \boldsymbol{\theta}^\star)$. The rest of the proof boils down to carefully control this error term. This is sensible; for instance, the approach used to study algorithmic stability [4, 9] is not satisfactory here: it would upper bound the aforementioned inner product generating in particular the term

$$\|\nabla \mathcal{L}(\mathbf{W}\boldsymbol{\theta}^t) - \nabla \bar{\mathcal{L}}(\mathbf{W}\boldsymbol{\theta}^t)\|^2 = \mathcal{O}(d/n),$$

which grows indefinitely under $d/n \to \infty$, the typical scaling in high-dimensions.

This suggests that the discrepancy of the two gradients needs to be controlled along the direction $\mathbf{W}(\boldsymbol{\theta}^{t+1} - \boldsymbol{\theta}^\star)$. This is the core of our analysis. Under Assumptions 3-4–limiting the deviation of the empirical loss $\mathcal{L}$ from the population loss $\bar{\mathcal{L}}$ along suitable directions–and proper choices of the regularization value $R$ and sufficiently small network connectivity $\rho$, we prove that the inner product in (17) generates an error that preserves contraction (from the first term) while adding an overall tolerance, which can be made of the order of the statistical error.

**Remark:** Our proof differs from existing convergence studies of first-order methods in high-dimensions, both *centralized* [2] and distributed [13, 28]; they all use exclusively the *empirical* loss, decoupling the estimation error $\|\theta_i^t - \theta^\star\|$ in optimization $\|\theta_i^t - \hat{\theta}\|$ and statistical $\|\hat{\theta} - \theta^\star\|$ errors. While $\|\hat{\theta} - \theta^\star\|$ is known for several estimators in the centralized setting [30], this is no longer the case for distributed ERM problems, e.g., as those arising from the design of some DGD-like algorithms [13]–which deal with lifted, penalized formulations–or primal-dual methods, based on saddle-point reformulations. This makes the analyses using the above decomposition problem-dependent and hard to generalize to other formulations (algorithms). Our approach passing throughout the population loss bypasses this issue.

## 5   Numerical Results

We provide some numerical results validating our theoretical findings; more experiments (including on real data) can be found in Appendix B of the supporting material. All simulations are performed on a computer with an Intel i7-8650U CPU @ 1.9 GHz using 16 GB RAM running Windows 10.

We simulate the distributed $s$-sparse linear regression problem, as in Corollary 1. We set $q = 0$, $R_0 = s$, $\Sigma = I_d$, $\sigma^2 = 0.25$, and $R = \|\theta^\star\|_1$. The statistical error for this problem reads $\Delta_{\text{stat}}^2 \triangleq$

$s \log d / N$. The communication network is generated using an Erdős-Rényi graph with $m$ agents and link activation probability $p$. Unspecified values above are given for each particular experiment, as needed. DGD$^2$ is tuned according to the theory, $\alpha = 1/(2L) = 1/2$; we use same stepsize for NetLASSO. All curves are obtained using 10 repetitions via Montecarlo simulation.

**(i) Dependence on $\Delta_{\text{stat}}^2$. Fig. 1(a):** We set $m = 5$, $p = 0.1$ to generate a base graph and $\bar{W}$; we let $W = \bar{W}^T$, with $T = 41$, to achieve a connectivity of $\rho = 0.023 \approx m^{-2.5}$, as required by Corollary 1. We plot the log-normalized estimation error $\log \left( \|\theta^t - \theta^\star\|_2^2 / \|\theta^\star\|_2^2 \right)$ versus the iteration, for different values of $d$ and $N$ as indicated in the legend and $s = \lceil \sqrt{d} \rceil$, so that $d$ grows faster than $N$ while keeping $\Delta_{\text{stat}}^2 \approx 0.2$. Dashed-line curves correspond to the solution to (1) obtained by PGA while solid-and dotted-line curves refer to DGD$^2$ and NetLASSO, respectively. **Fig. 1(b):** We use the same fixed network $W$ ($m = 5$) as in Fig. 1(a) and plot the same quantities, now parametrized on the statistical error $\Delta_{\text{stat}}^2 \in \{1, 0.5, 0.2, 0.4, 0.01\}$. To do so, we fix $d = 20000$, $s = \lceil 2^{-1} \log(d) \rceil$ and vary $N$ accordingly, i.e., $N \in \{50, 100, 250, 1250, 5000\}$. **Discussion:** Fig. 1(a)-(b) validates (14): under $s$, $d/N$ growing, the convergence rate of DGD$^2$ is linear and remains invariant, as long as $\Delta_{\text{stat}}^2$ stays constant (Fig. 1.(a)) while algorithm's performance degrades when $\Delta_{\text{stat}}^2$ increases (Fig. 1(b)). Note that DGD$^2$ matches the rate of the centralized PGA and performs as good as NetLASSO without employing any gradient tracking.

**(ii) Dependence on $m$ and $\rho$. Fig. 1(c):** We plot the same quantities as in Fig. 1(a), now parametrized on $m$, for fixed $d = 5000$, $s = \lceil \sqrt{d} \rceil$, and $N = 2500 = m \cdot n$ (this varying $n$). We choose $m \in \{50, 625, 1250, 2500\}$ and generate the underlying graph such that $\rho$ varies to satisfy the required condition $\rho \lesssim m^{-2.5}$. For this, we set $p = 0.8$, resulting in a base graph with associated gossip $\bar{W}_m$; we then build $W_m = \bar{W}_m^T$, $T = 7$, and use such $W_m$ in all the algorithms. **Fig. 1(d):** We use the same setup as in Fig. 1(c) with the difference that the underlying graphs are generated so that $\rho \approx 0.2$, for all $m$. To achieve this we set the link activation probabilities to $p \in \{0.87, 0.4, 0.22, 0.15\}$ for the associated values of $m$. **Discussion:** Fig. 1(c)-(d) demonstrates the necessity of a decreasing network connectivity $\rho$ for increasing $m$ and fixed $N$, as predicted by out theory. In fact, under $\rho \lesssim m^{-2.5}$, the convergence rate and statistical error remain roughly invariant, as $m$ grows (Fig. 1(c)). On the other hand, all the algorithms break down if $\rho$ does not decrease with $m$ (Fig. 1(d)).

# 6 Concluding Remarks

We proposed DGD$^2$, a decentralized scheme solving high-dimensional quadratic M-estimation problems over mesh networks, which employs a double-mixing procedure averaging suitably iterates and local gradients. DGD$^2$ converges linearly to statistically optimal solutions, without requiring any condition on the local sample size (but centralized statistical consistency); differently from DGD-like schemes, the rate is independent on the ambient dimension (under standard assumptions on statistical consistency of the estimators), Quite interestingly our results show that, despite common wisdom, centralized statistical consistency can be achieved over networks without employing any explicit gradient tracking mechanism but instead just mixing local gradients at sufficiently fast rate. The analysis of other distributed algorithms for high-dimensional M-estimation problem remains a task of future investigations.

## Acknowledgments and Disclosure of Funding

Funding in direct support of this work: ONR Grant N. N00014-21-1-2673.

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
