# Appendix

This appendix contains additional theoretical and numerical results as well as the proofs of the results presented in the paper. The appendix is organized as follows.

**Sec. A** summarizes the notation used in the paper;

**Sec. B** contains additional numerical experiments;

**Sec. C** contains some consequences of Theorem 1 for (i) the $\ell_1$ constrained sub-Gaussian regression problem under Assumption 5(ii) (Corollary 3); and (ii) and the nuclear norm constrained matrix regression problem under Assumption 5(ii) (Corollary 4). The proof of these corollaries along with Corollary 1 and Corollary 2 in the main text can be found in **Sec. G**;

**Sec. D** contains the main theorem of the paper (Theorem 2) along with its proof, which is a more general version of Theorem 1;

**Sec. E** contains the proof of some intermediate results stated in Sec. D and instrumental to prove Theorem 2;

**Sec. F** summarizes some concentration bounds regarding the discrepancy between empirical and population Hessian matrices, for the data models considered in Assumption 5, which are instrumental to show that Assumptions 3 and 4 hold with high probability; and

**Sec. H** studies statistical and computational guarantees of the DGD algorithm (4), applied to the $\ell_1$-constrained sparse regression problem. This validates (5) (with a slightly better scaling with $m$) and complements results in [13] obtained for the DGD algorithm applied to a Lagrangian formulation of the LASSO problem over networks.

## A   Summary of the Main Notaton Adopted in the Paper

Problem size:

| Symbols | Location | Description |
|---|---|---|
| $d$ | Section 1 | Problem dimension |
| $n$ | (1) | Number of local samples |
| $m$ | (1) | Number of agents |
| $N = n \cdot m$ | Section 1 | Total number of samples |

Population and Empirical risk variants:

| Symbols | Location | Description |
|---|---|---|
| $\mathcal{L}_i$ | (1) | Local Empirical Loss |
| $\mathcal{L}$ | (1) | Global Empirical Loss |
| $\bar{\mathcal{L}}$ | (2) | Population Loss |
| $\mathcal{L}(\boldsymbol{\theta})$ | (7) | Stacked empirical risk |
| $\bar{\mathcal{L}}(\boldsymbol{\theta})$ | (16) | Staked population risk |
| $\bar{\mathcal{L}}^\alpha(\boldsymbol{\theta})$ | (16) | Augmented stacked population risk |

Population curvature parameters:

| Symbols | Location | Description |
|---|---|---|
| $\kappa$ | (5), Assumption 1 | Condition number |
| $L, \mu$ | Assumption 1 | Lipschitz and strong convexity constants |

Network quantities:

| Symbols | Location | Description |
|---|---|---|
| $\rho$ | (5), Assumption 6 | Network connectivity |
| $W, \tilde{W}$ | Assumption 6 | Gossip matrices |
| $\mathbf{W}, \tilde{\mathbf{W}}$ | Section 3 | Stacked Gossip matrices |
| $J$ | Section 3 | $J = \frac{1}{m}\mathbf{1}\mathbf{1}_m^\top$ |
| $\mathbf{J}$ | Section 3 | Staked fully connected network |

Regularization quantities:

| Symbols | Location | Description |
|---|---|---|
| $\mathcal{R}$ | (1) | Norm constraint |
| $R$ | (1) | Constraint radius |
| $\mathcal{M}$ | Assumption 2 | Model subspace |
| $\bar{\mathcal{M}}^\perp$ | Assumption 2 | Perturbation subspace |
| $\Psi^2(\bar{\mathcal{M}})$ | Assumption 2 | $\mathcal{R} - \ell_2$ Lipschitz constant on $\mathcal{M}$ |
| $\mathcal{R}^*$ | Section 4 | Dual norm to $\mathcal{R}$ |

Tolerances:

| Symbols | Location | Description |
|---|---|---|
| $\gamma_{g,1},\ \gamma_{g,2},\ \gamma_{\ell,1},\ \gamma_{\ell,2}, \tau_g \Sigma'$ | Assumption 3 | Tolerances in global Hessian deviation |
| $\gamma_{\ell,1},\ \gamma_{\ell,2}, \tau_\ell$ | Assumption 4 | Tolerances in local Hessian deviations |

Miscellanea:

| Symbols | Location | Description |
|---|---|---|
| $S_i$ | (1) | Indexes of samples corresponding to agent $i$ |
| $\alpha$ | (4) | Step size |

For any given matrix $A \in \mathbb{R}^{p \times p}$ we denote by

$$\|A\|_1 \triangleq \sum_{i,j=1}^{p} |a_{i,j}|, \qquad\qquad \|\|A\|\|_1 \triangleq \max_{1 \le j \le p} \sum_{i=1}^{p} |a_{i,j}|,$$

$$\|A\|_F \triangleq \left( \sum_{i,j=1}^{p} |a_{i,j}|^2 \right)^{1/2}, \qquad\qquad \|\|A\|\|_2 \triangleq \left( \lambda_{\max}(A^\top A) \right)^{1/2},$$

$$\|A\|_\infty \triangleq \max_{1 \le i,j \le p} |a_{i,j}|, \qquad\qquad \|\|A\|\|_\infty \triangleq \max_{1 \le j \le p} \left| \sum_{j=1}^{p} a_{i,j} \right|.$$

## B   Additional Numerical Experiments

This section contains additional numerical experiments on read and synthetic data.

*1) Simulations on real data:* We test DGD$^2$ on the data set `eyedata` in the `NormalBetaPrime` package [3]; $d = 200$ and $N = 120$. We generate a base graph with $m = 10$ and $p = 0.9$, yielding a gossip matrix $\bar{W}$. To achieve $\rho \approx m^{-2.5}$ we build $W = \bar{W}^7$. We split the data set into $N_{\text{test}} = 40$ and $N_{\text{train}} = 80$, and further split the train data evenly across 10 agents. Fig 1(e) plots the logarithm of the objective function value along the iterates generated by PGA (as benchmark), DGD$^2$ and NetLASSO. Observe that the distributed schemes converge linearly at the same rate as the centralized scheme. Fig 1(f) evaluates the prediction errors on the test set, as follows. Denote by $y$ the output of the test set and by $X$ the test covariates. Then, we build the predictors $y_i^t = X\theta_i^t$ $i \in [m]$, where $\theta_i^t$ is the estimate of $\theta^\star$ at agent $i$'s side at iteration $t$ based on the training set.

**(a) Real data: Train average empirical loss**    **(b) Real data: Test average prediction error**

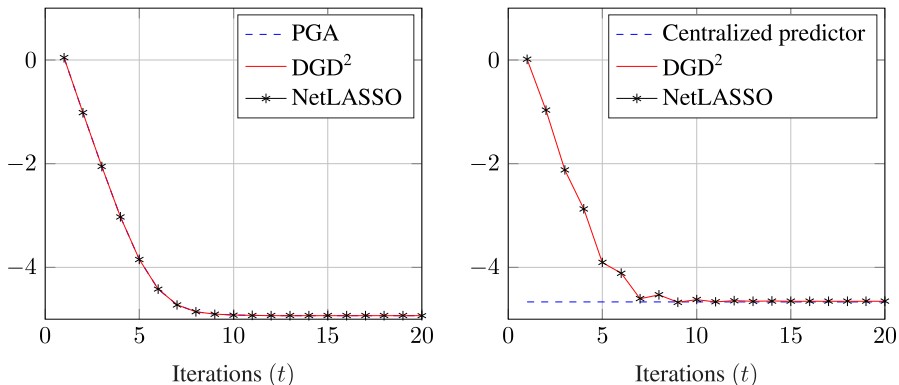

Figure 2: PGA, DGD$^2$ and NetLASSO on real data. **(a):** Rates (logarithm of objective function) vs. iterations; **(b):** Average prediction error on the test set (compared to centralized predictor on the test set).

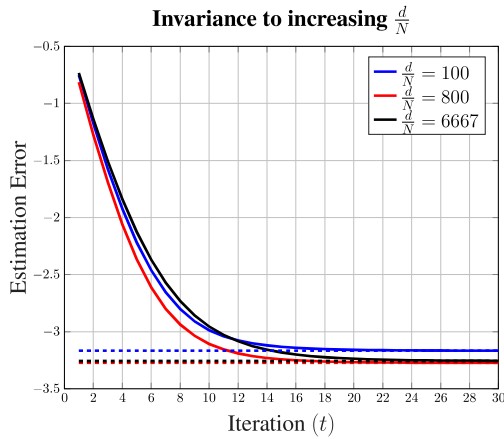

Figure 3: Estimation error generated by DGD$^2$ (solid line) vs. iterations, for different values of $d/N$ and $s \log d/N = 0.0921$, $m = 5$; horizontal dashed line curves mark centralized statistical accuracy.

Fig 1(f) plots $\log \left( (mN_{\text{test}})^{-1} \sum_{i=1}^{m} \|y - y_i^t\|_2^2 \right)$ versus the iterations, for DGD$^2$ and NetLASSO. The dashed line-curve corresponds to $\log((N_{\text{test}} m)^{-1} \|y - \hat{y}\|_2^2)$ with $\hat{y} = X\hat{\theta}$, where $\hat{\theta}$ denotes the estimator obtained via PGA. As on synthetic data, both distributed schemes converge linearly up to the (estimated) statistical error (as our theory predicts), which matches centralized guarantees.

*2) Synthetic data:* We provide some additional numerical results (on synthetic data) validating the rate invariance under the asymptotic scaling $s, d, N$ growing, for fixed $m$ and $s \log d/N$. Specifically, we simulate the distributed $s-$sparse linear regression problem as described in Section 5; we set $q = 0$, $R_0 = s = 1$, $\Sigma = I_d$, $\sigma_d^2 = 0.25$ and $R = \|\theta^\star\|_1$. We use the same communication network and number of communications as in Fig. 1(a), and set the step size $\alpha = \frac{1}{2}$. We vary $d = \{10^4, 10^5, 10^6\}$ and $n = \{20, 25, 30\}$, so that $\frac{s \log d}{N} \approx 0.0921$ for the three cases. We observe that, as predicted by the theory, the performance (convergence rate and achieved statistical error) of DGD$^2$ remain invariant as $d/N$ grows.

## C    Additional Statistical Models

In this section we apply Theorem 1 to the $\ell_1$-constrained regression problem (Sec. C.1) and the nuclear norm constrained matrix regression (Sec. C.2), under the data model satisfying Assumption 5(ii) and (i), respectively.

## C.1 Sub-Gaussian linear regression

Let $\theta^\star$ be (either hard-or weak-)sparse, which we capture by assuming $\|\theta^\star\|_q \leq R_q$, with $q \in [0,1]$. Consider the network data generation model

$$y_j = x_j^\top \theta^\star + w_j, \quad j \in S_i \quad \text{and} \quad i \in [m],$$

where $w_j$ is assumed to be $w_j \in \mathcal{N}(0, \sigma^2)$. Denote

$$X_{S_i} \triangleq [x_{j_1}^\top, \ldots, x_{j_m}^\top]^\top, \quad j_l \in S_i, \quad l \in [n], \quad \text{and} \quad i \in [m],$$

where $X_{S_i}$ is assumed to be $C$-column normalized.

**Corollary 3** ($\ell_1$-constrained sub-Gaussian regression). *Consider the ERM* (1) *solving the linear regression problem above, with* $\mathcal{R}(\cdot) = \|\cdot\|_1$, *and* $\Omega \equiv \mathbb{R}^d$. *Suppose that*

$$\Psi^2(\bar{\mathcal{M}}) \cdot \tau_g = C_1 R_q \max\left\{\frac{\tau^4}{\mu}, \mu\right\} \left(\frac{\log d}{N}\right)^{1-\frac{q}{2}} \leq C_0, \quad q \in [0,1],$$

*for some constants* $C_0, C_1 > 0$. *Let* $\{(\theta_i^t)_{i=1}^m\}$ *be the sequence generated by DGD$^2$ under the conditions of Theorem 2, where* (14) *becomes*

$$\rho \leq \frac{C_2}{m^{5/2}},$$

*for some constant* $C_2 > 0$. *Then, with probability at least*

$$1 - 2\left(-c_0 N \min\left\{\frac{\mu^2}{\tau^4}, 1\right\} + \log(m+1)\right) - 2d\exp(-2\log(d)) - 2md\exp(-2\log(md)),$$

*for some constant* $c_0 > 0$, *there holds*

$$r_t^2 \leq \left(\frac{1 - \frac{1}{8\kappa} - C_3 \frac{\Psi^2(\bar{\mathcal{M}}) \cdot \tau_g}{L}}{1 - C_3 \frac{\Psi^2(\bar{\mathcal{M}}) \cdot \tau_g}{L}}\right)^t r_0^2 + \mathcal{O}\left(\Delta^2\right) + o(\Delta^2),$$

*where* $C_3$, $C_4 > 0$ *and*

$$\Delta^2 = C_4 R_q \left(\frac{\log d}{N}\right)^{1-\frac{q}{2}} \left(\frac{C\sigma^2}{\mu^2} + \frac{C^{1/2}\sigma}{\mu}(1 - \delta(q))\right) \quad q \in [0,1].$$

*Proof.* See Appendix G. $\qquad\square$

## C.2 Gaussian Matrix Regression

Consider the observation model

$$y_j = \langle X_j, \Theta^\star\rangle + w_j, \quad j \in S_i \quad \text{and} \quad i \in [m],$$

where $X_j \in \mathbb{R}^{p\times p}$ is such that $\text{vec}(X_j)$ satistifes Assumption 5(i), and $w_j \sim \mathcal{N}(0, \sigma^2)$ are i.i.d. and independent of $X_j$,. The unknown matrix $\Theta^\star \in \mathbb{R}^{p\times p}$ satisfies $\|\Theta^\star\|_q \leq R_q$, with $q \in [0,1]$, and $R_q > 0$.

**Corollary 4** (Nuclear norm constrained Gaussian Matrix Regression). *Consider the ERM* (1) *solving the matrix regression problem above with* $\Omega \equiv \mathbb{R}^{p\times p}$, $\mathcal{R}(\Theta) = \|\Theta\|_1$, *and* $R = \|\Theta^\star\|_1$. *Suppose that*

$$\Psi^2(\bar{\mathcal{M}}) \cdot \tau_g = C_1 R_q \eta_\Sigma \sigma^2 \left(\frac{p}{N}\right)^{1-\frac{q}{2}} \leq C_0, \quad q \in [0,1],$$

*for some* $C_0$, $C_1 > 0$. *Let* $\{(\theta_i^t)_{i=1}^m\}_{t\geq 0}$ *be the sequence generated by DGD$^2$ under the conditions of Theorem 1, where* (14) *reduces to*

$$\rho \leq C_2 \frac{\kappa^{-1}}{m^2\sqrt{m}},$$

*for some constant $C_2 > 0$. Then, with probability at least*

$$1 - c_0 m \exp\left(-\frac{N}{2}\right) - \exp(-c_1 N) - (m+1)\exp(-c_2 p),$$

*for some $c_0$, $c_1$, $c_2 > 0$, there holds*

$$r_t^2 \leq \left(\frac{1 - \frac{\kappa^{-1}}{8} + C_3 \frac{\Psi^2(\bar{\mathcal{M}}) \cdot \tau_g}{L}}{1 - C_3 \frac{\Psi^2(\bar{\mathcal{M}}) \cdot \tau_g}{L}}\right) + \mathcal{O}(\Delta^2) + o(\Delta^2),$$

*where $C_3$, $C_4 > 0$ and*

$$\Delta^2 = C_4 R_q \left(\frac{p}{M}\right)^{1-\frac{q}{2}} \left(\frac{\sigma^2 \eta_\Sigma}{\mu^2} + \frac{\sigma \eta_\Sigma^{1/2}}{\mu}(1 - \delta(q))\right), \quad q \in [0,1],$$

*where $\delta(q) = 1$ for $q = 0$ and $0$ otherwise.*

*Proof.* See Appendix G. □

Corollaries 3 and 4 establish for the statistical models above the same type of statistical computational guarantees as Corollary 1 and Corollary 2.

## D   Theorem 1 and some Generalizations

In this section we introduce and prove a more general version of Theorem 1, which will follow as a special case. We begin introducing some notation. Denote by

$$\Delta_{\text{stat}}^2 \triangleq \frac{64\Psi^2(\bar{\mathcal{M}})}{\mu^2} \left(\mathcal{R}^* \left(\nabla\mathcal{L}(\theta^\star)\right)\right)^2 + \frac{24}{\mu}\mathcal{R}\left(\Pi_{\mathcal{M}^\perp}(\theta^\star)\right)\mathcal{R}^*\left(\nabla\mathcal{L}(\theta^\star)\right),$$

$$\Delta_{\text{net}}^2 \triangleq \frac{64\Psi^2(\bar{\mathcal{M}})}{m\mu} 3 \sum_{i=1}^m \left(\sum_{j=1}^m (w_{i,j} - 1/m)\nabla\mathcal{L}_j(\theta^\star)\right)^2$$

$$+ \frac{24}{m\mu}\mathcal{R}\left(\Pi_{\mathcal{M}^\perp}(\theta^\star)\right)\sum_{i=1}^m \mathcal{R}^*\left(\sum_{j=1}^m (w_{i,j} - 1/m)\nabla\mathcal{L}_j(\theta^\star)\right),$$

and the overall tolerance

$$\Delta^2 \triangleq (\Delta_{\text{stat}}^2 + \Delta_{\text{net}}^2)\left(1 + \frac{\alpha\zeta(\bar{\mathcal{M}}, W)}{1 - \alpha\zeta(\bar{\mathcal{M}}, W)}\right)$$

$$+ \frac{144}{\mu\Psi^2(\bar{\mathcal{M}})}\mathcal{R}^2\left(\Pi_{\mathcal{M}^\perp}(\theta^\star)\right)\zeta(\bar{\mathcal{M}}, W)\left(1 + \frac{\alpha\zeta(c\bar{M}, W)}{1 - \alpha\,\zeta(\bar{\mathcal{M}}, W)}\right),$$

where we recall that $\alpha$ is the stepsize and

$$\zeta(\bar{\mathcal{M}}, W) \triangleq 24\tau_\ell\Psi^2(\bar{\mathcal{M}})(\rho^2 m^3 + 2\rho m\sqrt{m}) + 24\tau_g\Psi^2(\bar{\mathcal{M}}).$$

Finally, for any matrix $V \in \mathbb{R}^{k \times k}$, we denote by $\sigma_i(V)$, $i \in [k]$, its $i$-th largest singular value.

We are now ready to provide our main result.

**Theorem 2.** *Consider the ERM problem* (1) *and the associated population minimization* (2) *under Assumptions 1-4, with parameters therein such that the following hold: $R = \mathcal{R}(\theta^\star)$,*

$$\gamma_{g,1}\Sigma' + \frac{\mu}{2}I_d \preceq \nabla^2\mathcal{L} \quad and \quad \zeta(\bar{\mathcal{M}}, W) \leq \frac{5\mu}{36}.$$

*Let $\{\theta^t\}_{t\geq 0}$ be the sequence generated by $DGD^2$ with stepsize*

$$\alpha \leq \frac{1}{L + \gamma_{g,2}\sigma_1(\Sigma')},$$

*and gossip matrix $W$ satisfying Assumption 6 and such that*

$$\rho \leq \frac{\mu}{108 m^{1.5} \sigma_1(\Sigma') \max\{\gamma_{\ell,1}, \gamma_{\ell,2}\}}.$$

*Then, the estimation error $r^t \triangleq \frac{1}{m} \sum_{i=1}^m \|\theta_i^t - \theta^\star\|_2^2$ satisfies*

$$r_t^2 \leq \left( \frac{1 - \frac{\alpha\mu}{2} + \alpha\sigma_1(\Sigma')3\gamma_\ell(2m\sqrt{m}\rho + \rho^2 m^3) + \alpha\zeta(\bar{\mathcal{M}}, W)}{1 - \alpha\zeta(\bar{\mathcal{M}}, W)} \right)^t r_0^2 + \Delta^2.$$

### D.1 Proof of Theorem 2

The key idea in proving the theorem is characterizing the dynamics of the estimation error based on the favorable landscape of the population risk while carefully controlling the error due to the mismatch between the empirical and population gradients.

We begin introducing some notation. Recall that DGD$^2$ can be interpreted as the gradient descent applied to the proxy problem

$$\min_{\theta_i \in \mathbb{B}_\mathcal{R}(\rho) \cap \Omega} \quad \mathcal{L}^\alpha(\boldsymbol{\theta}) \triangleq \mathcal{L}(\mathbf{W}\boldsymbol{\theta}) + \frac{1}{2\alpha m} \|\boldsymbol{\theta}\|_{I - \mathbf{W}^2}^2. \tag{18}$$

The population counterpart of (18) is based on the following population augmented loss:

$$\bar{\mathcal{L}}^\alpha(\boldsymbol{\theta}) \triangleq \bar{\mathcal{L}}(\mathbf{W}\boldsymbol{\theta}) + \frac{1}{2\alpha m} \|\boldsymbol{\theta}\|_{I - \mathbf{W}^2}^2, \tag{19}$$

where

$$\bar{\mathcal{L}}(\boldsymbol{\theta}) \triangleq \frac{1}{m} \sum_{i=1}^m \bar{\mathcal{L}}(\theta_i).$$

Consequently,

$$\nabla\bar{\mathcal{L}}(\boldsymbol{\theta}) = \frac{1}{m}[\nabla\bar{\mathcal{L}}(\theta_1)^\top, \ldots, \nabla\bar{\mathcal{L}}(\theta_m)^\top]^\top \quad \text{and} \quad \nabla^2\bar{\mathcal{L}} \triangleq \frac{1}{m}\nabla^2\bar{L} \otimes I_m = \frac{1}{m}\Sigma \otimes I_m.$$

Notice that, by construction, $\bar{\mathcal{L}}^\alpha(\boldsymbol{\theta})$ inherits much of the properties of the population risk $\bar{\mathcal{L}}$; in particular,

$$\bar{\mathcal{L}}^\alpha(\boldsymbol{\theta}') - \bar{\mathcal{L}}^\alpha(\boldsymbol{\theta}) - \langle\nabla\bar{\mathcal{L}}^\alpha(\boldsymbol{\theta}), \boldsymbol{\theta}' - \boldsymbol{\theta}\rangle = \frac{1}{2}\|\mathbf{W}(\boldsymbol{\theta} - \boldsymbol{\theta}')\|_{\nabla^2\bar{\mathcal{L}}}^2 + \frac{1}{2\alpha m}\|\boldsymbol{\theta} - \boldsymbol{\theta}'\|_{I - \mathbf{W}^2}^2, \tag{20}$$

for all $\boldsymbol{\theta}, \boldsymbol{\theta}' \in \mathbb{R}^{md}$. Also, $\boldsymbol{\theta}^\star$ is the unique minimizer of $\bar{\mathcal{L}}^\alpha$ as well.

Our analysis begins exploiting the positive curvature of $\bar{\mathcal{L}}$ in (20) to establish a decrease of the estimation error along the trajectory $\{\boldsymbol{\theta}^t\}$ of DGD$^2$ (11), as stated next.

**Lemma 2.** *Under Assumptions 1 and 6, the iterates $\{\boldsymbol{\theta}^t\}_{t\geq 0}$ generated by DGD$^2$ (11) for any $\alpha > 0$, satisfy*

$$\begin{aligned}
\|\boldsymbol{\theta}^{t+1} - \boldsymbol{\theta}^\star\|_2^2 \leq &\|\mathbf{W}(\boldsymbol{\theta}^t - \boldsymbol{\theta}^\star)\|_{\mathbf{I} - \alpha m\nabla^2\bar{\mathcal{L}}}^2 - \alpha m\|\mathbf{W}(\boldsymbol{\theta}^{t+1} - \boldsymbol{\theta}^\star)\|_{\nabla^2\bar{\mathcal{L}}}^2 \\
&- \|\mathbf{W}(\boldsymbol{\theta}^{t+1} - \boldsymbol{\theta}^t)\|_{I - \alpha m\nabla^2\bar{\mathcal{L}}}^2 - \|\boldsymbol{\theta}^{t+1}\|_{\mathbf{I} - \mathbf{W}^2}^2 \\
&+ 2\alpha m\langle\mathbf{W}\left(\nabla\bar{\mathcal{L}}(\mathbf{W}\boldsymbol{\theta}^t) - \nabla\mathcal{L}(\mathbf{W}\boldsymbol{\theta}^t)\right), \boldsymbol{\theta}^{t+1} - \boldsymbol{\theta}^\star\rangle.
\end{aligned} \tag{21}$$

*Proof.* See Appendix E. $\qquad\qquad\square$

Notice that the first term on the RHS of (21) captures the contraction properties of the gradient algorithm applied to the population loss while the inner product accounts for the trajectory mismatch between the empirical and population-based updates along sparse directions. The rest of the proof consists in suitably bounding this inner-product term.

Rewrite (21) as

$$\|\boldsymbol{\theta}^{t+1} - \boldsymbol{\theta}^\star\|_2^2 \le \|\boldsymbol{\theta}^t - \boldsymbol{\theta}^\star\|_2^2 - \alpha m \left( \|\mathbf{W}(\boldsymbol{\theta}^t - \boldsymbol{\theta}^\star)\|_{\nabla^2 \bar{\mathcal{L}}}^2 + \|\mathbf{W}(\boldsymbol{\theta}^{t+1} - \boldsymbol{\theta}^\star)\|_{\nabla^2 \bar{\mathcal{L}}}^2 \right)$$
$$- \left( \|\boldsymbol{\theta}^t\|_{I-\mathbf{W}^2}^2 + \|\boldsymbol{\theta}^{t+1}\|_{I-\mathbf{W}^2}^2 \right) + \alpha m \|\mathbf{W}(\boldsymbol{\theta}^{t+1} - \boldsymbol{\theta}^t)\|_{\nabla^2 \bar{\mathcal{L}}}^2 - \|\boldsymbol{\theta}^t - \boldsymbol{\theta}^{t+1}\|_{\mathbf{W}^2}^2$$
$$+ 2\alpha m \langle \mathbf{W}(\nabla \bar{\mathcal{L}}(\mathbf{W}\boldsymbol{\theta}^t) - \nabla \mathcal{L}(\mathbf{W}\boldsymbol{\theta}^t)), \boldsymbol{\theta}^{t+1} - \boldsymbol{\theta}^t \rangle.$$

Using $\nabla^2 \bar{\mathcal{L}} \preceq \frac{L}{m} I_{md}$ we have

$$\|\boldsymbol{\theta}^{t+1} - \boldsymbol{\theta}^\star\|_2^2 \le \|\mathbf{W}\boldsymbol{\theta}^t - \boldsymbol{\theta}^\star\|_2^2 - \alpha m \left( \|\mathbf{W}(\boldsymbol{\theta}^t - \boldsymbol{\theta}^\star)\|_{\nabla^2 \bar{\mathcal{L}}}^2 + \|\mathbf{W}(\boldsymbol{\theta}^{t+1} - \boldsymbol{\theta}^\star)\|_{\nabla^2 \bar{\mathcal{L}}}^2 \right)$$
$$- \|\boldsymbol{\theta}^{t+1}\|_{I-\mathbf{W}^2}^2 + (\alpha L - 1) \|\mathbf{W}(\boldsymbol{\theta}^{t+1} - \boldsymbol{\theta}^t)\|_2^2 \tag{22}$$
$$+ 2\alpha m \langle \mathbf{W}(\nabla \bar{\mathcal{L}}(\mathbf{W}\boldsymbol{\theta}^t) - \nabla \mathcal{L}(\mathbf{W}\boldsymbol{\theta}^t)), \boldsymbol{\theta}^{t+1} - \boldsymbol{\theta}^\star \rangle. \tag{23}$$

When $\alpha \le 1/L$, the only detrimental term on the RHS of the above inequality is the inner product; we aim at controlling this term using Assumptions 3 and 4. To do so, we add and subtract therein

$$2\alpha m \langle \nabla \bar{\mathcal{L}}(\boldsymbol{\theta}^\star) - \nabla \mathcal{L}(\boldsymbol{\theta}^\star), \mathbf{W}(\boldsymbol{\theta}^{t+1} - \boldsymbol{\theta}^\star) \rangle,$$

so that

$$2\alpha m \langle \mathbf{W}(\nabla \bar{\mathcal{L}}(\mathbf{W}\boldsymbol{\theta}^t) - \nabla \mathcal{L}(\mathbf{W}\boldsymbol{\theta}^t) - \nabla \bar{\mathcal{L}}(\boldsymbol{\theta}^\star) + \nabla \mathcal{L}(\boldsymbol{\theta}^\star)), \boldsymbol{\theta}^{t+1} - \boldsymbol{\theta}^\star \rangle =$$
$$2\alpha m \langle (\nabla^2 \bar{\mathcal{L}} - \nabla^2 \mathcal{L})(\mathbf{W}\boldsymbol{\theta}^t - \boldsymbol{\theta}^\star), \mathbf{W}(\boldsymbol{\theta}^{t+1} - \boldsymbol{\theta}^\star) \rangle,$$

and consequently

$$2\alpha m \langle \mathbf{W}(\nabla \bar{\mathcal{L}}(\mathbf{W}\boldsymbol{\theta}^t) - \nabla \mathcal{L}(\mathbf{W}\boldsymbol{\theta}^t)), \boldsymbol{\theta}^{t+1} - \boldsymbol{\theta}^\star \rangle$$
$$= 2\alpha m \langle (\nabla^2 \bar{\mathcal{L}} - \nabla^2 \mathcal{L})(\mathbf{J}\boldsymbol{\theta}^t - \boldsymbol{\theta}^\star), \mathbf{J}\boldsymbol{\theta}^{t+1} - \boldsymbol{\theta}^\star \rangle \tag{24}$$
$$+ 2\alpha m \langle (\nabla^2 \bar{\mathcal{L}} - \nabla^2 \mathcal{L})(\mathbf{W} - \mathbf{J})(\boldsymbol{\theta}^t - \boldsymbol{\theta}^\star), \mathbf{J}(\boldsymbol{\theta}^{t+1} - \boldsymbol{\theta}^\star) \rangle \tag{25}$$
$$+ 2\alpha m \langle (\nabla^2 \bar{\mathcal{L}} - \nabla^2 \mathcal{L})(\mathbf{J}\boldsymbol{\theta}^t - \boldsymbol{\theta}^\star), (\mathbf{W} - \mathbf{J})(\boldsymbol{\theta}^{t+1} - \boldsymbol{\theta}^\star) \rangle \tag{26}$$
$$+ 2\alpha m \langle (\nabla^2 \bar{\mathcal{L}} - \nabla^2 \mathcal{L})(\mathbf{W} - \mathbf{J})(\boldsymbol{\theta}^t - \boldsymbol{\theta}^\star), (\mathbf{W} - \mathbf{J})(\boldsymbol{\theta}^{t+1} - \boldsymbol{\theta}^\star) \rangle \tag{27}$$
$$+ 2\alpha m \langle \mathbf{W}(\nabla \bar{\mathcal{L}}(\boldsymbol{\theta}^\star) - \nabla \mathcal{L}(\boldsymbol{\theta}^\star)), \boldsymbol{\theta}^{t+1} - \boldsymbol{\theta}^\star \rangle.$$

The above factorization decomposes the quantities in the inner product in two contributions, those along the consensus directions $\mathbf{J}\boldsymbol{\theta}^{t+1}$ and $\mathbf{J}\boldsymbol{\theta}^{t+1}$, and those in the orthogonal space. Terms along the consensus directions are bounded employing Lemma 3 (see Sec. E) while those orthogonal to consensus, i.e. dependent on $(\mathbf{W} - \mathbf{J})$, are bounded using Lemma 4 (see Sec. E).

More specifically, denote by $\gamma_\ell \triangleq \max \{ \gamma_{\ell,1}, \gamma_{\ell,2} \}$; invoking Lemma 3 and Lemma 4, we have

$$2\alpha m \langle \mathbf{W}(\nabla \bar{\mathcal{L}}(\mathbf{W}\boldsymbol{\theta}^t) - \nabla \mathcal{L}(\mathbf{W}\boldsymbol{\theta}^t)), \boldsymbol{\theta}^{t+1} - \boldsymbol{\theta}^\star \rangle$$
$$\alpha \gamma_{g,1} \left( \|\mathbf{J}\boldsymbol{\theta}^t - \boldsymbol{\theta}^\star\|_{\Sigma'}^2 + \|\mathbf{J}\boldsymbol{\theta}^{t+1} - \boldsymbol{\theta}^\star\|_{\Sigma'}^2 \right) + \alpha \gamma_{g,2} \|\mathbf{J}\boldsymbol{\theta}^t - \mathbf{J}\boldsymbol{\theta}^{t+1}\|_{\Sigma'}^2$$
$$+ \gamma \left( 6m\sqrt{m}\rho \sigma_1(\Sigma')\gamma_\ell + 3\rho^2 m^3 \sigma_1(\Sigma')\gamma_\ell \right) \left( \|\boldsymbol{\theta}^t - \boldsymbol{\theta}^\star\|_2^2 + \|\boldsymbol{\theta}^{t+1} - \boldsymbol{\theta}^\star\|_2^2 \right)$$
$$+ 3\alpha m \tau_g \left( \mathcal{R}^2(\boldsymbol{\theta}_{\text{av}}^t - \boldsymbol{\theta}^\star) + \mathcal{R}^2(\boldsymbol{\theta}_{\text{av}}^{t+1} - \boldsymbol{\theta}^\star) \right)$$
$$+ \alpha 3 \left( \rho^2 m^3 + \rho m\sqrt{m} \right) \sum_{i=1}^m \tau_\ell \left( \mathcal{R}^2(\boldsymbol{\theta}_i^t - \boldsymbol{\theta}^\star) + \mathcal{R}^2(\boldsymbol{\theta}_i^{t+1} - \boldsymbol{\theta}^\star) \right) \tag{28}$$
$$+ \alpha 3m\sqrt{m}\rho \sum_{i=1}^m \tau_\ell \left( \mathcal{R}^2(\boldsymbol{\theta}_{\text{av}}^t - \boldsymbol{\theta}^\star) + \mathcal{R}^2(\boldsymbol{\theta}_{\text{av}}^{t+1} - \boldsymbol{\theta}^\star) \right)$$
$$+ 2\alpha m \langle \mathbf{W}(\nabla \bar{\mathcal{L}}(\boldsymbol{\theta}^\star) - \nabla \mathcal{L}(\boldsymbol{\theta}^\star)), \boldsymbol{\theta}^{t+1} - \boldsymbol{\theta}^\star \rangle.$$

Observe that because $\boldsymbol{\theta}^\star$ is the solution to (19) $\nabla \bar{\mathcal{L}}(\boldsymbol{\theta}^\star) = 0$.

In order to relate $\mathcal{R}(\cdot)$ to $\| \cdot \|_2$, we can proceed as in [2, Lemma 1] and write: for any $a > 0$,

$$\mathcal{R}^2(\boldsymbol{\theta}_i^t - \boldsymbol{\theta}^\star) \le 4(1+a)\Psi^2(\bar{\mathcal{M}})\|\boldsymbol{\theta} - \boldsymbol{\theta}^\star\|_2^2 + 4 \left( 1 + \frac{1}{a} \right) \mathcal{R}^2(\Pi_{\mathcal{M}^\perp}(\boldsymbol{\theta}^\star)), \quad \forall i \in [m]. \tag{29}$$

For convenience, we set $a = 1$ and define
$$\eta \triangleq 2\mathcal{R}^2(\Pi_{\mathcal{M}^\perp}(\theta^\star)).$$

The analogous to (29) holds for $\theta_i^{t+1}$, and $\theta_{\text{av}}^t$ and $\theta_{\text{av}}^{t+1}$ (the latter due to the convexity of the norm-ball). Consequently

$$\mathcal{R}^2(\theta_{\text{av}}^t - \theta^\star) + \mathcal{R}^2(\theta_{\text{av}}^{t+1} - \theta^\star) \leq 8\Psi^2(\bar{\mathcal{M}})(\|\theta_{\text{av}}^t - \theta^\star\|_2^2 + \|\theta_{\text{av}}^{t+1} - \theta^\star\|_2^2) + 8\eta$$
$$= \frac{8\Psi^2(\bar{\mathcal{M}})}{m}(\|\mathbf{J}\theta^t - \theta^\star\|_2^2 + \|\mathbf{J}\theta^{t+1} - \theta^\star\|_2^2) + 8\eta, \tag{30a}$$

and

$$\sum_{i=1}^m \left(\mathcal{R}^2(\theta_i^t - \theta^\star) + \mathcal{R}^2(\theta_i^{t+1} - \theta^\star)\right) \leq \sum_{i=1}^m 8\Psi^2(\bar{\mathcal{M}})\|\theta_i^t - \theta^\star\|_2^2 + \sum_{j=1}^m 8\Psi^2(\bar{\mathcal{M}})\|\theta_j^{t+1} - \theta^\star\|_2^2$$
$$+ 8m\eta = 8\Psi^2(\bar{\mathcal{M}})\|\theta^t - \theta^\star\|_2^2 + 8\Psi^2(\bar{\mathcal{M}})\|\theta^{t+1} - \theta^\star\|_2^2 + 8m\eta. \tag{30b}$$

Combining (28) and (30) and rearranging terms, yields

$$2\alpha m\langle\mathbf{W}(\nabla^2\bar{\mathcal{L}}(\mathbf{W}\theta^t) - \nabla^2\mathcal{L}(\mathbf{W}\theta^t)), \theta^{t+1} - \theta^\star\rangle \leq \alpha\gamma_{g,1}(\|\mathbf{J}\theta^t - \theta^\star\|_{\Sigma'}^2 + \|\mathbf{J}\theta^{t+1} - \theta^\star\|_{\Sigma'}^2)$$
$$+ \alpha\gamma_{g,2}\|\mathbf{J}\theta^t - \mathbf{J}\theta^{t+1}\|_{\Sigma'}^2$$
$$+ \alpha\left(6m\sqrt{m}\rho\sigma_1(\Sigma')\gamma_\ell + 3\rho^2 m^4\sigma_1(\Sigma')\gamma_\ell\right)\left(\|\theta^t - \theta^\star\|_2^2 + \|\theta^{t+1} - \theta^\star\|_2^2\right)$$
$$+ \alpha\left(24(\rho^2 m^3 + 2\rho m\sqrt{m})\tau_\ell\Psi^2(\bar{\mathcal{M}})\right)\left(\|\theta^t - \theta^\star\|_2^2 + \|\theta^{t+1} - \theta^\star\|_2^2\right)$$
$$+ 24\alpha\tau_g\Psi^2(\bar{\mathcal{M}})\left(\|\mathbf{J}\theta^t - \theta^\star\|_2^2 + \|\mathbf{J}\theta^{t+1} - \theta^\star\|_2^2\right) + \alpha\eta m\left(24\tau_g + 12\tau_\ell(\rho^2 m^3 + 2\rho m\sqrt{m})\right)$$
$$+ 2\alpha\langle\mathbf{W}(\nabla\mathcal{L}(\theta^\star)), \theta^{t+1} - \theta^\star\rangle. \tag{31}$$

We now focus on the remaining inner product, which is the main contributor to the final tolerance. We separate the remaining inner product in two terms: a part that will correspond to the centralized achievable statistical accuracy and an additional one which will represent the cost of decentralization. Specifically,

$$2\alpha m\langle\mathbf{W}(-\nabla\mathcal{L}(\theta^\star)), \theta^{t+1} - \theta^\star\rangle = 2\alpha m\langle-\nabla\mathcal{L}(\theta^\star), \mathbf{J}(\theta^{t+1} - \theta^\star)\rangle$$
$$+ 2\alpha m\langle(\nabla\mathcal{L}(\theta^\star)), (\mathbf{W} - \mathbf{J})(\theta^{t+1} - \theta^\star)\rangle.$$

Observe that

$$\langle-\nabla\mathcal{L}(\theta^\star), \mathbf{J}(\theta^{t+1} - \theta^\star)\rangle = \langle-\nabla\mathcal{L}(\theta^\star), \theta_{\text{av}}^{t+1} - \theta^\star\rangle$$

$$\langle-\nabla\mathcal{L}(\theta^\star), (\mathbf{W} - \mathbf{J})(\theta^{t+1} - \theta^\star)\rangle = -\frac{1}{m}\sum_{i=1}^m\left\langle\sum_{j=1}^m(w_{i,j} - 1/m)\nabla\mathcal{L}_j(\theta^\star), \theta_i^{t+1} - \theta^\star\right\rangle.$$

Therefore, using Hölder's inequality

$$2\alpha m\langle\mathbf{W}(-\nabla\mathcal{L}(\theta^\star)), \theta^{t+1} - \theta^\star\rangle \leq 2\alpha m\mathcal{R}^*\left(\nabla\mathcal{L}(\theta^\star)\right)\mathcal{R}(\theta_{\text{av}}^{t+1} - \theta^\star)$$

$$+ 2\alpha\sum_{i=1}^m\mathcal{R}^*\left(\sum_{j=1}^m(w_{i,j} - 1/m)\nabla\mathcal{L}_j(\theta^\star)\right)\mathcal{R}(\theta_i^{t+1} - \theta^\star).$$

From [2, Lemma 1] we deduce

$$\mathcal{R}(\theta - \theta^\star) \leq 2\Psi(\bar{\mathcal{M}})\|\theta - \theta^\star\|_2 + 2\mathcal{R}(\Pi_{\mathcal{M}}^\perp(\theta^\star)).$$

Then, it holds

$$2\alpha m\langle\mathbf{W}(-\nabla\mathcal{L}(\theta^\star)), \theta^{t+1} - \theta^\star\rangle \leq 4\alpha m\Psi(\bar{\mathcal{M}})\mathcal{R}^*(\nabla\mathcal{L}(\theta^\star))\|\bar{\theta}^{t+1} - \theta^\star\|_2$$

$$+ 4\alpha\Psi(\bar{\mathcal{M}})\sum_{i=1}^m\|\theta_i^{t+1} - \theta^\star\|\mathcal{R}^*\left(\sum_{j\in\mathcal{N}_i}(w_{i,j} - 1/m)\nabla\mathcal{L}_j(\theta^\star)\right)$$

$$+ 4\gamma\mathcal{R}(\Pi_{\mathcal{M}^\perp}(\theta^\star))\left(m\mathcal{R}^*\left(\nabla\mathcal{L}(\theta^\star)\right) + \sum_{i=1}^m\mathcal{R}^*\left(\sum_{j\in\mathcal{N}_i}(w_{i,j} - 1/m)\nabla\mathcal{L}_j(\theta^\star)\right)\right).$$

Applying Young's inequality we have

$$2\alpha m \langle \mathbf{W}(\nabla\bar{\mathcal{L}}(\boldsymbol{\theta}^\star) - \nabla\mathcal{L}(\boldsymbol{\theta}^\star)), \boldsymbol{\theta}^{t+1} - \boldsymbol{\theta}^\star \rangle \leq \alpha m \frac{2}{\varepsilon} \Psi^2(\bar{\mathcal{M}}) \mathcal{R}^*(\nabla\mathcal{L}(\boldsymbol{\theta}^\star))^2 \tag{32}$$

$$+ 2\varepsilon\alpha\|\mathbf{J}\boldsymbol{\theta}^{t+1} - \boldsymbol{\theta}^\star\|_2^2 + 2\frac{\alpha}{\epsilon}\sum_{i=1}^{m}\left(\Psi(\bar{\mathcal{M}})\mathcal{R}^*\left(\left(\sum_{j\in\mathcal{N}_i} w_{i,j} - 1/m\right)\nabla\mathcal{L}_j(\boldsymbol{\theta}^\star)\right)\right)^2 \tag{33}$$

$$+ 2\alpha\epsilon\|\boldsymbol{\theta}^{t+1} - \boldsymbol{\theta}^\star\|_2^2 + 4\alpha\mathcal{R}(\Pi_{\mathcal{M}^\perp}(\boldsymbol{\theta}^\star))(m\mathcal{R}^*(\nabla\mathcal{L}(\boldsymbol{\theta}^\star))) \tag{34}$$

$$+ 4\alpha\mathcal{R}(\Pi_{\mathcal{M}^\perp}(\boldsymbol{\theta}^\star))\sum_{i=1}^{m}\mathcal{R}^*\left(\left(\sum_{j\in\mathcal{N}_i} w_{i,j} - 1/m\right)\nabla\mathcal{L}_j(\boldsymbol{\theta}^\star)\right) \tag{35}$$

for any $\varepsilon, \epsilon > 0$.

For convenience, let $R_t^2 \triangleq \|\boldsymbol{\theta}^t - \boldsymbol{\theta}^\star\|_2^2$. Combining the bounds (23), (31) and (34) yields

$$R_{t+1}^2 \leq \|\mathbf{W}\boldsymbol{\theta}^t - \boldsymbol{\theta}^\star\|_2^2 - \alpha m \left(\|\mathbf{W}(\boldsymbol{\theta}^t - \boldsymbol{\theta}^\star)\|_{\nabla^2\bar{\mathcal{L}}}^2 - \frac{\gamma_{g,1}}{m}\|\mathbf{J}\boldsymbol{\theta}^t - \boldsymbol{\theta}^\star\|_{\Sigma'}^2 + \|\mathbf{W}(\boldsymbol{\theta}^{t+1} - \boldsymbol{\theta}^\star)\|_{\nabla^2\bar{\mathcal{L}}}^2\right)$$

$$+ 2\alpha(\varepsilon + \epsilon)R_{t+1}^2 + \alpha m\frac{\gamma_{g,1}}{m}\|\mathbf{J}\boldsymbol{\theta}^{t+1} - \boldsymbol{\theta}^\star\|_{\Sigma'}^2 + (\gamma L - 1)\|\mathbf{W}(\boldsymbol{\theta}^t - \boldsymbol{\theta}^{t+1})\|_2^2$$

$$- \|\boldsymbol{\theta}^{t+1}\|_{\mathbf{I}-\mathbf{W}^2}^2 + \alpha\left(24\left(\rho^2 m^3 + \rho m\sqrt{m} + m\sqrt{m}\rho + \frac{\tau_g}{\tau_\ell}\right)\tau_\ell\Psi^2(\bar{\mathcal{M}})\right)(R_t^2 + R_{t+1}^2)$$

$$+ \alpha\left(6m\sqrt{m}\rho\sigma_1(\Sigma')\gamma_\ell + 3\rho^2 m^3\sigma_1(\Sigma')\gamma_\ell\right)(R_t^2 + R_{t+1}^2) + \alpha\gamma_{g,2}\|\mathbf{J}\boldsymbol{\theta}^t - \mathbf{J}\boldsymbol{\theta}^{t+1}\|_{\Sigma'}^2$$

$$+ 2\alpha m\Psi^2(\bar{\mathcal{M}})\frac{(\mathcal{R}^*(\nabla\mathcal{L}(\boldsymbol{\theta}^\star)))^2}{\varepsilon} + 2\alpha\Psi^2(\bar{\mathcal{M}})\frac{\sum_{i=1}^{m}\left(\mathcal{R}^*\left(\sum_{j=1}^{m}(w_{i,j} - 1/m)\nabla\mathcal{L}_j(\boldsymbol{\theta}^\star)\right)\right)^2}{\epsilon}$$

$$+ \alpha\eta m\left(12\tau_\ell(\rho^2 m^3) + \alpha\eta m\left(2\rho m\sqrt{m} + 24\tau_g\right)\right.$$

$$+ 4\alpha\mathcal{R}(\Pi_{\mathcal{M}^\perp}(\boldsymbol{\theta}^\star))\left(m\mathcal{R}^*(\nabla\mathcal{L}(\boldsymbol{\theta}^\star)) + \sum_{i=1}^{m}\mathcal{R}^*\left(\left(\sum_{j=1}^{m} w_{i,j} - 1/m\right)\nabla\mathcal{L}_j(\boldsymbol{\theta}^\star)\right)\right).$$

Observe that

$$\|\mathbf{J}\boldsymbol{\theta} - \boldsymbol{\theta}^\star\|_{\Sigma'}^2 = \|\mathbf{J}\mathbf{W}\boldsymbol{\theta} - \boldsymbol{\theta}^\star\|_\Sigma^2 \leq \|\mathbf{W}\boldsymbol{\theta} - \boldsymbol{\theta}^\star\|_\Sigma^2.$$

Under $\nabla^2\bar{\mathcal{L}} - \gamma_{g,1}\Sigma' \succeq \frac{\mu}{2}I_d$, and using $\|\mathbf{W}\boldsymbol{\theta}^t - \boldsymbol{\theta}^\star\|^2 \leq \|\boldsymbol{\theta}^t - \boldsymbol{\theta}^\star\|^2$ and $\mathbf{W}^2 \succeq \mathbf{J}$ along with the $\frac{\mu}{m}$−strong convex of $\bar{\mathcal{L}}$, we obtain

$$R_{t+1}^2 \leq \left(1 - \alpha\frac{\mu}{2}\right)R_t^2 - \alpha\frac{\mu}{2}\|\mathbf{W}\boldsymbol{\theta}^{t+1} - \boldsymbol{\theta}^\star\|_2^2 - \|\boldsymbol{\theta}^{t+1}\|_{\mathbf{I}-\mathbf{W}^2}^2$$

$$+ (\alpha L + \alpha\gamma_{g,2}\sigma_1(\Sigma') - 1)\|\mathbf{W}(\boldsymbol{\theta}^t - \boldsymbol{\theta}^{t+1})\|^2$$

$$+ 2\alpha(\varepsilon + \epsilon)R_{t+1}^2 + 2\alpha\Psi^2(\bar{\mathcal{M}})\left(m\frac{(\mathcal{R}^*(\nabla\mathcal{L}(\boldsymbol{\theta}^\star)))^2}{\varepsilon} + \frac{\sum_{i=1}^{m}\left(\sum_{j=1}^{m}(w_{i,j} - 1/m)\nabla\mathcal{L}_i(\boldsymbol{\theta}^\star)\right)^2}{\epsilon}\right)$$

$$+ \alpha\left(12(1 + a)\left(\rho^2 m^3 + \rho m\sqrt{m} + m\sqrt{m}\rho + \frac{\tau_g}{\tau_\ell}\right)\tau_\ell\Psi^2(\bar{\mathcal{M}})\right)(R_t^2 + R_{t+1}^2)$$

$$+ \alpha\left(6m\sqrt{m}\rho\sigma_1(\Sigma')\gamma_\ell + 3\rho^2 m^3\sigma_1(\Sigma')\gamma_\ell\right)(R_{t+1}^2 + R_t^2) + \alpha\eta m\left(24\tau_g + 12\tau_\ell(\rho^2 m^3 + 2\rho m\sqrt{m})\right)$$

$$+ 4\alpha\mathcal{R}(\Pi_{\mathcal{M}^\perp}(\boldsymbol{\theta}^\star))(m\mathcal{R}^*(\nabla\mathcal{L}(\boldsymbol{\theta}^\star))) + 4\alpha\mathcal{R}(\Pi_{\mathcal{M}^\perp}(\boldsymbol{\theta}^\star))\sum_{i=1}^{m}\mathcal{R}^*\left(\left(\sum_{j=1}^{m} w_{i,j} - 1/m\right)\nabla\mathcal{L}_j(\boldsymbol{\theta}^\star)\right).$$

Under $\alpha \leq \frac{1}{L}$, which implies $\alpha\frac{\mu}{2} < 1$, and using $\|\mathbf{W}\boldsymbol{\theta}^{t+1} - \boldsymbol{\theta}^\star\|^2 \leq \|\boldsymbol{\theta}^{t+1} - \boldsymbol{\theta}^\star\|^2$, yields

$$-\alpha\frac{\mu}{2}\|\mathbf{W}\boldsymbol{\theta}^{t+1} - \boldsymbol{\theta}^\star\|^2 - \|\boldsymbol{\theta}^{t+1} - \boldsymbol{\theta}^\star\|_{\mathbf{I}-\mathbf{W}^2}^2 \leq -\alpha\frac{\mu}{2}\|\boldsymbol{\theta}^{t+1} - \boldsymbol{\theta}^\star\|_2^2 = -\alpha\frac{\mu}{2}R_{t+1}^2.$$

Choosing

$$\varepsilon = \frac{3\mu}{16}, \qquad\qquad\qquad \epsilon = \frac{\mu}{32},$$

$$3\gamma_\ell \sigma_1(\Sigma')m^{1.5}\rho(2 + \rho m^{1.5}) \leq \frac{\mu}{36}, \qquad \alpha \leq \frac{1}{L + \gamma_{g,2}\sigma_1(\Sigma')}, \qquad (36)$$

we have

$$R_{t+1}^2 \leq \left(1 - \alpha\frac{\mu}{2} + \alpha 6m\sqrt{m}\rho\sigma_1(\Sigma')\gamma_\ell + 3\alpha\rho^2 m^3 \sigma_1(\Sigma')\gamma_\ell\right)R_t^2 + \alpha 24\left(\tau_g\Psi^2(\bar{\mathcal{M}})\right)(R_t^2 + R_{t+1}^2)$$

$$+ \alpha\left(24\left(\rho^2 m^3 + 2\rho m\sqrt{m}\right)\tau_\ell\Psi^2(\bar{\mathcal{M}})\right)(R_t^2 + R_{t+1}^2) + \alpha\eta m\left(24\tau_g + 24\tau_\ell\rho m\sqrt{m}\right)$$

$$+ 32\alpha m\Psi^2(\bar{\mathcal{M}})\left(\frac{(\mathcal{R}^*(\nabla\mathcal{L}(\theta^\star)))^2}{3\mu} + \frac{\sum_{i=1}^m\left(\mathcal{R}^*(\sum_{j=1}^m(w_{i,j} - 1/m)\nabla\mathcal{L}_j(\theta^\star))\right)^2}{m\mu}\right)$$

$$+ 4\alpha\mathcal{R}(\Pi_{\mathcal{M}^\perp}(\theta^\star))\left(m\mathcal{R}^*(\nabla\mathcal{L}(\theta^\star)) + \sum_{i=1}^m\mathcal{R}^*\left(\left(\sum_{j=1}^m w_{i,j} - 1/m\right)\nabla\mathcal{L}_j(\theta^\star)\right)\right)$$

$$+ \alpha\eta(12\tau_\ell\rho^2 m^3).$$

If $1 - \alpha\zeta > 0$, which is implied by

$$\zeta \triangleq 24\tau_\ell\Psi^2(\bar{\mathcal{M}})\left(\rho^2 m^3 + 2\rho m\sqrt{m} + \frac{\tau_g}{\tau_\ell}\right) < \frac{1}{\alpha}, \qquad (37)$$

and one divides both sides by $1 - \alpha\zeta$, yields

$$R_{t+1}^2 \leq \frac{\left(1 - \alpha\frac{\mu}{2} + \alpha 3\sigma_1(\Sigma')\gamma_\ell(2m\sqrt{m}\rho + \rho^2 m^3) + \alpha\zeta\right)}{1 - \alpha\zeta}r_t^2$$

$$+ \frac{24\alpha m\mathcal{R}^2\left(\Pi_{\mathcal{M}^\perp}(\theta^\star)\right)\left(2\tau_g + \tau_\ell(\rho^2 m^3 + 2\rho m\sqrt{m})\right)}{1 - \alpha\zeta}$$

$$+ \frac{32\alpha m\Psi^2(\bar{\mathcal{M}})}{1 - \alpha\zeta}\left(\frac{(\mathcal{R}^*(\nabla\mathcal{L}(\theta^\star)))^2}{3\mu} + \frac{\sum_{i=1}^m\left(\mathcal{R}^*\left(\sum_{j=1}^m(w_{i,j} - 1/m)\nabla\mathcal{L}_j(\theta^\star)\right)\right)^2}{m\mu},\right)$$

$$+ \frac{4\alpha m\mathcal{R}(\Pi_{\mathcal{M}^\perp}(\theta^\star))}{1 - \alpha\zeta}\left(\mathcal{R}^*(\nabla\mathcal{L}(\theta^\star)) + \frac{1}{m}\sum_{i=1}^m\mathcal{R}^*\left(\left(\sum_{j=1}^m w_{i,j} - 1/m\right)\nabla\mathcal{L}_j(\theta^\star)\right)\right).$$

Contraction (up to some tolerance) is guaranteed under

$$1 - \alpha\frac{\mu}{2} + \alpha 3\sigma_1(\Sigma')\gamma_\ell(2m\sqrt{m}\rho + \rho^2 m^3) + \alpha\zeta < 1 - \alpha\zeta,$$

enforced via

$$\zeta < \frac{2}{9}\mu. \qquad (38)$$

Observe that

$$\frac{1}{1 - \alpha\zeta} = 1 + \frac{\alpha\zeta}{1 - \alpha\zeta}.$$

Let

$$\lambda \triangleq \frac{1 - \alpha\frac{\mu}{2} + \alpha\zeta + 3\alpha\sigma_1(\Sigma')\gamma_\ell(2m\sqrt{m}\rho + \rho^2 m^3)}{1 - \alpha\zeta}.$$

Then, it holds

$$R_{t+1}^2 \leq \lambda R_t^2 + \frac{32\alpha m \Psi^2(\bar{\mathcal{M}})}{\mu} \left( \frac{(\mathcal{R}^*(\nabla\mathcal{L}(\theta^\star)))^2}{3} + \frac{1}{m}\sum_{i=1}^m \left( \mathcal{R}^* \left( \sum_{j=1}^m (w_{i,j} - 1/m)\nabla\mathcal{L}_j(\theta^\star) \right) \right)^2 \right) \times$$

$$\times \left( 1 + \frac{\alpha\zeta}{1-\alpha\zeta} \right) + \left( 1 + \frac{\alpha\zeta}{1-\alpha\zeta} \right) \left( 24\alpha m \mathcal{R}^2(\Pi_{\mathcal{M}^\perp}(\theta^\star))(2\tau_g + \tau_\ell(\rho^2 m^3 + 2\rho m\sqrt{m})) \right)$$

$$+ \left( 1 + \frac{\alpha\zeta}{1-\alpha\zeta} \right) 4\alpha\mathcal{R}(\Pi_{\mathcal{M}^\perp}(\theta^\star)) \left( m\mathcal{R}^*(\nabla\mathcal{L}(\theta^\star)) + \sum_{i=1}^m \mathcal{R}^* \left( \left( \sum_{j=1}^m w_{i,j} - 1/m \right) \nabla\mathcal{L}_j(\theta^\star) \right) \right).$$

Telescoping, we obtain

$$R_{t+1}^2 \leq \lambda^{t+1} R_0^2 + \frac{32\alpha m \Psi^2(\bar{\mathcal{M}})}{\mu(1-\lambda)} \left( \frac{(\mathcal{R}^*(\nabla\mathcal{L}(\theta^\star)))^2}{3} + \frac{1}{m}\sum_{i=1}^m \left( \mathcal{R}^* \left( \sum_{j=1}^m (w_{i,j} - 1/m)\nabla\mathcal{L}_n^{(j)}(\theta^\star) \right) \right)^2 \right) \times$$

$$\times \left( 1 + \frac{\alpha\zeta}{1-\alpha\zeta} \right) + (1-\lambda)^{-1} \left( 1 + \frac{\alpha\zeta}{1-\alpha\zeta} \right) \left( 24\alpha m \mathcal{R}^2(\Pi_{\mathcal{M}^\perp}(\theta^\star))(2\tau_g + \tau_\ell(\rho^2 m^3 + 2\rho m\sqrt{m})) \right)$$

$$+ \left( 1 + \frac{\alpha\zeta}{1-\alpha\zeta} \right) \frac{4}{1-\lambda} \mathcal{R}(\Pi_{\mathcal{M}^\perp}(\theta^\star)) \left( m\mathcal{R}^*(\nabla\mathcal{L}(\theta^\star)) + \sum_{i=1}^m \mathcal{R}^* \left( \left( \sum_{j=1}^m w_{i,j} - 1/m \right) \nabla\mathcal{L}_j(\theta^\star) \right) \right).$$

We further require

$$(1 - \lambda) \geq \frac{\alpha\mu}{6},$$

implied by

$$\zeta + 3\sigma_1(\Sigma')\gamma_\ell(2m\sqrt{m}\rho + \rho^2 m^3) \leq \frac{\mu}{6}.$$

A sufficient condition for the above is

$$3\gamma_\ell\sigma_1(\Sigma')m^{1.5}\rho(2 + \rho\, m^{1.5}) \leq \frac{\mu}{36},$$

$$\zeta \leq \frac{5\mu}{36},$$

which is compatible with requirements (36),(37) and (38), and fulfilled whenever

$$\rho \leq \frac{\mu}{108\sigma_1(\Sigma')\gamma_\ell m^{1.5}}.$$

Dividing by $m$ and observing that $R_t^2 = m r_t^2$ yields the desired result.

# E    Technical Lemmata

In this section we provide the proofs of some technical intermediate results, used in the proof of Theorem 2.

## E.1    Proof of Lemma 2

We are interested in establishing an upper bound on the estimation error $\|\theta^{t+1} - \theta^\star\|^2$ based on that at time $t$. Observe that

$$\|\theta^{t+1} - \theta^\star\|^2 = \|\theta^t - \theta^\star\|^2 + 2\langle\theta^{t+1} - \theta^t, \theta^t - \theta^\star\rangle + \|\theta^t - \theta^{t+1}\|^2. \qquad (39)$$

We proceed to upper bound the latter two terms.

From (20) it follows that

$$\frac{1}{2}\|\mathbf{W}\theta^t - \theta^\star\|_{\nabla^2\bar{\mathcal{L}}}^2 + \frac{1}{2\alpha m}\|\theta^t\|_{\mathbf{I}-\mathbf{W}^2}^2 \leq \bar{\mathcal{L}}^\alpha(\theta^\star) - \bar{\mathcal{L}}^\alpha(\theta^t) - \langle\nabla\bar{\mathcal{L}}^\alpha(\theta^t), \theta^{t+1} - \theta^t\rangle$$

$$- \langle\nabla\bar{\mathcal{L}}^\alpha(\theta^t), \theta^\star - \theta^{t+1}\rangle.$$

Furthermore, still from (20),

$$\frac{1}{2}\|\mathbf{W}\boldsymbol{\theta}^t - \boldsymbol{\theta}^\star\|^2_{\nabla^2\bar{\mathcal{L}}} + \frac{1}{2\alpha m}\|\boldsymbol{\theta}^t\|^2_{I-\mathbf{W}^2} \leq \bar{\mathcal{L}}^\alpha(\boldsymbol{\theta}^\star) - \bar{\mathcal{L}}^\alpha(\boldsymbol{\theta}^{t+1}) + \frac{1}{2}\|\mathbf{W}(\boldsymbol{\theta}^{t+1} - \boldsymbol{\theta}^t)\|^2_{\nabla^2\bar{\mathcal{L}}}$$
$$+ \frac{1}{2\alpha m}\|\boldsymbol{\theta}^t - \boldsymbol{\theta}^{t+1}\|^2_{I-\mathbf{W}^2} - \langle\nabla\bar{\mathcal{L}}^\alpha(\boldsymbol{\theta}^t), \boldsymbol{\theta}^\star - \boldsymbol{\theta}^{t+1}\rangle.$$

Observe that

$$\bar{\mathcal{L}}^\alpha(\boldsymbol{\theta}^\star) = \bar{\mathcal{L}}(\boldsymbol{\theta}^\star) \quad \text{and} \quad \bar{\mathcal{L}}^\alpha(\boldsymbol{\theta}^{t+1}) = \bar{\mathcal{L}}(\mathbf{W}\boldsymbol{\theta}^{t+1}) + \frac{1}{2\alpha m}\|\boldsymbol{\theta}^{t+1}\|^2_{I-\mathbf{W}^2}.$$

Furthermore, since

$$\bar{\mathcal{L}}(\mathbf{W}\boldsymbol{\theta}^{t+1}) \geq \bar{\mathcal{L}}(\boldsymbol{\theta}^\star) + \langle\nabla\bar{\mathcal{L}}(\boldsymbol{\theta}^\star), \mathbf{W}(\boldsymbol{\theta}^{t+1} - \boldsymbol{\theta}^\star)\rangle + \frac{1}{2}\|\mathbf{W}\boldsymbol{\theta}^{t+1} - \boldsymbol{\theta}^\star\|^2_{\nabla^2\bar{\mathcal{L}}}$$

and $\langle\nabla\bar{\mathcal{L}}(\boldsymbol{\theta}^\star), \mathbf{W}\boldsymbol{\theta}^{t+1} - \boldsymbol{\theta}^\star\rangle \geq 0$, it holds

$$\frac{1}{2}\|\mathbf{W}\boldsymbol{\theta}^t - \boldsymbol{\theta}^\star\|^2_{\nabla^2\bar{\mathcal{L}}} + \frac{1}{2\alpha m}\|\boldsymbol{\theta}^t\|^2_{I-\mathbf{W}^2} \leq -\frac{1}{2}\|\mathbf{W}\boldsymbol{\theta}^{t+1} - \boldsymbol{\theta}^\star\|^2_{\nabla^2\bar{\mathcal{L}}} - \frac{1}{2\alpha m}\|\boldsymbol{\theta}^{t+1}\|^2_{I-\mathbf{W}^2} \quad (40)$$
$$+ \frac{1}{2}\|\mathbf{W}(\boldsymbol{\theta}^{t+1} - \boldsymbol{\theta}^t)\|^2_{\nabla^2\bar{\mathcal{L}}} + \frac{1}{2\alpha m}\|\boldsymbol{\theta}^t - \boldsymbol{\theta}^{t+1}\|^2_{I-\mathbf{W}^2} + \langle\nabla\bar{\mathcal{L}}^\alpha(\boldsymbol{\theta}^t), \boldsymbol{\theta}^{t+1} - \boldsymbol{\theta}^\star\rangle. \quad (41)$$

By the optimality of $\boldsymbol{\theta}^{t+1}$,

$$\boldsymbol{\theta}^{t+1} = \operatorname*{argmin}_{\theta_i \in \Omega \,:\, \mathcal{R}(\theta_i) \leq R, \forall i \in [m]} \left\{ \langle\nabla\mathcal{L}^\alpha(\boldsymbol{\theta}^t), \boldsymbol{\theta} - \boldsymbol{\theta}^t\rangle + \frac{1}{2\alpha m}\|\boldsymbol{\theta} - \boldsymbol{\theta}^t\|^2 \right\},$$

it follows

$$\langle\nabla\bar{\mathcal{L}}^\alpha(\boldsymbol{\theta}^t) - \mathbf{W}\left(\nabla\bar{\mathcal{L}}(\mathbf{W}\boldsymbol{\theta}^t) - \nabla\mathcal{L}(\mathbf{W}\boldsymbol{\theta}^t)\right) + \frac{1}{\alpha m}(\boldsymbol{\theta}^{t+1} - \boldsymbol{\theta}^t), \boldsymbol{\theta} - \boldsymbol{\theta}^{t+1}\rangle \geq 0, \quad (42)$$

for any feasible $\boldsymbol{\theta}$. Setting, in particular, $\boldsymbol{\theta} = \boldsymbol{\theta}^\star$ and combining (42) with (40), yield

$$\frac{1}{\alpha m}\langle\boldsymbol{\theta}^{t+1} - \boldsymbol{\theta}^t, \boldsymbol{\theta}^t - \boldsymbol{\theta}^\star\rangle + \frac{1}{2\alpha m}\|\boldsymbol{\theta}^{t+1} - \boldsymbol{\theta}^t\|^2_2 \leq -\frac{1}{2}\left(\|\mathbf{W}\boldsymbol{\theta}^t - \boldsymbol{\theta}^\star\|^2_{\nabla^2\bar{\mathcal{L}}} + \|\mathbf{W}\boldsymbol{\theta}^{t+1} - \boldsymbol{\theta}^\star\|^2_{\nabla^2\bar{\mathcal{L}}}\right)$$
$$- \frac{1}{2\alpha m}\left(\|\boldsymbol{\theta}^t\|^2_{I-\mathbf{W}^2} + \|\boldsymbol{\theta}^{t+1}\|^2_{I-\mathbf{W}^2}\right) + \frac{1}{2}\|\mathbf{W}(\boldsymbol{\theta}^{t+1} - \boldsymbol{\theta}^t)\|^2_{\nabla^2\bar{\mathcal{L}}} + \frac{1}{2\alpha m}\|\boldsymbol{\theta}^t - \boldsymbol{\theta}^{t+1}\|^2_{I-\mathbf{W}^2}$$
$$- \frac{1}{2\alpha m}\|\boldsymbol{\theta}^{t+1} - \boldsymbol{\theta}^t\|^2_2 + \langle\mathbf{W}\left(\nabla\bar{\mathcal{L}}(\mathbf{W}\boldsymbol{\theta}^t) - \nabla\mathcal{L}(\mathbf{W}\boldsymbol{\theta}^t)\right), \boldsymbol{\theta}^{t+1} - \boldsymbol{\theta}^\star\rangle.$$

Multiplying the above by $2\alpha m$ and combining with (39) yield

$$\|\boldsymbol{\theta}^{t+1} - \boldsymbol{\theta}^\star\|^2_2 \leq \|\boldsymbol{\theta}^t - \boldsymbol{\theta}^\star\|^2_2 - \alpha m\left(\|\mathbf{W}(\boldsymbol{\theta}^t - \boldsymbol{\theta}^\star)\|^2_{\nabla^2\bar{\mathcal{L}}} + \|\mathbf{W}\boldsymbol{\theta}^{t+1} - \boldsymbol{\theta}^\star\|^2_{\nabla^2\bar{\mathcal{L}}}\right)$$
$$- \|\boldsymbol{\theta}^t\|^2_{I-\mathbf{W}^2} - \|\boldsymbol{\theta}^{t+1}\|^2_{I-\mathbf{W}^2}$$
$$+ \alpha m\|\mathbf{W}(\boldsymbol{\theta}^{t+1} - \boldsymbol{\theta}^t)\|^2_{\nabla^2\bar{\mathcal{L}}} - \|\boldsymbol{\theta}^t - \boldsymbol{\theta}^{t+1}\|^2_{\mathbf{W}^2}$$
$$+ 2\alpha m\langle\mathbf{W}\left(\nabla\bar{\mathcal{L}}(\mathbf{W}\boldsymbol{\theta}^t) - \mathcal{L}(\mathbf{W}\boldsymbol{\theta}^t)\right), \boldsymbol{\theta}^{t+1} - \boldsymbol{\theta}^\star\rangle.$$

## E.2  Bounds on inner product elements (24) - (27)

**Lemma 3.** *Under Assumption 1 and 3 it holds*

$$\langle(\nabla^2\bar{\mathcal{L}} - \nabla^2\mathcal{L})(\mathbf{J}\boldsymbol{\theta} - \boldsymbol{\theta}^\star), \mathbf{J}\boldsymbol{\theta}' - \boldsymbol{\theta}^\star\rangle \leq \frac{\gamma_{g,1}}{2m}\left(\|\mathbf{J}\boldsymbol{\theta} - \boldsymbol{\theta}^\star\|^2_{\boldsymbol{\Sigma}'} + \|\mathbf{J}\boldsymbol{\theta}' - \boldsymbol{\theta}^\star\|^2_{\boldsymbol{\Sigma}'}\right) + \frac{\gamma_{g,2}}{2m}\|\mathbf{J}\boldsymbol{\theta} - \mathbf{J}\boldsymbol{\theta}'\|^2_{\boldsymbol{\Sigma}'}$$
$$(43)$$

$$+ \frac{3}{2}\tau_g\left(\mathcal{R}^2(\theta_{\text{av}} - \theta^\star) + \mathcal{R}^2(\theta'_{\text{av}} - \theta^\star)\right),$$

*where $\boldsymbol{\Sigma}' \triangleq \Sigma' \otimes I_m$, $\theta_{\text{av}} \triangleq \frac{1}{m}\sum_{i=1}^m \theta_i$ and analogousloy for $\theta'_{\text{av}}$.*

*Proof.* Observe that the RHS of (43) can be expanded as

$$\langle (\nabla^2 \bar{\mathcal{L}} - \nabla^2 \mathcal{L})(\mathbf{J}\boldsymbol{\theta} - \boldsymbol{\theta}^\star), \mathbf{J}(\boldsymbol{\theta}' - \boldsymbol{\theta}^\star) \rangle = \frac{1}{2} \langle \mathbf{J}\boldsymbol{\theta} - \boldsymbol{\theta}^\star, (\nabla^2 \bar{\mathcal{L}} - \nabla^2 \mathcal{L})(\mathbf{J}\boldsymbol{\theta} - \boldsymbol{\theta}^\star) \rangle$$

$$+ \frac{1}{2} \langle \mathbf{J}\boldsymbol{\theta}' - \boldsymbol{\theta}^\star, (\nabla^2 \bar{\mathcal{L}} - \nabla^2 \mathcal{L})(\mathbf{J}\boldsymbol{\theta}' - \boldsymbol{\theta}^\star) \rangle - \frac{1}{2} \langle \mathbf{J}\boldsymbol{\theta} - \mathbf{J}\boldsymbol{\theta}', (\nabla^2 \bar{\mathcal{L}} - \nabla^2 \mathcal{L})(\mathbf{J}\boldsymbol{\theta} - \mathbf{J}\boldsymbol{\theta}') \rangle$$

$$= \frac{1}{2} \langle (\nabla^2 \bar{\mathcal{L}} - \nabla^2 \mathcal{L})(\theta_{\text{av}} - \theta^\star), \theta_{\text{av}} - \theta^\star \rangle + \frac{1}{2} \langle (\nabla^2 \bar{\mathcal{L}} - \nabla^2 \mathcal{L})(\theta'_{\text{av}} - \theta^\star), \theta'_{\text{av}} - \theta^\star \rangle$$

$$- \frac{1}{2} \langle (\nabla^2 \bar{\mathcal{L}} - \nabla^2 \mathcal{L})(\theta_{\text{av}} - \theta'_{\text{av}}, \theta_{\text{av}} - \theta'_{\text{av}} \rangle.$$

From Assumption 3 it follows that

$$\langle (\nabla^2 \bar{\mathcal{L}} - \nabla^2 \mathcal{L})(\mathbf{J}\boldsymbol{\theta} - \boldsymbol{\theta}^\star), \mathbf{J}(\boldsymbol{\theta}' - \boldsymbol{\theta}^\star) \rangle \leq \frac{1}{2} \gamma_{g,1} \left( \|\theta_{\text{av}} - \theta^\star\|_{\Sigma'}^2 + \|\theta'_{\text{av}} - \theta^\star\|_{\Sigma'}^2 \right)$$

$$+ \frac{1}{2} \tau_g \left( \mathcal{R}^2(\theta_{\text{av}} - \theta^\star) + \mathcal{R}^2(\theta'_{\text{av}} - \theta^\star) \right) + \frac{1}{2} \gamma_{g,2} \|\theta_{\text{av}} - \theta'_{\text{av}}\|_{\Sigma'}^2 + \frac{1}{2} \tau_g \mathcal{R}^2(\theta_{\text{av}} - \theta'_{\text{av}}).$$

Then, using triangle and Young's inequality on $\mathcal{R}^2(\theta_{\text{av}} - \theta'_{\text{av}})$ we obtain

$$\langle (\nabla^2 \bar{\mathcal{L}} - \nabla^2 \mathcal{L})(\mathbf{J}\boldsymbol{\theta} - \boldsymbol{\theta}^\star), \mathbf{J}(\boldsymbol{\theta}' - \boldsymbol{\theta}^\star) \rangle \leq \frac{\gamma_{g,1}}{2m} (\|\mathbf{J}\boldsymbol{\theta} - \boldsymbol{\theta}^\star\|_{\Sigma'}^2 + \|\mathbf{J}\boldsymbol{\theta}' - \boldsymbol{\theta}^\star\|_{\Sigma'}^2) + \frac{\gamma_{g,2}}{2m} \|\mathbf{J}\boldsymbol{\theta} - \mathbf{J}\boldsymbol{\theta}'\|_{\Sigma'}^2$$

$$+ \frac{1}{2} \tau_g \left( \mathcal{R}^2(\theta_{\text{av}} - \theta^\star) + \mathcal{R}^2(\theta'_{\text{av}} - \theta^\star) \right).$$

$\square$

**Lemma 4.** *Under Assumptions 1, 3 4, and 6, and given $\gamma_\ell \triangleq \max\{\gamma_{\ell,1}, \gamma_{\ell,2}\}$, there holds*

$$\langle (\nabla^2 \bar{\mathcal{L}} - \nabla^2 \mathcal{L})(\mathbf{J}\boldsymbol{\theta} - \boldsymbol{\theta}^\star), (\mathbf{W} - \mathbf{J})(\boldsymbol{\theta}' - \boldsymbol{\theta}^\star) \rangle \leq \frac{3\sqrt{m}\rho}{2} \gamma_\ell \left( \|\boldsymbol{\theta} - \boldsymbol{\theta}^\star\|_{\Sigma'}^2 + \|\boldsymbol{\theta}' - \boldsymbol{\theta}^\star\|_{\Sigma'}^2 \right) \quad (44)$$

$$+ \frac{3\sqrt{m}\rho}{2} \sum_{j=1}^{m} \tau_\ell \left( \mathcal{R}^2(\theta_{\text{av}} - \theta^\star) + \mathcal{R}^2(\theta'_j - \theta^\star) \right)$$

$$\langle (\nabla^2 \bar{\mathcal{L}} - \nabla^2 \mathcal{L})(\mathbf{W} - \mathbf{J})(\boldsymbol{\theta} - \boldsymbol{\theta}^\star), (\mathbf{W} - \mathbf{J})(\boldsymbol{\theta}' - \boldsymbol{\theta}^\star) \rangle \leq \frac{3\rho^2 m^2 \gamma_\ell}{2} \left( \|\boldsymbol{\theta} - \boldsymbol{\theta}^\star\|_{\Sigma'}^2 + \|\boldsymbol{\theta}' - \boldsymbol{\theta}^\star\|_{\Sigma'}^2 \right)$$

$$(45)$$

$$+ \frac{3\rho^2 m^2}{2} \sum_{i=1}^{m} \tau_\ell \left( \mathcal{R}^2(\theta_i - \theta^\star) + \mathcal{R}^2(\theta'_i - \theta^\star) \right)$$

*where $\Sigma' \triangleq \Sigma' \otimes I_m$ and $\theta_{\text{av}} \triangleq \frac{1}{m} \sum_{i=1}^{m} \theta_i$ and analogously for $\theta'_{\text{av}}$.*

*Proof.* We write the first bound as

$$\langle (\nabla^2 \bar{\mathcal{L}} - \nabla^2 \mathcal{L})(\mathbf{J}\boldsymbol{\theta} - \boldsymbol{\theta}^\star), (\mathbf{W} - \mathbf{J})(\boldsymbol{\theta}' - \boldsymbol{\theta}^\star) \rangle$$

$$= \frac{1}{m} \sum_{i=1}^{m} \langle (\nabla^2 \bar{\mathcal{L}} - \nabla^2 \mathcal{L}_i)(\theta_{\text{av}} - \theta^\star), \sum_{j=1}^{m} (w_{i,j} - 1/m)(\theta'_j - \theta^\star) \rangle.$$

Then,

$$\langle (\nabla^2 \bar{\mathcal{L}} - \nabla^2 \mathcal{L})(\mathbf{J}\boldsymbol{\theta} - \boldsymbol{\theta}^\star), (\mathbf{W} - \mathbf{J})(\boldsymbol{\theta}' - \boldsymbol{\theta}^\star) \rangle$$

$$\leq \frac{1}{m} \sum_{i=1}^{m} \sum_{j=1}^{m} |w_{i,j} - 1/m| |\langle (\nabla^2 \bar{\mathcal{L}} - \nabla^2 \mathcal{L}_i)(\theta_{\text{av}} - \theta^\star), \theta'_j - \theta^\star \rangle|.$$

Using

$$|\langle v, Au \rangle| \leq \frac{1}{2} |\langle v, Av \rangle| + \frac{1}{2} |\langle u, Au \rangle| + \frac{1}{2} |\langle u - v, A(u - v) \rangle|,$$

we have

$$\langle(\nabla^2\bar{\mathcal{L}} - \nabla^2\mathcal{L})(\mathbf{J}\boldsymbol{\theta} - \boldsymbol{\theta}^\star), (\mathbf{W} - \mathbf{J})(\boldsymbol{\theta}' - \boldsymbol{\theta}^\star)\rangle$$

$$\leq \frac{1}{m}\sum_{i=1}^{m}\sum_{j=1}^{m}\frac{|w_{i,j} - 1/m|}{2}|\langle(\nabla^2\bar{\mathcal{L}} - \nabla^2\mathcal{L}_i)(\theta_{\text{av}} - \theta^\star), \theta_{\text{av}} - \theta^\star\rangle|$$

$$+ \frac{1}{m}\sum_{i=1}^{m}\sum_{j=1}^{m}\frac{|w_{i,j} - 1/m|}{2}\left(|\langle(\nabla^2\bar{\mathcal{L}} - \nabla^2\mathcal{L}_i)(\theta_j' - \theta^\star), \theta_j' - \theta^\star\rangle|\right)$$

$$+ \frac{1}{m}\sum_{i=1}^{m}\sum_{j=1}^{m}\frac{|w_{i,j} - 1/m|}{2}\left(|\langle(\nabla^2\bar{\mathcal{L}} - \nabla^2\mathcal{L}_i)(\theta_{\text{av}} - \theta_j'), \theta_{\text{av}} - \theta_j'\rangle|\right).$$

Further $\||W - J\||_\infty \leq \rho\sqrt{m}$ [28] and therefore

$$\langle(\nabla^2\bar{\mathcal{L}} - \nabla^2\mathcal{L})(\mathbf{J}\boldsymbol{\theta} - \boldsymbol{\theta}^\star), (\mathbf{W} - \mathbf{J})(\boldsymbol{\theta}' - \boldsymbol{\theta}^\star)\rangle \leq \frac{\rho\sqrt{m}}{2m}\sum_{i=1}^{m}\sum_{j=1}^{m}|\langle(\nabla^2\bar{\mathcal{L}} - \nabla^2\mathcal{L}_i)(\theta_{\text{av}} - \theta^\star), \theta_{\text{av}} - \theta^\star\rangle|$$

$$+ \frac{\rho\sqrt{m}}{2m}\sum_{i=1}^{m}\sum_{j=1}^{m}\left(|\langle(\nabla^2\bar{\mathcal{L}} - \nabla^2\mathcal{L}_i)(\theta_j' - \theta^\star), \theta_j' - \theta^\star\rangle| + |\langle(\nabla^2\bar{\mathcal{L}} - \nabla^2\mathcal{L}_i)(\theta_j' - \theta_{\text{av}}), \theta_j' - \theta_{\text{av}}\rangle|\right).$$

From Assumption 4 it follows that

$$\langle(\nabla^2\bar{\mathcal{L}} - \nabla^2\mathcal{L})(\mathbf{J}\boldsymbol{\theta} - \boldsymbol{\theta}^\star), (\mathbf{W} - \mathbf{J})(\boldsymbol{\theta}' - \boldsymbol{\theta}^\star)\rangle \leq \frac{\rho\sqrt{m}}{2m}\left(\sum_{i=1}^{m}\sum_{j=1}^{m}\gamma_\ell\left(\|\theta_{\text{av}} - \theta^\star\|_{\Sigma'}^2 + \|\theta_j' - \theta^\star\|_{\Sigma'}^2\right)\right)$$

$$+ \frac{\rho\sqrt{m}}{2m}\sum_{i=1}^{m}\sum_{j=1}^{m}\gamma_\ell\|\theta_{\text{av}} - \theta_j'\|_{\Sigma'}^2 + \frac{\rho\sqrt{m}}{2m}\sum_{i=1}^{m}\sum_{j=1}^{m}\tau_\ell\left(\mathcal{R}^2(\theta_{\text{av}} - \theta^\star) + \mathcal{R}^2(\theta_j' - \theta^\star) + \mathcal{R}^2(\theta_j' - \theta_{\text{av}})\right).$$

Using the triangle and Young's inequalities on both $\|\theta_{\text{av}} - \theta_j'\|_{\Sigma'}^2$ and $\mathcal{R}^2(\theta_{\text{av}} - \theta_j')$, $j \in [m]$, yields

$$\langle(\nabla^2\bar{\mathcal{L}} - \nabla^2\mathcal{L})(\mathbf{J}\boldsymbol{\theta} - \boldsymbol{\theta}^\star), (\mathbf{W} - \mathbf{J})(\boldsymbol{\theta}' - \boldsymbol{\theta}^\star)\rangle \leq \frac{3\rho\sqrt{m}}{2}\gamma_\ell\left(\|\mathbf{J}\boldsymbol{\theta} - \boldsymbol{\theta}^\star\|_{\Sigma'}^2 + \|\boldsymbol{\theta}' - \boldsymbol{\theta}^\star\|_{\Sigma}^2\right)$$

$$+ \frac{3\rho\sqrt{m}}{2}\sum_{j=1}^{m}\tau_\ell\left(\mathcal{R}^2(\theta_{\text{av}} - \theta^\star) + \mathcal{R}^2(\theta_j' - \theta^\star)\right).$$

The next bound can be written as

$$\langle(\nabla^2\bar{\mathcal{L}} - \nabla^2\mathcal{L})(\mathbf{W} - \mathbf{J})(\boldsymbol{\theta} - \boldsymbol{\theta}^\star), (\mathbf{W} - \mathbf{J})(\boldsymbol{\theta}' - \boldsymbol{\theta}^\star)\rangle$$

$$= \frac{1}{m}\sum_{i=1}^{m}\left\langle(\nabla^2\bar{\mathcal{L}} - \nabla^2\mathcal{L}_i)\left(\sum_{j=1}^{m}(w_{i,j} - 1/m)(\theta_j - \theta^\star)\right), \left(\sum_{l=1}^{m}(w_{i,l} - 1/m)(\theta_l' - \theta^\star)\right)\right\rangle,$$

where following the same procedure as for the previous bound we obtain the desired result with the main difference that an additional network term and additional summation is required. $\qquad\square$

## F  Empirical-Population Hessian Deviation Bounds

We provide here some technical lemmata that are necessary to establish that Assumptions 3 and 4 hold with high probability, for the considered data generation models. Specifically, Lemma 1 is used to establish Corollaries 1 and 1 while Lemma 5 and Lemma 6 permit to establish Corollaries 3 and 2,respectively. Since these lemmata are minor modifications of existing results, we omit their proofs.

**Proof of Lemma 1**

The global parameters $\gamma_{g,1}$ $\gamma_{g,2}$ $\tau_g$ and $\Sigma$ follow from Proposition 1 in [1]. Establishing the local quantities $\gamma_{\ell,1}$, $\gamma_{\ell,2}$ and $\tau_\ell$ follows the same steps as the proof of Proposition 5 in [28]. Observe that while the proof in [28] is done for the particular case in which $\mathcal{R}(\cdot) = \|\cdot\|_1$ it can be extended without significant differences to more general decomposable regularizers $\mathcal{R}$.

**Sub-Gaussian data generation model (c.f. Assumption 5(ii))**

For Corollary 3 to hold under the sub-Gaussian data generation model we require that Assumptions 3 and 4 hold. We establish that this is the case with overwhelming probability in the following Lemma.

**Lemma 5** (Sub-Gaussian data model and $\mathcal{R}(\cdot) = \|\cdot\|_1$). *Consider the problem* (1) *and assume that Assumptions 1 and 5(ii) hold. Denote by $X = [x_1^\top; \ldots; x_N^\top]$ and $X_{S_i} = [x_{j_1}^\top; \ldots; x_{j_n}^\top]$, $j_l \in S_i$, $l \in [n]$, $i \in [m]$. Then, there exists universal constants $c_0, c_1, c_2 > 0$ such that given that*

$$N \geq c_0 \log d \max\left\{1, \frac{\tau^4}{\mu^2}\right\}$$

*there exist universal constants $c_0$ and $c_1$ such that with probability at least*

$$1 - 2\exp\left(-c_1 N \min\left\{\frac{\mu^2}{\tau^4}, 1\right\} + \log(m+1)\right)$$

*it holds*

$$\left|\left\langle v, \left(\Sigma - \frac{X^\top X}{N}\right) v\right\rangle\right| \leq \frac{\mu}{2}\|v\|_2^2 + c_2 \frac{\log d}{N} \max\left\{\frac{\tau^4}{\mu}, \mu\right\}\|v\|_1^2, \ \forall v \in \mathbb{R}^d \tag{46}$$

$$\left|\left\langle v_i, \left(\Sigma - \frac{X_{S_i}^\top X_{S_i}}{n}\right) v_i\right\rangle\right| \leq \frac{m\mu}{2}\|v_i\|_2^2 + c_2 \frac{\log d}{n} \max\left\{\frac{\tau^4}{\mu}, \mu\right\}\|v_i\|_1^2, \ \forall v_i \in \mathbb{R}^d, \ i \in [m]. \tag{47}$$

*Proof.* The inequality (46) is a restatement of Corollary 1 in [15]. To obtain (47) similar steps as in [28] are taken on Lemma 12 in [15] followed by a union bound argument. $\square$

**Uniform data generation model (c.f. Assumption 5(iii))**

For Corollary 2 to hold under the uniform data generation model we require that a variation of Assumptions 3 and 4 hold–see Lemma 8 and 9 in [1] for a discussion on this matter. We provide next a technical result required to establish a variant of Assumptions 3 and 4 tailored to this statistical model. We refer the reader to the proof of Corollary 2 (c.f. Sec. G) to see how this result is used.

**Lemma 6.** *Let*

$$X^{(i)} = d\, s^{(i)} e_{a(i)} e_{b(i)}^\top \in \mathbb{R}^{p \times p}, \quad i \in [N],$$

*where $p^2 = d$, $s^{(i)} \in \{-1, 1\}$ uniformly, and are i.i.d. Further, $a(i), b(i) \sim \mathcal{U}\{1, p\}$ are i.i.d. and independent of $s^{(i)}$. For convenience, denote by*

$$X \triangleq \begin{pmatrix} \mathrm{vec}(X^{(1)})^\top \\ \vdots \\ \mathrm{vec}(X^{(m)})^\top \end{pmatrix}, \qquad X_{S_i} \triangleq \begin{pmatrix} \mathrm{vec}(X^{(j_1)})^\top \\ \vdots \\ \mathrm{vec}(X^{j_m})^\top \end{pmatrix},$$

*where $j_l \in S_i$, $l \in [n]$, and $i \in [m]$. Then, there exist universal constants $c_i > 0$, $i \in \{0, \ldots, 5\}$, such that, given*

$$N \geq c_0 p \log p,$$

*with probability at least*

$$1 - \exp\left(-p\log(p)\right) - c_1 m \exp(-p\log(p)),$$

*it holds for all $V \in \mathbb{R}^{p \times p}$ and $V_i \in \mathbb{R}^{p \times p}$, $i \in [m]$,*

$$\left|\left\langle \mathrm{vec}(V), \left(\frac{XX^\top}{N} - I_{p^2}\right) \mathrm{vec}(V)\right\rangle\right| \leq c_2 p \|V\|_\infty \|V\|_1 \sqrt{\frac{p\log p}{N}} + c_3 \left(p\|V\|_\infty \sqrt{\frac{p\log p}{N}}\right)^2, \tag{48}$$

$$\left|\left\langle \mathrm{vec}(V_i), \left(\frac{X_{S_i} X_{S_i}^\top}{n} - I_{p^2}\right) \mathrm{vec}(V_i)\right\rangle\right| \leq c_4 pm \|V_i\|_\infty \|V\|_1 \sqrt{\frac{p\log p}{n}} + c_5 \left(pm\|V\|_\infty \sqrt{\frac{p\log p}{n}}\right)^2. \tag{49}$$

*Proof.* The statement in (48) restates Proposition 2 in [1] and (49) corresponds to following a very similar process as that in the proof of Proposition 5 in [28] but applied to Theorem 1 in [20] which is a restatement of (48). □

# G  Proofs of Corollaries 1-4

## G.1  Proof of Corollary 1

We establish that for the given statistical model, the parameters are

$$\gamma_{g,1} = \frac{1}{2}, \qquad\qquad \gamma_{g,2} = 1$$
$$\gamma_{\ell,1} = 4m - 1 \qquad\qquad \gamma_{\ell,1} = 1,$$
$$\tau_g = \frac{9\eta_\Sigma c_1 \log d}{N}, \qquad\qquad \tau_\ell = 9\eta_\Sigma \frac{\log d}{n},$$
$$\Sigma' = \Sigma = \nabla^2 \bar{\mathcal{L}}, \qquad\qquad \Psi^2(\bar{\mathcal{M}}) = \sqrt{s},$$
$$\mathcal{R}^*\left(\nabla\mathcal{L}(\theta^\star)\right) \le \sqrt{3\eta_\Sigma \sigma^2 \frac{\log d}{N}}, \qquad \max_{j\in[n]} \mathcal{R}^*\left(\nabla\mathcal{L}_i(\theta^\star)\right) = \sqrt{3m\eta_\Sigma \sigma^2 \frac{\log(md)}{N}},$$

and $\Pi_{\mathcal{M}^\perp}(\theta^\star) = 0$, for $q = 0$, where the suitable choice of sub-spaces is given in [2]. In the case $q \in (0, 1]$ if $\|\theta^\star\|_q \le R_q$, for appropriate choice of $\bar{\mathcal{M}}$ [2]

$$\Psi^2(\bar{\mathcal{M}}) \le \left(\frac{\log d}{N}\right)^{-\frac{q}{2}} R_q,$$

$$\|\Pi_{\mathcal{M}^\perp}(\theta^\star)\|_1 \le \left(\frac{\log d}{N}\right)^{\frac{1-q}{2}} R_q.$$

All statements except the bounds on the dual norms of the gradient(s) follow with high probability from Lemma 1. The bounds of the dual norms of the gradient(s) can be established by combining results in [30, Ch. 7] and the bound on the largest of the dual norms can be established via a minor alteration of the proof of [13, Th. 4]. Taking the union bound over all events guarantees that they all hold simultaneously with high probability. The result then follows by applying Theorem 2.

## G.2  Proof of Corollary 2

Consider the problem of recovering $\Theta^\star \in \mathbb{R}^{p\times p}$ from

$$\tilde{y}_i = \Theta^\star_{r(i),c(i)} + \frac{\sigma}{d}\tilde{w}_i,$$

where $d = p^2$, and $r(i)$ and $c(i)$ are such that $(\text{vec}(\Theta^\star))_{v(i)} = \Theta^\star_{r(i),c(i)}$, and where $v(i) \sim \mathcal{U}\{1, d\}$, and are i.i.d. Further $\tilde{w}_i$ is zero mean, symmetric, i.i.d. and independent of $v(i)$. The problem above is statistically equivalent to recovering $\Theta^\star \in \mathbb{R}^p$ from [20]

$$y_i = \langle X_i, \Theta^\star\rangle + \sigma w_i,$$

where

$$X_i = ps_i e_{r(i)} e_{c(i)}^\top,$$

where $w_i$ follows the same distribution as $\tilde{w}_i$, and $s_i \in \{-1, 1\}$ uniformly and i.i.d. Thus, the problem formulation corresponds to

$$\min_{\Theta:\|\Theta\|_\infty \le \frac{\alpha}{p}, \|\Theta\|_1 \le \rho} \underbrace{\frac{1}{2N}\sum_{i=1}^N \left(\langle X_i, \Theta^\star\rangle - y_i^2\right)}_{\triangleq \mathcal{L}(\Theta)} = \underbrace{\frac{1}{2N}\sum_{i=1}^N \left(\text{vec}(X_i)^\top \text{vec}(\Theta^\star) - y_i\right)^2}_{\triangleq \mathcal{L}(\theta)}.$$

The population risk is given by

$$\|\Sigma^{1/2}(\text{vec}(\Theta) - \text{vec}(\Theta^\star))\|_2^2.$$

Observe that by setting $\mathcal{R}(\cdot) = \| \cdot \|_1$ we have that $\mathcal{R}^*(\cdot) = \|\|\cdot\|\|_2$. Under the Assumption that $\tilde{w}_i$ is sub-exponential with parameter one we establish bounds

$$\|\nabla\mathcal{L}(\Theta^\star)\|_2 = \frac{\left\|\left\|\sum_{i=1}^N X_i \sigma w_i\right\|\right\|_2}{N},$$

$$\|\nabla\mathcal{L}_i(\Theta^\star)\|_2 = \frac{\left\|\left\|\sum_{j \in S_i} X_j \sigma w_j\right\|\right\|_2}{n}.$$

Using Lemma 7 in [20] and using the union bound it follows that for some $c_1$, $c_2$, $c_3 > 0$

with probability at least

$$1 - c_2(m+1)\exp\left(-c_3 \log(p)\right),$$

it holds that

$$\mathcal{R}^*\left(\nabla\mathcal{L}(\Theta^\star)\right) \leq c_1 \sigma \sqrt{\frac{p \log p}{N}},$$

$$\mathcal{R}^*\left(\sum_{j=1}^m (w_{i,j} - 1/m)\nabla\mathcal{L}_j(\Theta^\star)\right)$$

$$\leq \sum_{j=1}^m |w_{i,j} - 1/m|\mathcal{R}^*\left(\nabla\mathcal{L}_j(\theta^\star)\right) \leq \sum_{j=1}^m \|\|W - J\|\|_\infty m c_1 \sigma \sqrt{\frac{p \log p}{N}} \leq m^2 \sqrt{m}\rho c_1 \sigma \sqrt{\frac{p \log p}{N}}.$$

As highlighted in the discussion prior to Lemma 6 and by observing the statement in Lemma 6 itself, Assumptions 3 and 4 are not fulfilled exactly. This is consistent with the results in [1] (observe that in this work the authors also follow a slightly modified procedure tailored to this statistical model). Consequently, we derive variants of Lemmas 3 and 4 under the bounds established via Lemma 6. Observe the clear resemblance in procedure when compared to the proof of Lemmas 3 and 4. After this, the statement can be established by following the steps to establish Theorem 2 with very minor modifications.

In order to obtain a usable bound that is similar in structure to those in Assumption 3 and 4 we employ Lemma 6 as follows. Let $X \triangleq (\text{vec}(X_1), \ldots, \text{vec}(X_N))^\top$. Then, using Lemma 6 we have that for any pair $\Theta^{(1)} \in \mathbb{R}^{p \times p}$ and $\Theta^{(2)} \in \mathbb{R}^{p \times p}$ such that $\|\Theta_i\|_\infty \leq \frac{\omega}{p}$ $i \in [2]$ it holds that for some $c_4$, $c_5 > 0$

$$\left|\langle\text{vec}(\Theta^{(1)} - \Theta^{(2)}), \left(\frac{X^\top X}{N} - I_{p^2}\right)\text{vec}(\Theta^{(1)} - \Theta^{(2)})\rangle\right| \tag{50}$$

$$\leq 2\omega c_4 \|\Theta^{(1)} - \Theta^{(2)}\|_1 \sqrt{\frac{p \log p}{N}} + 4\omega^2 c_5 \frac{p \log p}{N}. \tag{51}$$

As discussed previously, we now use the result above to obtain a parallel to Lemma 3. For convenience, denote by $\boldsymbol{\theta} \in \mathbb{R}^{mp^2}$ such that $\theta_i = \text{vec}(\Theta_i)$ and $\boldsymbol{\theta} = [\theta_1^\top, \ldots, \theta_m^\top]^\top$, and analogously for $\boldsymbol{\theta}'$. Assume all $\Theta_i$ and $\Theta_i'$ are feasible. In this way, we have following the same steps as in the proof of Lemma 3

$$\langle\mathbf{J}(\boldsymbol{\theta} - \theta^\star), \left(\nabla^2\bar{\mathcal{L}} - \nabla^2\bar{\mathcal{L}}\right)(\mathbf{J}\boldsymbol{\theta}' - \theta^\star)\rangle = \frac{1}{2}\left\langle\theta_{\text{av}} - \theta^\star, \left(\frac{X^\top X}{N} - I_{p^2}\right)(\theta_{\text{av}} - \theta^\star)\right\rangle$$

$$+ \frac{1}{2}\left\langle\theta_{\text{av}}' - \theta^\star, \left(\frac{X^\top X}{N} - I_{p^2}\right)(\theta_{\text{av}}' - \theta^\star)\right\rangle - \frac{1}{2}\left\langle\theta_{\text{av}} - \theta_{\text{av}}', \left(\frac{X^\top X}{N} - I_{p^2}\right)(\theta_{\text{av}} - \theta_{\text{av}}')\right\rangle$$

$$\leq \sqrt{\frac{p \log p}{N}}\omega c_4\left(\|\Theta_{\text{av}}' - \Theta^\star\|_1 + \|\Theta_{\text{av}} - \Theta^\star\|_1 + \|\Theta_{\text{av}}' - \Theta_{\text{av}}\|_1\right) + 12\omega^2 c_5 \frac{p \log p}{N}.$$

where we have used (50) to obtain the last inequality, $\theta_{\text{av}} \triangleq \frac{1}{m}\sum_{i=1}^m \theta_i$, $\Theta_{\text{av}} \triangleq \frac{1}{m}\sum_{i=1}^m \Theta_i$ and analogously for $\theta_{\text{av}}'$ and $\Theta_{\text{av}}'$. Because $\Theta$ and $\Theta'$ are feasible, so are $\Theta_{\text{av}}$ and $\Theta_{\text{av}}'$. By using the

triangle inequality on $\|\Theta_{\mathrm{av}} - \Theta'_{\mathrm{av}}\|_1$, and invoking Lemma 1 in [2] we obtain

$$\langle \mathbf{J}(\boldsymbol{\theta} - \boldsymbol{\theta}^\star), (\nabla^2 \bar{\mathcal{L}} - \nabla^2 \mathcal{L})(\mathbf{J}\boldsymbol{\theta}' - \boldsymbol{\theta}^\star)\rangle \leq \omega c_4 \sqrt{\frac{p \log p}{N}} 12 \|\Pi_{\mathcal{M}^\perp}(\Theta^\star)\|_1$$

$$+ 4\Psi(\bar{\mathcal{M}})\sqrt{\frac{p \log p}{N}}\omega c_4 \left(\|\theta'_{\mathrm{av}} - \theta^\star\|_2 + \|\theta_{\mathrm{av}} - \theta^\star\|_2\right) + 12\omega^2 c_5 \frac{p \log p}{N},$$

whereby using Young's inequality we finally obtain

$$\langle \mathbf{J}(\boldsymbol{\theta} - \boldsymbol{\theta}^\star), (\nabla^2 \bar{\mathcal{L}} - \nabla^2 \mathcal{L})(\mathbf{J}\boldsymbol{\theta}' - \boldsymbol{\theta}^\star)\rangle$$

$$\leq \frac{1}{6}\left(\|\theta_{\mathrm{av}} - \theta^\star\|_2^2 + \|\theta'_{\mathrm{av}} - \theta^\star\|_2^2\right) + 12\omega c_4 \sqrt{\frac{p \log p}{N}}\|\Pi_{\mathcal{M}^\perp}(\Theta^\star)\|_1$$

$$+ 12\omega^2 c_5 \frac{p \log p}{N} + 24\omega^2 c_4^2 \Psi^2(\bar{\mathcal{M}})\frac{p \log p}{N} = \frac{1}{6m}\left(\|\mathbf{J}\boldsymbol{\theta} - \boldsymbol{\theta}^\star\|_2^2 + \|\mathbf{J}\boldsymbol{\theta}' - \boldsymbol{\theta}^\star\|_2^2\right)$$

$$+ \omega^2 \frac{p \log p}{N}\left(12c_5 + 24c_4^2 \Psi^2(\bar{\mathcal{M}})\right) + 12\omega c_4 \sqrt{\frac{p \log p}{N}}\|\Pi_{\mathcal{M}^\perp}(\Theta^\star)\|_1.$$

The above can be applied in place of Lemma 3 in Theorem 2 to establish this corollary. Observe that for DGD$^2$ not only is the inequality above important, but, due to the presence of the network, we are to obtain similar inequalities as those in Lemma 4. This follows an almost identical procedure as that in the proof of Lemma 4 with a few minor modifications that we now highlight. Following the same steps as in the proof of Lemma 4 we have

$$\langle (\mathbf{W} - \mathbf{J})(\boldsymbol{\theta}' - \boldsymbol{\theta}^\star), (\nabla^2 \bar{\mathcal{L}} - \nabla^2 \mathcal{L})(\mathbf{J}\boldsymbol{\theta} - \boldsymbol{\theta}^\star)\rangle \leq$$

$$\frac{\rho\sqrt{m}}{2m}\sum_{i=1}^m \sum_{j=1}^m \left|\langle (\nabla^2 \bar{\mathcal{L}} - \nabla^2 \mathcal{L})(\theta_{\mathrm{av}} - \theta^\star), \theta_{\mathrm{av}} - \theta^\star\rangle\right|$$

$$+ \frac{\rho\sqrt{m}}{2m}\sum_{i=1}^m \sum_{j=1}^m \left(\left|\langle (\nabla^2 \bar{\mathcal{L}} - \nabla^2 \mathcal{L})(\theta'_j - \theta^\star), \theta'_j - \theta^\star\rangle\right| + \left|\langle (\nabla^2 \bar{\mathcal{L}} - \nabla^2 \mathcal{L}_i)(\theta'_j - \theta_{\mathrm{av}}), \theta'_j - \theta_{\mathrm{av}}\rangle\right|\right)$$

$$\leq \frac{\rho\sqrt{m}}{2m}\sum_{i=1}^m \sum_{j=1}^m \left(c_6 pm\|\Theta_{\mathrm{av}} - \Theta^\star\|_\infty \|\Theta_{\mathrm{av}} - \Theta^\star\|_1 \sqrt{\frac{p \log p}{n}} + \left(c_7 p^2 m^2 \|\Theta_{\mathrm{av}} - \Theta^\star\|_\infty^2 \frac{p \log p}{n}\right)\right)$$

$$+ \frac{\rho\sqrt{m}}{2}\sum_{i=1}^m \sum_{j=1}^m \left(c_6 pm\|\Theta'_j - \Theta^\star\|_\infty \|\Theta'_j - \Theta^\star\|_1 \sqrt{\frac{p \log p}{n}} + \left(c_7 p^2 m^2 \|\Theta'_j - \Theta^\star\|_\infty^2 \frac{p \log p}{n}\right)\right)$$

$$+ \frac{\rho\sqrt{m}}{2}\sum_{i=1}^m \sum_{j=1}^m \left(c_6 pm\|\Theta_{\mathrm{av}} - \Theta'_j\|_\infty \|\Theta_{\mathrm{av}} - \Theta'_j\|_1 \sqrt{\frac{p \log p}{n}} + \left(c_7 p^2 m^2 \|\Theta_{\mathrm{av}} - \Theta'_j\|_\infty^2 \frac{p \log p}{n}\right)\right),$$

where the last inequality follows from Lemma 6. Observe that because $\Theta_{\mathrm{av}}$, $\Theta^\star$ and $\Theta'_j$ are all feasible it holds that the $\|\cdot\|_\infty$ can all be upper bounded by $\frac{2\omega}{p}$. Further, we use the triangle inequality on $\|\Theta_{\mathrm{av}} - \Theta'_j\|_1$ to yield

$$\langle (\mathbf{W} - \mathbf{J})(\boldsymbol{\theta}' - \boldsymbol{\theta}^\star), (\nabla^2 \bar{\mathcal{L}} - \nabla^2 \mathcal{L})(\mathbf{J}\boldsymbol{\theta} - \boldsymbol{\theta}^\star)\rangle \leq$$

$$c_6 \rho\sqrt{m}\sum_{i=1}^m \sum_{j=1}^m \left(2\omega\|\Theta_{\mathrm{av}} - \Theta^\star\|_1 + 2\omega m\|\Theta'_j - \Theta^\star\|_1\right)\sqrt{\frac{p \log p}{n}} + \frac{3\rho\sqrt{m}}{2}\left(\omega^2 4c_7 m^3 \frac{p \log p}{n}\right).$$

Due to feasibility of $\Theta_{\mathrm{av}}$ and $\Theta'_j$ we invoke Lemma 1 in [2] to obtain

$$\langle (\mathbf{W} - \mathbf{J})(\boldsymbol{\theta}' - \boldsymbol{\theta}^\star), (\nabla^2 \bar{\mathcal{L}} - \nabla^2 \mathcal{L})(\mathbf{J}\boldsymbol{\theta} - \boldsymbol{\theta}^\star)\rangle$$

$$\leq c_6 \rho\sqrt{m}4\Psi(\bar{\mathcal{M}})\omega m \sqrt{\frac{p \log p}{n}}\sum_{j=1}^m \left(\|\theta_{\mathrm{av}} - \theta^\star\|_2 + \|\theta'_j - \theta^\star\|_2\right)$$

$$+ c_7 \rho\sqrt{m}4\omega m^2 \sqrt{\frac{p \log p}{n}}\|\Pi_{\mathcal{M}^\perp}(\Theta^\star)\|_1 + \frac{3\rho\omega^2\sqrt{m}m^3}{2}4c_7 \frac{p \log p}{n}.$$

Using Young's inequality yields

$$\langle (\mathbf{W} - \mathbf{J})(\boldsymbol{\theta}' - \boldsymbol{\theta}^\star), (\nabla^2 \bar{\mathcal{L}} - \nabla^2 \mathcal{L})(\mathbf{J}\boldsymbol{\theta} - \boldsymbol{\theta}^\star) \rangle \leq c_5 \frac{\rho \sqrt{m} m}{2} \sum_{j=1}^{m} \left( \|\theta_{\mathrm{av}} - \theta^\star\|_2^2 + \|\theta_j' - \theta^\star\|_2^2 \right)$$

$$+ c_6 \frac{\rho \sqrt{m} m^2}{2} 16 \Psi^2(\bar{\mathcal{M}}) \omega^2 \frac{p \log p}{n} + 4 c_6 \omega \sqrt{m} m^3 \rho \sqrt{\frac{p \log p}{n}} \|\Pi_{\mathcal{M}^\perp}(\Theta^\star)\|_1 + 12 c_7 \rho \sqrt{m} m^3 \omega^2 \frac{p \log p}{n}$$

$$= c_6 \frac{\rho \sqrt{m} m}{2} \left( \|\mathbf{J}\boldsymbol{\theta} - \boldsymbol{\theta}^\star\|_2^2 + \|\boldsymbol{\theta}' - \boldsymbol{\theta}^\star\|_2^2 \right) + 4 c_6 \omega m^3 \sqrt{m} \rho \sqrt{\frac{p \log p}{n}} \|\Pi_{\mathcal{M}^\perp}(\Theta^\star)\|_1$$

$$+ \left( c_6 8 \Psi^2(\bar{\mathcal{M}}) + 12 m c_6 \right) \rho \omega^2 \sqrt{m} m^2 \frac{p \log p}{n}.$$

We have left to find bounds on $\Psi^2(\bar{\mathcal{M}})$ and $\|\Pi_{\mathcal{M}^\perp}(\Theta^\star)\|_1$. Under the conditions of the Corollary we have from [2] (subspace suitably defined by combining elements of the proof of Corollaries 4 and 5 in [2]) that

$$\Psi^2(\mathcal{M}) \leq R_q \omega^{-q} \left( \frac{p \log p}{N} \right)^{-\frac{q}{2}},$$

$$\|\Pi_{\mathcal{M}^\perp}(\Theta^\star)\|_1 \leq R_q \omega^{1-q} \left( \frac{p \log p}{N} \right)^{\frac{1-q}{2}}.$$

Observe that in [2] $\Psi^2(\bar{\mathcal{M}})$ is computed with respect to the Frobenius norm $\| \cdot \|_{\mathrm{F}}$ however, observe that for any $\Theta \in \mathbb{R}^{p \times p}$ it holds that $\|\Theta\|_{\mathrm{F}} = \|\mathrm{vec}(\Theta)\|_2$.

By deriving similar bounds for the remaining quantities in Lemma 4 and following the same steps as in the proof of Theorem 2 and treating the extra tolerance in the same way as the misspecification error in the proof of Theorem 2 the desired result follows yielding

$$r_t^2 \leq \lambda^t r_0^2 + \mathcal{O} \left( \left( \frac{p \log p}{N} \right)^{1 - \frac{q}{2}} R_q \left( \sigma^2 \omega^{-q} + \omega^{2-q} + \sigma \omega^{1-q} \right) \left( 1 + \rho m^5 \right) \right).$$

where the latter term stems from the tolerances discussed throughout this proof. Observe that in this particular case $\zeta(\bar{\mathcal{M}}, W)$ (c.f. Theorem 2) is equal to zero and that the dominating term is the latter since $\omega \geq 1$.

### G.3   Proof of Corollary 3

Under the column normalized assumption, following the same steps as in the proof of Corollary 1 it follows that

$$\mathbb{P} \left( \max_{i \in [m]} \frac{\|X_{S_i}^\top w_{S_i}\|_\infty}{n} \leq C \sigma \sqrt{\frac{2m \log(md)}{N}}, \frac{\|X^\top w\|_\infty}{N} \leq C \sigma \sqrt{\frac{2 \log d}{N}} \right)$$

$$\geq 1 - 2d \exp(-2 \log(d)) - 2md \exp(-2 \log(dm))).$$

From Lemma 5 it follows that Assumptions 3 and 4 hold with high probability. Finally bounds for $\Psi^2(\bar{\mathcal{M}})$ and $\mathcal{R}(\Pi_{\mathcal{M}^\perp}(\theta^\star))$ as in the proof of Corollary 1 apply. Thus, taking the overall union bound and invoking Theorem 1 yields the desired result.

### G.4   Proof of Corollary 4

Observe that the empirical risk minimization problem can be written as

$$\min_{\Theta : \|\Theta\|_1 \leq \bar{\rho}} \underbrace{\frac{1}{2N} \sum_{i=1}^{N} \left( \langle X_i, \Theta^\star \rangle - y_i \right)^2}_{\triangleq \mathcal{L}(\Theta)} = \underbrace{\frac{1}{2N} \sum_{i=1}^{N} \left( \mathrm{vec}(X_i)^\top \mathrm{vec}(\Theta^\star) - y_i \right)^2}_{\triangleq \mathcal{L}(\theta)}.$$

The population risk is given by

$$\bar{\mathcal{L}}(\theta) \triangleq \|\Sigma^{1/2}(\theta - \theta^\star)\|_2^2,$$

where $\theta^\star = \text{vec}(\Theta^\star)$. Then, from Lemma 1 it follows that Assumptions 3 and 4 hold with the probability given in Lemma 1. Observe that $\mathbb{E}[\mathcal{R}^*(\text{vec}(x_i))] \leq c\eta_\Sigma \frac{p}{n}$, which establishes the terms accompanying $\mathcal{R}(\cdot) = \|\cdot\|_1$.

In this case, the statement in Lemma 1 in [2] can be written as

$$\|\Theta - \Theta^\star\|_1 \leq 2\Psi(\bar{\mathcal{M}})\|\Theta - \Theta^\star\|_F + 2\|\Pi_{\mathcal{M}^\perp}(\Theta^\star)\|_1.$$

As in [2] for $q = 0$ and $q \in (0, 1]$ we have (see [2] for the appropriate choices of $\mathcal{M}$, $\bar{\mathcal{M}}$).

$$\Psi^2(\bar{\mathcal{M}}) = r = R_0, \qquad\qquad \Psi^2(\bar{\mathcal{M}}) \leq R_q \frac{p}{N}^{-\frac{q}{2}},$$

$$\|\Pi_{\mathcal{M}^\perp}(\Theta^\star)\|_1 = 0, \qquad\qquad \|\Pi_{\mathcal{M}^\perp}(\Theta^\star)\|_1 \leq R_q \left(\frac{p}{N}\right)^{\frac{1-q}{2}},$$

where $r$ denotes the rank of $\Theta^\star$, the left column corresponds to $q = 0$ and the right column to $q \in (0, 1]$.

Thus, we have left to establish upper bounds on

$$\frac{\|\sum_{i=1}^m X_i w_i\|_2}{N}, \qquad\qquad \frac{\|\sum_{j \in S_i} X_j w_j\|_2}{n}.$$

Following the proof of [30, Cor. 10.10] it can be established that for some universal constants $c_0$, $c_1$, $c_2 > 0$ if

$$N \geq c_0 p$$

it holds that

$$\mathbb{P}\left(\cap_{i \in [m]}\left(\frac{\|\sum_{j \in S_i} X_j w_j\|_2}{N} \leq c_1 \sigma \eta_\Sigma^{1/2} m \sqrt{\frac{p}{N}}\right) \cap \left(\frac{\|\sum_{i=1}^m X_i w_i\|_2}{N} \leq c_1 \sigma \eta_\Sigma^{1/2} \sqrt{\frac{p}{N}}\right)\right)$$
$$\geq 1 - (m+1)\left(\exp(-c_2 p)\right).$$

Invoking Lemma 1 and using the union bound yields the desired result.

## H  DGD Algorithm (4) for Gaussian $\ell_1$-sparse regression

Recall that the DGD-CTA iterates can be written as

$$\theta^{t+1} = \prod_{\Omega \cap \mathcal{R}(\theta) \leq R}\left(\mathbf{W}\theta^t - \alpha m \nabla \mathcal{L}(\theta^t)\right). \tag{52}$$

We establish convergence of DGD-CTA for the particular case of $\ell_1$ constrained Gaussian linear regression. We begin by introducing some notation. Denote by

$$\Delta_{\text{stat}}^2 \triangleq \frac{128\Psi^2(\bar{\mathcal{M}})}{\mu^2}\left(\mathcal{R}^*(\nabla \mathcal{L}(\theta^\star))\right)^2 + \frac{16}{\mu}\mathcal{R}(\Pi_{\mathcal{M}^\perp}(\theta^\star))\mathcal{R}(\nabla \mathcal{L}(\theta^\star)),$$

$$\Delta_{\text{net}}^2 \triangleq \frac{16m\alpha}{(1-\rho)\mu}\|\nabla \mathcal{L}(\theta^\star)\|_2^2,$$

and the tolerance

$$\Delta^2 \triangleq \left(\Delta_{\text{stat}}^2 + \Delta_{\text{net}}^2\right)\left(1 + \frac{\alpha\zeta(\bar{\mathcal{M}})}{1 - \zeta(\bar{\mathcal{M}})}\right) + \frac{144}{\mu\Psi^2(\bar{\mathcal{M}})}\mathcal{R}^2(\Pi_{\mathcal{M}^\perp}(\theta^\star))\zeta(\bar{\mathcal{M}})\left(1 + \frac{\alpha\zeta(\bar{\mathcal{M}})}{1 - \alpha\zeta(\bar{\mathcal{M}})}\right),$$

where

$$\zeta(\bar{\mathcal{M}}) \triangleq 9\Psi^2(\bar{\mathcal{M}})\tau_g.$$

**Theorem 3.** *Consider the ERM problem* (1) *and the associated population minimization* (2). *Given that Assumptions 1-3 hold with $\gamma_{g,1} = \frac{1}{2}$ and $\gamma_{g,2} = 1$, that Assumption 6 holds with $W \succ 0$, and that*

$$\|\nabla^2 \bar{\mathcal{L}} - \nabla^2 \bar{\mathcal{L}}_i\|_2 \leq \eta, \ i \in [m], \tag{53}$$

$$\Sigma' \preceq \nabla^2 \mathcal{L},$$

$$\alpha \leq \min\left\{\frac{\sigma_m(W)}{2L}, \frac{1-\rho}{18\eta + 16L}\right\}, \qquad\qquad \zeta(\bar{\mathcal{M}}) \leq \frac{\mu}{16},$$

*then, the iterates of DGD-CTA* (52) *fulfil*

$$r_t^2 \leq \left(\frac{1 - \alpha\frac{\mu}{4} + \alpha\zeta(\bar{\mathcal{M}})}{1 - \alpha\zeta(\bar{\mathcal{M}})}\right)^t r_0^2 + \Delta^2.$$

Contrasting Theorem 2 and Theorem 3, the following comments are in order. First, in Theorem 2 $\Delta_{\text{net}}^2$ can be made of the same order of $\Delta_{\text{stat}}^2$ leveraging the network to mix local information. On the other hand, in the case of Theorem 3, even if the network is fully connected, i.e. $\rho = 0$ $\Delta_{\text{net}}^2 = \frac{16m\alpha}{\mu}\|\nabla\mathcal{L}(\theta^\star)\|_2^2$, where for linear Gaussian regression, with overwhelming probability $\|\nabla\mathcal{L}(\theta^\star)\|_2^2 = \mathcal{O}(d/n)$. This implies, that the step-size $\alpha$ is to be chosen as $\alpha = \mathcal{O}(d^{-1})$ for $\Delta_{\text{net}}^2$ to be of the same order as $\Delta_{\text{stat}}^2$.

### H.1  Proof of Theorem 3 and auxiliary results

We begin establishing a variant of Lemma 2 for the proxy

$$\mathcal{L}(\theta) + \frac{1}{2\alpha m}\|\theta\|_{I-\mathbf{W}}^2,$$

which can be done following the same procedure as in the proof of Lemma 2 and yields

$$\|\theta^{t+1} - \theta^\star\|^2 \leq \|\theta^t - \theta^\star\|^2 - \alpha m\left(\|\theta^t - \theta^\star\|_{\nabla^2\bar{\mathcal{L}}}^2 + \|\theta^{t+1} - \theta^\star\|_{\nabla^2\bar{\mathcal{L}}}^2\right)$$
$$- \left(\|\theta^t\|_{I-\mathbf{W}}^2 + \|\theta^{t+1}\|_{I-\mathbf{W}}^2\right) + \alpha m\|\theta^{t+1} - \theta^t\|_{\nabla^2\bar{\mathcal{L}}}^2 - \|\theta^t - \theta^{t+1}\|_{\mathbf{W}}^2$$
$$+ 2\alpha m\langle\nabla\bar{\mathcal{L}}(\theta^t) - \nabla\mathcal{L}(\theta^t), \theta^{t+1} - \theta^t\rangle.$$

We split the inner product similarly to (24)-(27). Under Assumption 3 and (53) we have from Lemmas 3 and 7 that

$$2\alpha m\langle\nabla\bar{\mathcal{L}}(\theta^t) - \nabla\mathcal{L}(\theta^t), (\theta^{t+1} - \theta^\star)\rangle \leq (\alpha\gamma_{g,1} + \alpha\epsilon\gamma_{g,1})\left(\|\mathbf{J}\theta^t - \theta^\star\|_{\Sigma'}^2 + \|\mathbf{J}\theta^{t+1} - \theta^\star\|_{\Sigma'}^2\right)$$
$$+ m\alpha\epsilon\left(\|\mathbf{J}\theta^t - \theta^\star\|_{\nabla^2\bar{\mathcal{L}}}^2 + \|\mathbf{J}\theta^{t+1} - \theta^\star\|_{\nabla^2\bar{\mathcal{L}}}^2\right) + \alpha\gamma_{g,2}\|\mathbf{J}\theta^t - \mathbf{J}\theta^{t+1}\|_{\Sigma'}^2$$
$$+ \left(1 + \frac{1}{\epsilon}\right)\alpha\eta\left(\|\theta^t - \mathbf{J}\theta^t\|_2^2 + \|\theta^{t+1} - \mathbf{J}\theta^{t+1}\|_2^2\right) + \frac{\alpha m}{\epsilon}\left(\|\theta^t - \mathbf{J}\theta\|_{\nabla^2\bar{\mathcal{L}}}^2 + \|\theta^t - \mathbf{J}\theta^t\|_{\nabla^2\bar{\mathcal{L}}}^2\right)$$
$$+ \alpha m\tau_g(1+\epsilon)\left(\mathcal{R}^2(\theta_{\text{av}}^t - \theta^\star) + \mathcal{R}^2(\theta_{\text{av}}^{t+1} - \theta^\star)\right) - 2\alpha m\langle\nabla\mathcal{L}(\theta^\star), \theta^{t+1} - \theta^\star\rangle.$$

Using [2, Lemma 1], we have

$$\$\mathcal{R}^2(\theta_{\text{av}}^t - \theta^\star) \leq 4(1+a)\Psi^2(\bar{\mathcal{M}})\|\theta_{\text{av}}^t - \theta^\star\|_2^2 + 4\left(1 + \frac{1}{a}\right)\mathcal{R}^2(\Pi_{\mathcal{M}^\perp}(\theta^\star)),$$

and analogously for $\theta_{\text{av}}^{t+1}$, implying that

$$\alpha m\tau_g(1+\epsilon)\left(\mathcal{R}^2(\theta_{\text{av}}^t - \theta^\star) + \mathcal{R}^2(\theta_{\text{av}}^{t+1} - \theta^\star)\right) \leq$$
$$\alpha m\tau_g(1+\epsilon)4(1+a)\Psi^2(\bar{\mathcal{M}})\left(\|\theta_{\text{av}}^t - \theta^\star\|_2^2 + \|\theta_{\text{av}}^{t+1} - \theta^\star\|_2^2\right)$$
$$+ 8\alpha m\tau_g(1+\epsilon)\delta,$$

where for convenience we set $\delta \triangleq \left(1 + \frac{1}{a}\right)\mathcal{R}^2(\Pi_{\mathcal{M}^\perp}(\theta^\star))$. Further, using Hölder's inequality and Young's inequality, for any $b_1, b_2 > 0$ it holds that

$$-2\alpha m\langle\nabla\mathcal{L}(\theta^\star), \theta^{t+1} - \theta^\star\rangle = -2\alpha m\langle\nabla\mathcal{L}(\theta^\star), \theta_{\text{av}}^{t+1} - \theta^\star\rangle - 2\alpha m\langle\nabla\mathcal{L}(\theta^\star), (I-\mathbf{J})\theta^{t+1}\rangle$$
$$\leq 2\alpha m\mathcal{R}^*(\nabla\mathcal{L}(\theta^\star))\mathcal{R}(\theta_{\text{av}}^{t+1} - \theta^\star) + 2\alpha m\|(I-\mathbf{J})\theta^{t+1}\|_2\|\nabla\mathcal{L}(\theta^\star)\|_2$$
$$\leq 4\alpha m\Psi(\bar{\mathcal{M}})\mathcal{R}^*(\nabla\mathcal{L}(\theta^\star))\|\theta_{\text{av}}^{t+1} - \theta^\star\|_2$$
$$+ 2\alpha m\mathcal{R}^*(\nabla\mathcal{L}(\theta^\star))\mathcal{R}(\Pi_{\mathcal{M}^\perp}(\theta^\star)) + b_2\|(I-\mathbf{J})\theta^{t+1}\|_2^2 + \frac{m^2\alpha^2}{b_2}\|\nabla\mathcal{L}(\theta^\star)\|_2^2 \leq$$
$$+ 2\alpha b_1 m\|\theta_{\text{av}}^{t+1} - \theta^\star\|_2^2 + \frac{2\alpha m}{b_1}\Psi^2(\bar{\mathcal{M}})\left(\mathcal{R}^*\left(\nabla\mathcal{L}(\theta^\star)\right)\right)^2 + 2m\alpha\mathcal{R}^*\left(\nabla\mathcal{L}(\theta^\star)\right)\mathcal{R}(\Pi_{\mathcal{M}^\perp}(\theta^\star))$$
$$+ b_2\|(I-\mathbf{J})\theta^{t+1}\|_2^2 + \frac{m^2\alpha^2}{b_2}\|\nabla\mathcal{L}(\theta^\star)\|_2^2.$$

Combining yields

$$2\alpha m\langle \nabla\bar{\mathcal{L}}(\boldsymbol{\theta}^t) - \nabla\mathcal{L}(\boldsymbol{\theta}^t), (\boldsymbol{\theta}^{t+1} - \boldsymbol{\theta}^\star)\rangle \leq (\alpha\gamma_{g,1} + \alpha\epsilon\gamma_{g,1}) \left(\|\mathbf{J}\boldsymbol{\theta}^t - \boldsymbol{\theta}^\star\|_{\Sigma'}^2 + \|\mathbf{J}\boldsymbol{\theta}^{t+1} - \boldsymbol{\theta}^\star\|_{\Sigma'}^2\right)$$

$$+ m\alpha\epsilon \left(\|\mathbf{J}\boldsymbol{\theta}^t - \boldsymbol{\theta}^\star\|_{\nabla^2\bar{\mathcal{L}}}^2 + \|\mathbf{J}\boldsymbol{\theta}^{t+1} - \boldsymbol{\theta}^\star\|_{\nabla^2\bar{\mathcal{L}}}^2\right) + \alpha\gamma_{g,2}\|\mathbf{J}\boldsymbol{\theta}^t - \mathbf{J}\boldsymbol{\theta}^{t+1}\|_{\Sigma'}^2$$

$$+ \left(1 + \frac{1}{\epsilon}\right)\alpha\eta \left(\|\boldsymbol{\theta}^t - \mathbf{J}\boldsymbol{\theta}^t\|_2^2 + \|\boldsymbol{\theta}^{t+1} - \mathbf{J}\boldsymbol{\theta}^{t+1}\|_2^2\right) + \frac{\alpha m}{\epsilon}\left(\|\boldsymbol{\theta}^t - \mathbf{J}\boldsymbol{\theta}\|_{\nabla^2\bar{\mathcal{L}}}^2 + \|\boldsymbol{\theta}^t - \mathbf{J}\boldsymbol{\theta}^t\|_{\nabla^2\bar{\mathcal{L}}}^2\right)$$

$$+ \alpha\tau_g(1+\epsilon)4(1+a)\Psi^2(\bar{\mathcal{M}})(\|\mathbf{J}\boldsymbol{\theta}^t - \boldsymbol{\theta}^\star\|_2^2 + \|\mathbf{J}\boldsymbol{\theta}^{t+1} - \boldsymbol{\theta}^\star\|_2^2) + 8\alpha m\tau_g(1+\epsilon)\delta$$

$$+ 2\alpha b_1\|\mathbf{J}\boldsymbol{\theta}^{t+1} - \boldsymbol{\theta}^\star\|_2^2 + b_2\|(I - \mathbf{J})\boldsymbol{\theta}^{t+1}\|_2^2 + \frac{m^2\alpha^2}{b_2}\|\nabla\bar{\mathcal{L}}(\boldsymbol{\theta}^\star)\|_2^2$$

$$+ \frac{2\alpha m}{b_1}\Psi^2(\bar{\mathcal{M}})\left(\mathcal{R}^*\left(\nabla\mathcal{L}(\boldsymbol{\theta}^\star)\right)\right)^2 + 2m\alpha\mathcal{R}^*\left(\nabla\mathcal{L}(\boldsymbol{\theta}^\star)\right)\mathcal{R}(\Pi_{\mathcal{M}^\perp}(\boldsymbol{\theta}^\star)).$$

For convenience, denote by $R_t^2 \triangleq \|\boldsymbol{\theta}^t - \boldsymbol{\theta}^\star\|_2^2$ and $C_t^2 \triangleq \|\boldsymbol{\theta}\|_{I-\mathbf{J}}^2$. Then, using the above to upper bound $R_{t+1}^2$ yields

$$R_{t+1}^2 \leq R_t^2 - \alpha m\left(\|\boldsymbol{\theta}^t - \boldsymbol{\theta}^\star\|_{\nabla^2\bar{\mathcal{L}}}^2 + \|\boldsymbol{\theta}^{t+1} - \boldsymbol{\theta}^\star\|_{\nabla^2\bar{\mathcal{L}}}^2\right) - (1-\rho)\left(C_{t+1}^2 + C_t^2\right)$$

$$(\alpha\gamma_{g,1} + \alpha\epsilon\gamma_{g,1})\left(\|\mathbf{J}\boldsymbol{\theta}^t - \boldsymbol{\theta}^\star\|_{\Sigma'}^2 + \|\mathbf{J}\boldsymbol{\theta}^{t+1} - \boldsymbol{\theta}^\star\|_{\Sigma'}^2\right) + m\alpha\epsilon\left(\|\mathbf{J}\boldsymbol{\theta}^t - \boldsymbol{\theta}^\star\|_{\nabla^2\bar{\mathcal{L}}}^2 + \|\mathbf{J}\boldsymbol{\theta}^{t+1} - \boldsymbol{\theta}^\star\|_{\nabla^2\bar{\mathcal{L}}}^2\right)$$

$$+ \alpha\tau_g(1+\epsilon)4(1+a)\Psi^2(\bar{\mathcal{M}})(R_t^2 + R_{t+1}^2) + 2\alpha b_1 R_{t+1}^2 + \left(1 + \frac{1}{\epsilon}\right)\alpha\eta(C_t^2 + C_{t+1}^2)$$

$$+ \frac{\alpha L}{\epsilon}(C_t^2 + C_{t+1}^2) + b_2 C_{t+1}^2 + \frac{m^2\alpha^2}{b_2}\|\nabla\bar{\mathcal{L}}(\boldsymbol{\theta}^\star)\|_2^2 + 8\alpha m\tau_g(1+\epsilon)\delta$$

$$+ \frac{2\alpha m}{b_1}\Psi^2(\bar{\mathcal{M}})\left(\mathcal{R}^*\left(\nabla\mathcal{L}(\boldsymbol{\theta}^\star)\right)\right)^2 + 2m\alpha\mathcal{R}^*\left(\nabla\mathcal{L}(\boldsymbol{\theta}^\star)\right)\mathcal{R}(\Pi_{\mathcal{M}^\perp}(\boldsymbol{\theta}^\star)) + \alpha\gamma_{g,2}\|\mathbf{J}\boldsymbol{\theta}^t - \mathbf{J}\boldsymbol{\theta}^{t+1}\|_{\Sigma'}^2$$

$$+ \alpha m\|\boldsymbol{\theta}^{t+1} - \boldsymbol{\theta}^t\|_{\nabla^2\bar{\mathcal{L}}}^2 - \sigma_m(W)\|\boldsymbol{\theta}^t - \boldsymbol{\theta}^{t+1}\|^2.$$

If $\Sigma' \preceq \nabla^2\bar{\mathcal{L}}$ Setting $b_1 = \frac{\mu}{8}$, $\epsilon = \frac{1}{8}$ $b_2 = \frac{1-\rho}{2}$, for $\alpha \leq \frac{\sigma_m(W)}{2L}$ we have

$$R_{t+1}^2 \leq R_t^2 - \frac{\alpha\mu}{4}R_t^2 + \frac{36}{8}\alpha(1+a)\tau_g\Psi^2(\bar{\mathcal{M}})(R_t^2 + R_{t+1}^2) - \frac{(1-\rho)}{2}(C_{t+1}^2 + C_t^2)$$

$$+ (9\alpha\eta + 8\alpha L)\left(C_t^2 + C_{t+1}^2\right) + \frac{m^2\alpha^2}{b_2}\|\nabla\bar{\mathcal{L}}(\boldsymbol{\theta}^\star)\|_2^2 + 9\alpha m\tau_g\delta + 16\alpha m\frac{\Psi^2(\bar{\mathcal{M}})}{\mu}\left(\mathcal{R}^*\left(\nabla\mathcal{L}(\boldsymbol{\theta}^\star)\right)\right)^2$$

$$+ 2m\alpha\mathcal{R}^*(\nabla\mathcal{L}(\boldsymbol{\theta}^\star))\mathcal{R}(\Pi_{\mathcal{M}^\perp}(\boldsymbol{\theta}^\star)).$$

For the terms in $C_t^2$ and $C_{t+1}^2$ to vanish from the RHS of the above inequality we require that

$$\alpha \leq \min\left\{\frac{\sigma_m(W)}{2L}, \frac{1-\rho}{2(9\eta + 8L)}\right\}.$$

Then, for convenience, define

$$\zeta \triangleq \frac{36}{8}(1+a)\tau_g\Psi^2(\bar{\mathcal{M}}).$$

Given that $\zeta$ is such that $1 - \alpha\zeta > 0$ we have by rearranging and dividing both sides by $1 - \alpha\zeta$ we obtain

$$R_{t+1}^2 \leq \underbrace{\left(1 - \frac{\alpha\mu}{4} + \alpha\zeta\right)(1 - \alpha\zeta)^{-1}}_{\triangleq \lambda} R_t^2 +$$

$$(1 - \alpha\zeta)^{-1}\left(\frac{m^2\alpha^2}{b_2}\|\nabla\bar{\mathcal{L}}(\boldsymbol{\theta}^\star)\|_2^2 + 9\alpha m\tau_g\delta\right)$$

$$+ (1 - \alpha\zeta)^{-1}\left(16\alpha m\frac{\Psi^2(\bar{\mathcal{M}})}{\mu}\left(\mathcal{R}^*\left(\nabla\mathcal{L}(\boldsymbol{\theta}^\star)\right)\right)^2 + 2m\alpha\mathcal{R}^*(\nabla\mathcal{L}(\boldsymbol{\theta}^\star))\mathcal{R}(\Pi_{\mathcal{M}^\perp}(\boldsymbol{\theta}^\star))\right).$$

Observe that if

$$\zeta \leq \frac{\mu}{16}$$

$\lambda$ is at most $1 - \frac{\gamma\mu}{8}$ and therefore by telescoping

$$R_{t+1}^2 \leq \lambda^{t+1} R_0^2 + (1-\alpha\zeta)^{-1} \left( \frac{16m^2\alpha}{1-\rho} \|\nabla\bar{\mathcal{L}}(\boldsymbol{\theta}^\star)\|_2^2 + \frac{128m}{\mu^2} \Psi^2(\bar{\mathcal{M}})(\mathcal{R}^*(\nabla\mathcal{L}(\boldsymbol{\theta}^\star)))^2 \right)$$
$$+ (1-\alpha\zeta)^{-1}(1-\lambda)^{-1}\alpha \left( 9m\tau_g\delta + 2m\alpha\mathcal{R}^*\left(\nabla\mathcal{L}(\boldsymbol{\theta}^\star)\right)\mathcal{R}(\Pi_{\mathcal{M}^\perp}(\boldsymbol{\theta}^\star)) \right).$$

Setting $a=1$, using that $(1-\alpha\zeta)^{-1} = 1 + \frac{\alpha\zeta}{1-\alpha\zeta}$ substituting the definition of $\delta$ and dividing by $m$ yields the desired result.

The following Lemma contains results analogous to those in Lemma 4 required for the proof of DGD-CTA.

**Lemma 7.** *Under Assumptions 1, 3 and (53) for any $\epsilon > 0$ it holds*

$$\langle(\nabla^2\bar{\mathcal{L}} - \nabla^2\mathcal{L})(\mathbf{J}\boldsymbol{\theta} - \boldsymbol{\theta}^\star), (I-\mathbf{J})\boldsymbol{\theta}'\rangle \leq \frac{\epsilon\gamma_{g,1}}{2m}\|\mathbf{J}\boldsymbol{\theta} - \boldsymbol{\theta}^\star\|_{\Sigma'}^2 + \frac{\epsilon\tau_g}{2}\mathcal{R}^2(\boldsymbol{\theta}_{\mathrm{av}} - \boldsymbol{\theta}^\star) + \frac{\epsilon}{2}\|\mathbf{J}\boldsymbol{\theta} - \boldsymbol{\theta}^\star\|_{\nabla^2\bar{\mathcal{L}}}^2$$
$$+ \frac{\eta}{2\epsilon m}\|\boldsymbol{\theta}' - \mathbf{J}\boldsymbol{\theta}'\|_2^2 + \frac{1}{2\epsilon}\|\boldsymbol{\theta}' - \mathbf{J}\boldsymbol{\theta}'\|_{\nabla^2\bar{\mathcal{L}}}^2,$$

*and*

$$\langle(\nabla^2\bar{\mathcal{L}} - \nabla^2\mathcal{L})(\boldsymbol{\theta} - \mathbf{J}\boldsymbol{\theta}), \boldsymbol{\theta}' - \mathbf{J}\boldsymbol{\theta}'\rangle \leq \frac{\eta}{2m} \left( \|\boldsymbol{\theta} - \mathbf{J}\boldsymbol{\theta}\|_2^2 + \|\boldsymbol{\theta}' - \mathbf{J}\boldsymbol{\theta}'\|_2^2 \right).$$

*Proof.* We start with

$$\langle(\nabla^2\bar{\mathcal{L}} - \nabla^2\mathcal{L})(\mathbf{J}\boldsymbol{\theta} - \boldsymbol{\theta}^\star), (I-\mathbf{J})(\boldsymbol{\theta}' - \boldsymbol{\theta}^\star)\rangle = -\langle\nabla^2\mathcal{L}(\mathbf{J}\boldsymbol{\theta} - \boldsymbol{\theta}^\star), (I-\mathbf{J})(\boldsymbol{\theta}' - \boldsymbol{\theta}^\star)\rangle$$
$$\leq \frac{\epsilon}{2}\|(\nabla^2\mathcal{L})^{1/2}(\mathbf{J}\boldsymbol{\theta} - \boldsymbol{\theta}^\star)\|_2^2 + \frac{1}{2\epsilon}\|(\nabla^2\mathcal{L})^{1/2}(\boldsymbol{\theta}' - \mathbf{J}\boldsymbol{\theta}')\|_2^2$$

for any $\epsilon > 0$. Then,

$$\langle(\nabla^2\bar{\mathcal{L}} - \nabla^2\mathcal{L})(\mathbf{J}\boldsymbol{\theta} - \boldsymbol{\theta}^\star), (I-\mathbf{J})(\boldsymbol{\theta}' - \boldsymbol{\theta}^\star)\rangle \leq \frac{\epsilon}{2}\langle(\nabla^2\mathcal{L} - \nabla^2\bar{\mathcal{L}})(\mathbf{J}\boldsymbol{\theta} - \boldsymbol{\theta}^\star), (\mathbf{J}\boldsymbol{\theta} - \boldsymbol{\theta}^\star)\rangle$$
$$+ \frac{\epsilon}{2}\|\mathbf{J}\boldsymbol{\theta} - \boldsymbol{\theta}^\star\|_{\nabla^2\bar{\mathcal{L}}}^2 + \frac{1}{2\epsilon}\langle\nabla^2\bar{\mathcal{L}}(I-\mathbf{J})\boldsymbol{\theta}', (I-\mathbf{J})\boldsymbol{\theta}'\rangle + \frac{1}{2\epsilon}\|\boldsymbol{\theta}' - \mathbf{J}\boldsymbol{\theta}'\|_{\nabla^2\bar{\mathcal{L}}}^2.$$

Under Assumption 3 and (53) it follows that

$$\langle(\nabla^2\bar{\mathcal{L}} - \nabla^2\mathcal{L})(\mathbf{J}\boldsymbol{\theta} - \boldsymbol{\theta}^\star), (I-\mathbf{J})(\boldsymbol{\theta}' - \boldsymbol{\theta}^\star)\rangle \leq \frac{\epsilon\gamma_{g,1}}{2m}\|\mathbf{J}\boldsymbol{\theta} - \boldsymbol{\theta}^\star\|_{\Sigma'}^2$$
$$+ \frac{\epsilon\tau_g}{2}\mathcal{R}^2(\boldsymbol{\theta}_{\mathrm{av}} - \boldsymbol{\theta}^\star) + \frac{\epsilon}{2}\|\mathbf{J}\boldsymbol{\theta} - \boldsymbol{\theta}^\star\|_{\nabla^2\bar{\mathcal{L}}}^2 + \frac{\eta}{2\epsilon m}\|\boldsymbol{\theta}' - \mathbf{J}\boldsymbol{\theta}'\|_2^2 + \frac{1}{2\epsilon}\|\boldsymbol{\theta}' - \mathbf{J}\boldsymbol{\theta}'\|_{\nabla^2\bar{\mathcal{L}}}^2.$$

Finally, from (53)

$$\langle(\nabla^2\bar{\mathcal{L}} - \nabla^2\mathcal{L})(\boldsymbol{\theta} - \mathbf{J}\boldsymbol{\theta}), (\boldsymbol{\theta}' - \mathbf{J}\boldsymbol{\theta}')\rangle \leq \frac{\eta}{2m}\|\boldsymbol{\theta} - \mathbf{J}\boldsymbol{\theta}\|_2^2 + \frac{\eta}{2m}\|\boldsymbol{\theta}' - \mathbf{J}\boldsymbol{\theta}'\|_2^2.$$

$\square$

In order to establish the convergence of DGD-CTA when used to solve the Gaussian $\ell_1$ constrained regression problem (c.f. Corollary 1) we require that with high probability (53) holds. The following Lemma states that the condition holds with high probability for all agents $i \in [m]$.

**Lemma 8.** *Under the data model described above, for all $i \in [m]$ it holds that*

$$\left\| \frac{X_{S_i}^\top X_{S_i}}{n} - \Sigma \right\|_2 \leq 2L \left( 3\sqrt{\frac{d}{n}} + 4\frac{d}{n} \right)$$

*with probability at least*

$$1 - 2\exp(-d/2 + \log(m)).$$

*Proof.* See Example 6.3 in [30] for the single agent statement. Then apply the union bound. $\square$

The above lemma implies that by taking the union bound with the statements in Lemma 1 the requirements for convergence of DGD CTA are met. However, observe that $\eta$ in (53) corresponds to

$$\eta = 2L\left(3\sqrt{\frac{d}{n}} + 4\frac{d}{n}\right).$$

Further, the ball $\Delta_{\text{net}}^2$ scales as

$$m\alpha\|\nabla\boldsymbol{\mathcal{L}}(\boldsymbol{\theta}^\star)\|_2^2 = \frac{\alpha}{m}\sum_{i=1}^{m}\|\nabla\mathcal{L}_i(\theta^\star)\|_2^2,$$

which corresponds to the rate of standard linear regression with random designs which with high probability scales as $\mathcal{O}\left(\alpha\frac{d}{n}\right)$ [10].