# OpenReview forum: "DGD^2: A Linearly Convergent Distributed Algorithm For High-dimensional Statistical Recovery"
_NeurIPS.cc/2022/Conference — NeurIPS 2022 Accept_

### Official Review · Reviewer_VuFR · 2022-07-10

**Rating:** 7
**Confidence:** 3
**Soundness:** 3 good
**Presentation:** 2 fair
**Contribution:** 3 good

**Summary:**

This paper analyzes DGD in a high-dimensional statistical estimation setting, which converges at the same rate as its centralized counterpart. This paper also provides several specific statistical estimation problems, and shows how the algorithm adapts to these problems.

**Questions:**

1. Line 22, should "undetermined" be "underdetermined"?

**Limitations:**

1. Is it possible to extend the analysis to stochastic updates?

**Strengths And Weaknesses:**

Strengths:
1. This paper analyzes DGD for high-dimensional statistical estimation problems, which is a novel setting.
2. The analysis takes into account both the eigen gap of the weight matrix and the condition number.
3. This paper shows how the general analysis can be applied to sparse vector regression and matrix completion problems.

---

> ### Author Response · Authors · 2022-08-02
> **Reply to Reviewer VuFR**
>
> We thank the Reviewer for the positive assessment of the paper, which captures the main contributions.
>
> The Reviewer asked a very good question about the potential extension of the method to the stochastic setting. While the algorithm is technically applicable using stochastic gradients, the analysis would follow a different path, which is worth exploring. There are works in the centralized setting and high-dimension following this route, such as   *Stochastic optimization and sparse statistical recovery: Optimal algorithms for high dimensions'*. This makes the extension to the distributed case promising although not trivial, given the ``sensitivity'' of distributed gradient estimators on what information is mixed over the network.  Thanks for the suggestion.

---

### Official Review · Reviewer_5YHX · 2022-07-12

**Rating:** 8
**Confidence:** 5
**Soundness:** 4 excellent
**Presentation:** 4 excellent
**Contribution:** 4 excellent

**Summary:**

This manuscript considers solving M-estimation problems over a network of agents, where the dimension d of the decision variable is much larger than the number of samples (d>>N), thus necessitating a regularizer that enforces the regularity of solution. Due to d>>N, the problem is usually ill-conditioned and do not have desirable strong convexity (why may be 0) and smoothness (which may grow unbounded with d). Thus, the authors design an algorithm building on DGD as well as a double-mixing scheme, which turns out to be enjoying a global linear convergence up to the statistical error of the model (independent of the dimension d by ignoring log factors), which is comparable to that of the centralized counterpart. The theoretical analysis they used to prove this result is based on the key idea that the gap between the gradient of ERM loss and population loss can be made arbitrarily small with multiple rounds of consensus (i.e., a sufficient number of mixing of gradients among agents). They have also developed similar statistical guarantees for several popular examples in M-estimation, such as sparse vector regression and Gassian matrix regression.

**Questions:**

- The proposed DGD^2 algorithm shares a very similar form with the adapt-then-combine (ATC) version of DGD algorithms. The reviewer would thus like to know if DGD of ATC version also works. If yes, this would save some communication cost since it only needs a single mixing at each iteration. If not, what is the key difference?

- Will it be possible to extend the theoretical results developed in this work to stochastic algorithms which is necessary when it comes to large number of samples?

- It seems that the algorithm needs to transmit variables of d dimension, incurring high communication burden as d is very large; will it be possible to reducing the communication cost?


**Limitations:**

It would be desirable if the authors can provide more insights in designing the algorithm and illustrate why the double mixing is needed to obtain the results. The reviewer guess that distributed algorithms may work as long as they employ an extra mixing on gradients.

**Strengths And Weaknesses:**

The paper is very well written and easy to follow. The M-estimation problem considered under the high-dimension setting is very important and definitely of great interest to the machine learning community. The algorithm proposed, along with the corresponding proofs, is very interesting and seems new to the reviewer.

Strengths:
- The paper proposes a new algorithm and proves that it enjoys a linear convergence up to a tolerance at the same order of statistical error without scaling with the dimension d. This is a new important result which is not known before to the best knowledge of the reviewer.
- The techniques used for proving the main result seems novel and the reviewer have checked most of the proofs which seems correct.
- The paper developed statistical guarantees for several popular examples in M-estimation, which may be beneficial to the machine learning community

Weaknesses:
- In the main result of Theorem 1, it requires an extra constraint (c.f. (14)) on the connectivity of the network, which is scaling with the condition number \kappa and the number of nodes m. This requirement may be difficult to meet in the actual decentralized networks, especially when the value of \kappa is large.

---

> ### Author Response · Authors · 2022-08-02
> **Reply to the Reviewer 5YHX - Part I**
>
> We thank the Reviewer for her/his careful assessment of our work, which clearly captures the key contributions of our work,  and for her/his insightful questions. Our reply to the comments/questions follows.
>
> $\textbf{1. On the requirement on the network connectivity:}$ We agree with the Reviewer that the condition on the network connectivity $\\rho$, as in Eq. (14), can become hard to satisfy a-priori from any given network and weight matrix $W$, when $\\kappa$ or $m$ are large. When this is the case,  for any given arbitrary network topology and associated weight matrix $W$, the aforementioned condition can still be enforced  just running  multiple steps of consensus in steps  [S.1]-[S.2] per iteration $t$. It is not difficult to check that, mathematically, $K$ consecutive rounds of consensus  per iteration using each time weights  $W=[w\_{ij}]$  (with associated $\\rho=||W-J||$) correspond to use the same updates as in [S.1] and [S.2]  in the paper  but with a new set of weights, say   $W'=[w\_{ij}']\_{ij=1}^m$, satisfying the property: $W'=W^K$. Denoting by  $\\rho'= ||W'-J||$ the network connectivity of $W'$, the above relationship implies   $$\\rho'=\\rho^K.$$  Roughly speaking,   $K$ rounds of communications as above induce an ``effective'' network connectivity $\\rho'$. Therefore, the condition in Corollary 1  (or similarly in Eq. (14)) on the network connectivity, $\\rho \\leq C_2 m^{-2.5}\\kappa^{-1}$, under $K$ rounds of communications, becomes
> $$  \\rho'\leq C m^{-2.5} \\kappa^{-1}\\quad \\Rightarrow {\\rho}^K\\leq  C m^{-2.5} \\kappa^{-1},$$ which is satisfied when the number of communications rounds $K$ is    $$ K \\geq \\frac{1}{1-{\\rho}} \\log \\big(\\frac{m^{2.5}\\kappa}{C}\\big).$$ This shows that, for any given graph and weight matrix $W$ (with associate, arbitrarily large  connectivity $\\rho\\in [0,1)$), the conditions on $\\rho$ in Eq. (18) in the paper can *always* be satisfied by employing $K$ rounds of communications as above, resulting in a communication cost per iteration of $${\\mathcal{O}}(\\frac{\\log (m\\,\\kappa)}{1-\\rho}).$$  Note that this multiple rounds of communications are already incorporated in the total communication complexity $$\\widetilde{\\mathcal{O}}\\left(\\sqrt{\\kappa}\\frac{\\log m}{1-\\rho}\\log1/\\varepsilon\\right),$$ as commented in the paper.
>
> Therefore, ill conditioned population losses will translate in a $\\log \\kappa$ increase on the number of communications.

---

> > ### Author Response · Authors · 2022-08-02
> > **Reply to the Reviewer 5YHX - Part III**
> >
> > $\textbf{Reply to Question 2:}$ This is an interesting suggestion. While the algorithm is technically applicable using stochastic gradients, the analysis would follow a different path, which is worth exploring. There are works in the centralized setting and high-dimension following this route, such as   *Stochastic optimization and sparse statistical recovery: Optimal algorithms for high dimensions'*. This makes the extension to the distributed case promising although not trivial, given the ``sensitivity'' of distributed gradient estimators on what information is mixed over the network.  Thanks for the suggestion.
> >
> > $\textbf{Reply to Question 3:}$  This is a very good question. The Reviewer is correct:  the proposed scheme, as all the other distributed algorithms, transmits vectors of dimension $d$ at each iteration. On the positive side, the total communication cost scales with   $\\mathcal{O}(d)$ rather than $\\mathcal{O}(d^2)$ (DGD-CTA) or  $\\mathcal{O}(d\\log d)$ (DGD-ATC). However, we agree that this can be a significant cost in high-dimension.
> >
> > On the other hand, since it seems that transmitting the gradient is the key enabler to achieve $\\mathcal{O}(d)$ communication cost, and unfortunately the gradient in general is not sparse (even if the iterates are so), it is unclear whether sparsity in the estimates can help to transmit only a subset of components of the gradients (and iterates) at each iterations (e.g., invoking some hard-thresholding technique) while preserving *centralized* statistical consistency or without requiring conditions on the minimum local sample size (more restrictive than what suggested by information theoretical lower bounds).
> >
> > Our conjecture is that it is not possible to achieve centralized statistical consistency in sparse linear regression over mesh networks with a communication cost sublinear with the dimension (without extra assumptions on the model). This conjecture seems to be consistent with the following facts.   **Fact 1:** In the context of Divide & Conquer methods on star-networks, no method can achieve centralized statistical consistency without transmitting $\\mathcal{O}(d)$ scalars (let alone mentioning that extra conditions on the local samples size or maximum number of nodes are made).  **Fact 2:** It is known ``Communication lower bounds for statistical estimation problems via a distributed data processing inequality'' that it is impossible to estimate the $s$-sparse average of $d$ dimensional standard Gaussian i.i.d. random variables over a network without exchanging $\\mathcal{O}(d)$ bits. This implies that some information regarding each of the coordinates must be exchanged. Consequently, in a more complicated problem (sparse mean estimation would be a particular case of sparse linear regression) it seems unlikely we can obtain satisfactory guarantees without transmitting information regarding each of the coordinates.

---

> > ### Author Response · Authors · 2022-08-02
> > **Reply to the Reviewer 5HYX - Part II**
> >
> > $\\textbf{Reply to Question 1:}$ This is a very good question, which interestingly complements Questions 1&3 of Reviewer  jZqb, and  will offer us the chance to provide an interesting picture of the behavior of DGD-ATC, DGD-CTA and DGD$^2$ in high-dimensions. Following the same argument as in the Reply of Reviewer jZqb, we can rewrite DGD-ACT and DGD$^2$ based upon the type of approximation they employ of the gradient of the population loss $\\bar{\\mathcal{L}}$ as:
> > $$\\text{DGD-ATC}:\\quad \\theta\_i^{t+1} = \\text{argmin}\_{\\theta: \\mathcal{R}(\\theta)\\leq r}   \\quad \\langle \\sum\_{j=1}^m w\_{i,j} \\nabla \\mathcal{L}\_j(\\theta\_j^t), \\theta  \\rangle + \\frac{1}{2\\alpha}||\\sum\_{j=1}^m w\_{i,j}\\theta\_j -\\theta ||^2  \\quad(1) $$
> >
> > $$\\text{DGD}^2:\\quad \\theta\_i^{t+1}= \\text{argmin}\_{\\theta:\\mathcal{R}(\\theta) \\leq r} \\quad \\langle \\sum_{j=1}^m w\_{i,j} \\nabla \\mathcal{L}\_{j}(\\sum\_{l=1}^m w\_{j,l}\\theta\_l^t),\\theta\\rangle  + \\frac{1}{2\\alpha}\\sum\_{j=1}^m w\_{i,j} || \\theta - \\sum\_{l=1}^m w_{j,l}\\theta\_l^t ||^2 \\quad(2)$$
> > In the DGD-ATC an estimate of the population gradient $\\nabla \\bar{\\mathcal{L}}$ is formed  by   the sample approximation (see the linear term in (1)): $$
> > \\sum\_{j} w\_{ij} \\nabla {\\mathcal{L}}\_{j} (\\theta\_{j}^{t})\\quad (3) $$
> > while $\\text{DGD}^2$ leverages  (see linear term in (2)):
> > $$\\sum_{j=1}^m w\_{i,j} \\nabla \\mathcal{L}\_{j}(\\sum\_{l=1}^m w\_{j,l}\\theta\_l^t) \\quad (4)$$
> > which is a better approximation than that of DGD-ACT in (3)  because first a consensus step is performed among the local copies--replacing $\\theta\_j^t$ in $\\nabla \\mathcal{L}\_j(\\theta\_j^t)$ with $\\sum\_{l} w\_{jl}\\theta\_{l}^t$--and then the local gradients $\\nabla \\mathcal{L}\_j$ are evaluated in such a point, producing (4).
> >
> > Therefore, under the same number of communications, DGD$^2$ produces a better estimate of the population gradient than that of DGD-ATC, resulting in a less demanding condition on the network connectivity. In fact, a more formal analysis of DGD-ATC following the population-sample machinery developed in the paper shows that the error resulting from the the DGD-ATC approximation of the population gradient can be controlled by the network as long as the network connectivity $\\rho$ scales with $\\mathcal{O}(1/d)$, resulting in $$\\mathcal{O}\\left(\\frac{\\log d}{1-\\rho}\\right)$$ communications per iteration versus the $$\\mathcal{O}\\left(\\frac{\\log m}{1-\\rho}\\right)$$ of  DGD$^2$.
> >
> > In summary, if we contrast the three DGD-like schemes above, we can conclude that the total communication rounds of the three schemes to achieve centralized statistical consistency will scale as:
> >
> > $$\\bullet\\,\\text{DGD-CTA (no mixing of the local gradients)}: \\mathcal{O}(d)$$ (due to the constraint on the stepwise $\alpha=\mathcal{O}(1/d)$)
> >
> > $$\\bullet\\,\\text{DGD-ATC (mixing of the local gradients and iterates)}: \\mathcal{O}(\log d)$$ (due to the constraint on the network connectivity  $\rho=\mathcal{O}(1/d)$)
> >
> > $$\bullet\\,\text{DGD}^2 \\text{ (double mixing)}: \\mathcal{O}(\\log m)$$ (due to the constraint on the network connectivity  $\rho=\mathcal{O}(1/m^{2.5})$)

---

> ### Comment · Reviewer_5YHX · 2022-08-10
> **Thank you for the detailed and insightful response**
>
> The reviewer would like to thank the authors for their detailed and insightful responses which have satisfactorily addressed the reviewer's concerns. The reviewer finds this work valuable and thinks that it might spark some sebsequent works towards improving communication efficiency. The reviewer is thus happy to maintain the score.

---

### Official Review · Reviewer_JZqb · 2022-07-14

**Rating:** 7
**Confidence:** 4
**Soundness:** 3 good
**Presentation:** 3 good
**Contribution:** 3 good

**Summary:**

In this paper, the authors propose a distributed algorithm that converges linearly to the statistical precision for high-dimensional linear regression problems.

**Questions:**

1. There are two possible reasons to achieve the better dependency on d/N: (a) exchanging both parameters and gradients; (b) introducing multiple rounds of communication. The authors claim the reason is (a). Is it possible to show that even with (b), we cannot obtain the expected dependency on d/N?
2. Related to the previous comment, can the authors give some insights about why (a) helps us improve the dependency on d/N?
3. In the numerical experiments, the ratios d/N are not large. This contradicts with the motivation.


**Ethics Review Area:**

["I don’t know"]

**Limitations:**

N/A.

**Strengths And Weaknesses:**

Strengths:

1. Overall this is a good paper. The investigated problem is important. The favorable dependency on d/N is novel.
2. I did not check the proofs, but the analysis seems sound.
3. The presentation is smooth, although this paper is mathematically extensive.

Weaknesses:

1. There are two possible reasons to achieve the better dependency on d/N: (a) exchanging both parameters and gradients; (b) introducing multiple rounds of communication. The authors claim the reason is (a). Is it possible to show that even with (b), we cannot obtain the expected dependency on d/N?
2. Related to the previous comment, can the authors give some insights about why (a) helps us improve the dependency on d/N?
3. In the numerical experiments, the ratios d/N are not large. This contradicts with the motivation.

---

> ### Author Response · Authors · 2022-08-02
> **Reply to Reviewer jZqb**
>
> We are grateful to the Reviewer for her/his careful assessment of our work and  comments. We are glad that the Reviewer liked our work. Our reply to her/his questions (as listed in the review) follows.
>
> $\textbf{Reply to 1 and 2:}$ Thanks for asking such a good question, which will help to provide some intuition on why DGD$^2$ is superior in high-dimension to other distributed algorithms, in particular those that do not mix the local gradients.  The Reviewer is suggesting  very plausible explanations on the reason why the rate can be made independent on  $\frac{d}{N}$. While we do not have a converse result, we believe that such a desideratum calls for the exchange of the local gradients over the network whereas  a good network connectivity (or equivalently multiple rounds of communications) alone is not enough. This conjecture is supported by the following facts/insights.
>
> $\bullet\\, \textbf{Fact:}$ In Theorem 3,  Appendix H, we study the convergence and statistical guarantees of the  DGD-CTA algorithm  [13], which is a notorious example of decentralization of the projected gradient algorithm whereby agents exchange only the iterates but not the local gradients--see Eq. (4) in the paper.  This algorithm is proved to achieve centralized statistical optimal solutions in $$\mathcal{O}\left(d/n \frac{\kappa}{1-\rho} \log(1/\varepsilon)\right)$$ iterations. This is the case *regardless of the network connectivity*, meaning that even setting $\rho = 0$ (equivalent to having infinitely many communication rounds) does not improve the dependence on $\frac{d}{n}.$ This conclusion is also supported by extensive numerical results, see [13].
>
> $\bullet\\, \textbf{Some insights:}$  The mixing mechanism (of the iterates alone vs. iterates and local gradients) affects the type of approximation of the gradient of the population loss $\overline{\mathcal{L}}$ agents build over the network,  and thus the ''effective'' loss landscape the algorithm travels over.   This is clear contrasting, e.g.,  DGD-CTA (a scheme that does not exchange the gradients)  and the proposed   DGD$^2$ (which mixes gradient and iterates) under the *same* number of communications (thus two rounds also for DGD). Specifically, DGD-CTA can be equivalently written as
>
> $$\\theta_i^{t+1} = \\text{argmin}_{\\theta \\in \\Omega, \\mathcal{R}(\\theta)\\leq r} \\text{  } \\langle  \\nabla \\mathcal{L}(\\theta_i^t),\\theta\\rangle + \\frac{1}{2\\alpha}\\sum\_{j=1}^m w\_{i,j} || \theta - \\sum\_{l=1}^m w\_{j,l}\theta\_l^t ||^2 \\quad(1), $$
>
> while DGD$^2$ can be written as
>
> $$\\theta_i^{t+1} = \\text{argmin}_{\\theta \\in \\Omega, \\mathcal{R}(\\theta) \\leq r} \\text{  } \\langle \\sum\_{i=1}^m w\_{i,,j} \\nabla \\mathcal{L}\_j(\\sum\_{l=1}^m w\_{l,j}\\theta\_j^t),\\theta  \\rangle + \\frac{1}{2\alpha}\\sum\_{j=1}^m w\_{i,j}|| \\theta - \\sum\_{l=1}^m w\_{j,l}\\theta\_{l}^t||^2\\quad(2) $$
>
> Both schemes perform now two rounds of communications on the local   parameters $\\theta\_i^t$'s, represented by the proximal term in (1) and (2). In   DGD-CTA, this update is followed by a correction whereby  each agent  approximates the population gradient  $\\nabla \\bar{\\mathcal{L}}(\\theta\_i^t)$ using its own sample gradient (see linear term in (1)) $$\\nabla \\bar{\\mathcal{L}}(\\theta\_i^t)\\approx\\nabla \\mathcal{L}\_{i}(\\theta\_{i}^{t})\\quad (3)$$  while     DGD$^2$ employs (see linear term in (2)) $$\\nabla \\bar{\\mathcal{L}}(\\theta\_i^t)\\approx \\sum\_{j=1}^m w\_{ij}\\nabla \\mathcal{L}\_j \\big({\\sum\_{j'=1}^m w\_{jj'}\\theta_{j'}^t}\\big).\\quad (4)$$
>
> When the mixing is ``fast'' enough ($\rho$ sufficiently small, which explains why we have a condition on $\rho$), (4) is a more accurate approximation of $\nabla \overline{\mathcal{L}}(\theta_i^t)$ than (3) because it can benefit from aggregating gradients *and*  parameters $\theta_j$'s of neighbors.  This also explains why just increasing the number of communications in (1)   cannot improve the scaling of DGD: the linear term in (1)  does not change. As already mentioned, the above interpretation is supported by numerical experiments, see [13]:   even under infinitely many communication rounds per gradient computations (resulting in $\rho = 0$), the number of gradient iterations of DGD-CTA to achieve centralized statistical consistency still  scales   linearly with $d$.
>
> $\textbf{Reply to 3:}$ We provided an additional simulation in Appendix B, see  Figure 3, with larger values of $\frac{d}{N}.$ Given the space limit, we prioritized in the main paper simulations supporting the theoretical  invariance of the rate and final statistical estimates with respect to $\frac{s \log d}{N}$ and empirical evidence of the requirements on network connectivity.

---

> > ### Comment · Reviewer_JZqb · 2022-08-04
> > **Update**
> >
> > Thank the authors for the explanations. Questions 1 and 2 seem not easy to answer, but I believe that the authors have tried the best to give some insights. I am also satisfied with the replies to the other reviewers, and would like to keep the score of 7.

---

### Meta-Review · Area_Chair_Cukk · 2022-08-23

**Recommendation:** Accept
**Confidence:** Certain

**Metareview:**

This paper studies the problem of high-dimensional linear regression in a decentralized setting. A distributed algorithm, incorporating so-called "double mixing" with decentralized gradient descent, is proposed that converges linearly to statistical precision in the regime where the problem dimension is much larger than the number of samples. The paper is well-written, clearly positions the contributions with respect to previous work, and provides significant results. The reviewers are also unanimous in recommending that the work be accepted, and recognizing the contribution as significant.

The reviewers found the post-rebuttal discussion to be very informative, and we recommend that the authors expand on the following points when preparing the camera ready manuscript:
* Adding some discussion of communication overhead associated with double mixing
* The tradeoff between double mixing (parameters and gradients) vs. additional rounds of communication to cope with poor conditioning
* Discussion of potential challenges extending the results to the stochastic gradient setting


**Award:**

No

---

### Decision · Program_Chairs · 2022-09-14

Accept